# A Capacity-Based Rationale for Multi-Head Attention

Micah Adler [1]

## Abstract

We study the capacity of the self-attention key–query channel: for a fixed budget, how many distinct token–token relations can a single layer reliably encode? We introduce *Relational Graph Recognition*, where the key–query channel encodes a directed graph and, given a context (a subset of the vertices), must recover the neighbors of each vertex in the context. We measure resources by the total key dimension $D_K = h\,d_k$. In a tractable multi-head model, we prove matching information-theoretic lower bounds and upper bounds via explicit constructions showing that recovering a graph with $m'$ relations in $d_{\text{model}}$-dimensional embeddings requires $D_K$ to grow essentially as $m'/d_{\text{model}}$ up to logarithmic factors, and we obtain corresponding guarantees for scaled-softmax attention. This analysis yields a new, capacity-based rationale for multi-head attention: even in permutation graphs, where **all queries attend to a single target**, splitting a fixed $D_K$ budget into multiple heads increases capacity by reducing interference from embedding superposition. Controlled experiments mirror the theory, revealing sharp phase transitions at the predicted capacity, and the multi-head advantage persists when adding softmax normalization, value routing, and a full Transformer block trained with frozen GPT-2 embeddings.

## 1. Introduction

At the core of the Transformer architecture is self-attention, a mechanism that computes a similarity-weighted pattern of pairwise relationships among items in a context: queries match keys; the resulting scores route information via values (Vaswani et al., 2017; Santoro et al., 2017). We ask a basic question: for a fixed self-attention mechanism size, how many target relationships can a single attention layer

represent and reliably recover? We call this the layer's *capacity*. Capacity is foundational for several reasons. (i) It imposes a hard ceiling on relational computation: beyond a threshold, no training procedure or dataset can make a layer recover all relations, much like rank bounds in linear models. (ii) It complements mechanistic work that isolates specific attention circuits in trained transformers (Clark et al., 2019; Vig & Belinkov, 2019; Olsson et al., 2022; Kamath et al., 2025), by asking how many independent circuits can coexist and quantifying the interference that limits their coexistence—recently termed the "Semantic Stability Gap" (Zahn et al., 2026). (iii) It provides an actionable resource scaling law, describing how relationship capacity grows with increasing attention budget.

We also demonstrate that capacity impacts when increasing the number of heads is useful. Multiple heads are often conceived as a way for a source concept to attend to multiple different targets (Vaswani et al., 2017), but looking at self-attention through the lens of capacity shows that multiple heads are beneficial even in the simple case where each concept attends to only a single target. Specifically, when the embedding dimension is smaller than the vocabulary size, the item embeddings cannot all be near-orthogonal: they share subspace directions, so each carries spurious correlations with the others—*embedding superposition*—which appears in our analysis as interference noise in the attention scores. Distributing the self-attention budget across many small heads reduces this interference and increases the number of cleanly separable relations—consistent with both pruning/specialization studies and expressivity results for attention (Voita et al., 2019; Michel et al., 2019; Cordonnier et al., 2020b).

One might hope to answer capacity empirically by probing large trained models, but this is ill-posed: heads are polyfunctional and context-dependent, so the number of "active" relations is not directly observable; attention weights need not align with causal importance (Jain & Wallace, 2019); and circuit-tracing pipelines still miss parts of the QK computation that determines *where* a head attends (Kamath et al., 2025). Superposition compounds the problem (Elhage et al., 2022; Bricken et al., 2023), so interpretability work has not revealed how many relationships can be supported by a fixed attention budget.

[1]MIT CSAIL, Cambridge MA, USA. Correspondence to: Micah Adler <micah@csail.mit.edu>.

*Proceedings of the 43rd International Conference on Machine Learning*, Seoul, South Korea. PMLR 306, 2026. Copyright 2026 by the author(s).

We therefore introduce a *framework*—**Relational Graph Recognition (RGR)**—and analyze an idealized self-attention *model* for solving RGR. The framework allows us to explicitly control both the structure and the number of attention relationships by casting self-attention as recovering edges of a relational graph among $m$ items, while the model preserves the computational constraints and symmetries of attention. This allows predictions through principled analysis as well as controlled simulations that directly test those predictions. Our abstraction isolates the *key–query* computation that determines *where* a head attends, separating it from the OV pathway that determines *what* is written—a split made explicit in recent mechanistic analyses of attention heads (Kamath et al., 2025). As a result, our attention budget is the *total key dimension* $D_K = h\, d_k$, where $h$ is the number of heads and $d_k$ the per-head key (and query) width.

**Problem Formulation: Relational Graph Recognition (RGR)** We cast the core task of self-attention as a graph recovery problem. We use "relational graph" to mean a directed graph whose edges encode which items should attend to which. *It is not a knowledge graph in the subject–predicate–object sense*: edges are unlabeled, vertices are individual vocabulary items (not entity slots), and the graph itself is fixed across all examples—only the context varies.

**Task.** Let $G = (V, E)$ be a directed graph on $m = |V|$ items with $m' = |E|$ edges. The vertex set $V$ is the *entire vocabulary* of items the layer could ever encounter (not just the tokens in a single context), and an edge $(u, v) \in E$ means: "*whenever both $u$ and $v$ appear in the same context, $u$ should attend to $v$.*" A single example is a *context*: an ordered subset of distinct vertices $\mathcal{C} = (v_{i_1}, \ldots, v_{i_\ell})$ with $1 \le \ell \le m$. Given $G$, we want a parameterization $\Theta(G)$ of an attention layer such that for every context $\mathcal{C}$ and every $v \in \mathcal{C}$, the layer recovers the in-context neighbors $N_G(v; \mathcal{C}) = \{v' \in \mathcal{C} : (v, v') \in E\}$—equivalently, exactly the edges of $G$ whose endpoints both fall inside $\mathcal{C}$. Note that $G$ represents the relationships to be learned and not the attention graph; our models always compare all pairs of items in the context.

**Capacity question.** Fix an input embedding dimension $d_{\mathrm{model}}$. For a graph family $\mathcal{G}_{m,m'} = \{\, G : |V| = m,\ |E| = m' \,\}$, we ask for the minimal total key dimension $D_K = h\, d_k$ such that the self-attention model in Section 3 can realize the RGR mapping for all $G \in \mathcal{G}_{m,m'}$ and all contexts $\mathcal{C}$. We refer to this minimal $D_K$ as the *capacity required* by $\mathcal{G}_{m,m'}$ at embedding dimension $d_{\mathrm{model}}$.

**Why this abstraction.** RGR isolates the key–query channel that determines *where* attention goes, while preserving the permutation symmetries and parameter sharing

of self-attention[1]. It lets us dial graph complexity $(m, m')$ and the budget $D_K$ independently, enabling the results reported below. Also, given our focus on a rationale for multi-head attention, it allows us to perform analysis and experiments on permutation graphs, where parallel attention to multiple items is not necessary, removing that factor in the advantage of multiple heads. Exact mechanics (score computation and aggregation across heads) are specified in Section 3 and Appendix C.

**Summary of Results** Our analysis yields both fundamental limits and constructive proofs of capability for self-attention as a relational reasoner, and our experiments validate these predictions in both the idealized model and more realistic extensions. The main contributions are:

**Formal model and budget.** We cast "where to attend" as *Relational Graph Recognition (RGR)* and analyze two QK variants: the usual *scaled-softmax* and *max-over-heads* aggregation, a surrogate for head-wise competition that preserves a key nonlinearity absent from fully linear attention, but is also tractable enough to enable tighter and more extensive analysis (Sec. 3). In both cases, the complexity measure is the total key dimension $D_K = h\, d_k$.

**Information-theoretic lower bound.** We prove that recovering graphs with $m'$ edges on $m$ items requires a total key dimension of $D_K = \Omega\left(\frac{m'}{d_{\mathrm{model}}} \log \frac{m^2}{m'}\right)$. This bound applies to both standard softmax and our max-over-heads variant, quantifying how capacity must grow with the number of relationships, and showing that a smaller model dimension requires a larger total key dimension (App. D).

**Asymptotically Optimal Constructions.** We provide explicit constructions for RGR. For our max-over-heads model, we achieve $D_K = O((\frac{m'}{d_{\mathrm{model}}} + \Delta) \log m')$, where $\Delta$ is the maximum degree of $G$ (Sec. 4). This is asymptotically optimal for sparse graphs with mild degree imbalance. For softmax attention on permutation graphs, we show $D_K = \Theta(\frac{m}{d_{\mathrm{model}}} (\log m)^3)$ is sufficient (Thm. C.1, App. C), where the extra logarithmic factors account for softmax concentration against distractors. These constructions surface the core principles of self-attention and serve as concrete, testable hypotheses about the mechanisms transformers learn.

**Capacity-Based Rationale for Multi-Head Attention.** We show a multi-head advantage even for permutation graphs (single target per query, and thus no need to attend to multiple targets in parallel - Sec. 5). When $d_{\mathrm{model}} \ll m$, the signals for different relationships are superposed, causing interference which grows with the number of relationships

---

[1]We incorporate positional embeddings below.

assigned to each head. Thus, more smaller heads outperform fewer larger ones. This noise-reduction mechanism appears under both the max-over-heads and softmax model variants, and provides a principled capacity-centric justification for multi-head attention as a method to reduce noise.

**Empirical Validation and Robustness to Model Extensions.** We confirm our theoretical results in controlled single-layer experiments. **Capacity:** we observe sharp phase transitions in performance as $D_K$ increases (Sec. 6, App. B.1,B.3,A.1,A.3). **Scaling:** the minimal budget $D_K^\star$ follows our predicted scaling for the max-over-heads variant, $D_K^\star \approx \frac{m' \log m}{d_{\text{model}}}$, for both permutation graphs and denser regular graphs (Sec. 6). **Head Count:** the optimal number of heads in our max-over-heads variant scales linearly with $m/d_{\text{model}}$, as predicted theoretically (Sec. 6). **Robustness:** these trends hold in progressively more realistic model variants: adding softmax (App. A.1), adding an OV/value channel (App. A.2), and even a **full single-layer Transformer block trained with frozen GPT-2 embeddings** on an induction-style retrieval task only requiring attention to a single target (App. A.3).

Together, these findings give a quantitative and falsifiable picture of how key–query budget enables relational computation in attention: the required total key dimension scales primarily with the number of stored relationships (edges) relative to the embedding dimension, and distributing a fixed budget across multiple heads can be essential even when each query has a single correct target. The alignment between lower bounds, constructive designs, and empirical thresholds persists across denser graphs, softmax normalization, value-routing objectives, and a full Transformer block with realistic frozen embeddings. In addition, our findings are consistent with several widely reported behaviors of attention in LLMs: see Sec. 2.1.

## 2. Related Work

Given the breadth of prior work on attention, we defer an extended survey to Appendix G, covering expressivity, language-theoretic limits, connectivity, memorization, superposition, interpretability, and graph-structured models.

Closest to our focus are works on *memorization capacity* (Vardi et al., 2021; Kim et al., 2023; Kajitsuka & Sato, 2025), including analyses of memorization in attention modules (Mahdavi et al., 2024). While aligned in spirit, the problem formulations are different: memorization typically maps each *context* to a single output token/label, whereas our RGR setting asks for the recovery of in-context neighbors for *every* context from a set of possible tokens. Reductions between the two would require memorization handling a combinatorial number of contexts (polynomial in $m$ for fixed $\ell$, and exponential when $\ell$ scales with $m$), and we

are not aware of efficient reductions that preserve guarantees in either direction. Accordingly, bounds in one setting do not directly imply bounds in the other. Not surprisingly, capacity results for memorization provide different scaling laws than ours. Our abstraction isolates the key–query *addressing* step—"where to attend"—which mechanistic analyses identify as central to head routing (Kamath et al., 2025). In this sense, RGR complements parameter-centric memorization settings that emphasize "what to output": we target the capacity required to *select* the correct neighbors across contexts.

Beyond memorization, prior theory characterizes *what* Transformers can compute with sufficient resources—universality/approximation (Yun et al., 2020a), fine-grained attention-matrix expressivity (Likhosherstov et al., 2021), and structural bottlenecks such as per-head low rank and rank collapse without mixing (Bhojanapalli et al., 2020; Dong et al., 2021). Language-theoretic and composition results map limitations at fixed budgets (Hahn, 2020; Peng et al., 2024); orthogonally, restricting connectivity rather than dimensions shows universality of $O(\ell)$-sparse patterns and principled pruning of dense ones (Yun et al., 2020b; Wang et al., 2022); and algorithmic views analyze in-context procedures (Li et al., 2023b). Mechanistic studies find head specialization and prune-ability (Clark et al., 2019; Michel et al., 2019), while memory-centric views link attention and FFNs to associative/key–value memories (Ramsauer et al., 2021; Geva et al., 2021). In recent blog-style reflections on associate memories, Zhong et al. (Zhong et al., 2025) demonstrate an exponential gap between linear and softmax attention due to interference. This contrasts with our polynomial gap, highlighting significant differences in problem formulation. They also empirically show that multi-head outperforms single-head attention, and even hypothesize (but do not quantify) this to be due to interference. They do not experiment with single-target scenarios, leaving open the potential role of parallel attention.

### 2.1. Our findings vs. observations in trained LLMs

Our findings are consistent with several widely reported behaviors of attention in LLMs. Specifically:

**Why so many heads?** LLM hyperparameter optimization leads to many heads with modest $d_k$ (e.g., GPT-3 175B: 96 heads (Brown et al., 2020)). Instead of explaining this through needing to pay attention to dozens of targets simultaneously, and then somehow making use of all the varied information provided by these targets, our theory predicts that increasing head count keeps per-head dimension small, thereby decreasing interference.

**Similar effects in pre-trained LLMs.** A similar head-count tradeoff and impact of competitive aggregation as we theorize has been observed in GPT-2 training (Zhong et al., 2025): when model width is held fixed, collapsing to fewer heads (down to 1) improves a softmax-free *linear-kernel* attention variant, but *degrades* standard softmax attention.

**Pruning, specialization and redundancy.** Many heads can be pruned with little loss, a small subset is specialized and important, and substantial head redundancy has been found (Voita et al., 2019; Michel et al., 2019; Bian et al., 2021); this matches an *above-capacity* regime where only a fraction of potential relations are heavily used, and is consistent with pressure to increase heads coming from a need to keep per-head dimension small, instead of paying parallel attention to many targets. Redundancy of high-value relations across heads can act as noise-reduction, consistent with our theory of interference.

**Softmax vs. linear attention.** Linear attention variants often underperform softmax (Han et al., 2024); our analysis predicts this: removing competitive gating collapses multi-head into a single $D_K$-wide linear map, increasing interference.

**MQA/GQA as a capacity knob.** Sharing KV heads (MQA) trades quality for speed (Shazeer, 2019), while restoring multiple KV groups (GQA) recovers quality (Ainslie et al., 2023) and is used in deployed models (e.g., Llama 3 (Meta AI, 2024)); this is consistent with reducing/restoring *effective* key capacity.

Taken together, these observations span *architecture choices* (many small heads; MQA/GQA), *ablation behavior* (pruning/specialization/head merging), *representational structure* (redundancy), and *algorithmic variants* (softmax vs. linear). This provides empirical support that the abstractions studied in RGR capture constraints that trained LLMs already operate under.

## 3. Modeling the Self-Attention Mechanism

We model the *key–query (QK)* computation of a single self-attention layer for RGR, retaining permutation symmetry and parameter sharing while omitting the OV pathway (justified below). The input is an ordered context of distinct vertices of length $\ell \leq m$, $\mathcal{C} = (v_{i_1}, \ldots, v_{i_\ell})$, one vertex per attention unit. Each $v \in V$ is described by a unique embedding $\mathbf{x}_v \in \mathbb{R}^{d_{\text{model}}}$. Positional information is not explicitly modeled; if needed, positions can be incorporated by treating (token, position) as distinct vertices.

We here describe the *max-over-heads* variant of our model; its simplicity lends itself well to more extensive and precise analysis. In App. C we describe and analyze the more

standard version, in which each head applies a softmax over items in $\mathcal{C}$ and the resulting per-head probabilities are summed across heads. In both variants, the benefit of multiple heads arises from noise reduction.

**Single head.** Each attention unit with one head uses the same shared projection matrices $W_Q, W_K \in \mathbb{R}^{d_{\text{model}} \times d_k}$. For each $v_{i_p} \in \mathcal{C}$, $\mathbf{q}_{i_p} = \mathbf{x}_{i_p} W_Q, \mathbf{k}_{i_p} = \mathbf{x}_{i_p} W_K$. The unnormalized score from source $v_{i_p}$ to target $v_{i_q}$ is $S_{pq} = \mathbf{q}_{i_p} \cdot \mathbf{k}_{i_q}^\top$. We declare an edge $(v_{i_p}, v_{i_q})$ present iff $S_{pq} > \tau$ for a global threshold $\tau$. Only pairs inside $\mathcal{C}$ are tested.

**Multi-head, max-over-heads version.** With $h$ heads, each head $k$ has $(W_Q^{(k)}, W_K^{(k)}) \in \mathbb{R}^{d_{\text{model}} \times d_k}$ and produces $S_{pq}^{(k)}$. We aggregate by $S_{pq}^{\max} = \max_{k \in \{1,\ldots,h\}} S_{pq}^{(k)}$, and decide $(v_{i_p}, v_{i_q}) \in E \Leftrightarrow S_{pq}^{\max} > \tau$.

**Remark: why max-over-heads (and why this is not a linear model).** The $\max$ aggregation is a deliberate minimal nonlinearity that preserves the core "competitive routing" behavior of attention while making analysis tractable. It prevents non-target heads from contributing additively to the decision for a given relation, which is a crucial function of softmax as well. If one aggregates scores *fully linearly*, then multi-heads are equivalent to having a single head (by concatenating the per-head projection matrices (Cordonnier et al., 2020a)).[2] Thus, the multi-head advantage requires a nonlinearity to suppress cross-head interference. "Max-over-heads" is the cleanest theoretical abstraction of this suppression, and we also replicate key phenomena with standard scaled-softmax (App. C, App. A.1), to show our conclusions are not an artifact of this simpler model.

**Algorithmic objective and budget.** A *construction for RGR* maps a graph $G = (V, E)$ to weights $\{(W_Q^{(k)}, W_K^{(k)})\}_{k=1}^h$ and a threshold $\tau$ that realize the correct edge decisions for *all* contexts $\mathcal{C}$, regardless of length $\ell$. We measure complexity by the *total key dimension* $D_K = h \, d_k$, since both compute and parameter footprint for the QK channel scale with the width of the concatenated projections (see Appendix F).[3] Our goal is to minimize $D_K$ over a graph family $\mathcal{G}_{m,m'}$ for a given $d_{\text{model}}$.

**Analyzing the QK channel in isolation.** Since RGR asks *where* a source should connect, the key–query computation is the gating step: a correct edge can dominate only if the QK channel already separates the true neighbor from in-context distractors. The OV pathway then acts down-

---

[2]Viewed through an associative-memory lens, purely linear aggregation accumulates cross-term interference, yielding retrieval noise that scales with the number of stored associations relative to key dimension (Zhong et al., 2025).

[3]All statements remain (up to a factor of two) with $D_Q + D_K$.

stream of this routing decision—reweighting and propagating what QK has selected—so OV can amplify signal but cannot reliably fix systematic mis-routing.[4] This motivates omitting the OV channel in our model and treating $D_K$ as the relevant budget for capacity. This separation between QK ("where") and OV ("what") is also supported by recent mechanistic analyses of attention heads (Kamath et al., 2025; Franco & Crovella, 2025). Finally, our experiments show that the conclusions from this abstraction are robust: when we add back a value channel and when we move to a full Transformer block with realistic frozen embeddings, the same core phenomena predicted by the QK analysis continue to govern performance (App. A.2, App. A.3).

## 4. Explicit Constructions for RGR

We give QK constructions in the max-over-heads model of Section 3, yielding upper bounds on the key budget $D_K = h\,d_k$ to solve RGR. This provides a concrete measure of the self-attention mechanism's capacity for this task. We first sketch the one-hot embedding, permutation graph case (details in App. E), then present our main construction for compressive embeddings, which achieves $D_K = \Theta\!\big(\frac{m \log m}{d_{\text{model}}}\big)$ for permutation graphs. In App. E, we show how to generalize these results to arbitrary graphs and general embeddings in the max-over-heads model, and in App. C to permutation graphs in the softmax model. Throughout, $i$ indexes the *source* and $j$ the *target*; keys are tied to targets and queries are tied to sources.

**Construction I: one-hot permutation graphs.** Let $G$ be a permutation graph on $m$ items with edges $(i, \pi(i))$, and let $\mathbf{x}_i = \mathbf{e}_i \in \mathbb{R}^m$ (so $d_{\text{model}} = m$). Draw signatures $W_{\text{sig}} \in \{\pm 1\}^{m \times d_k}$ with i.i.d. Rademacher entries and let $\mathbf{w}_j$ be the $j$-th row. Use one head ($h = 1$) and set $\mathbf{k}_j = \mathbf{w}_j$ and $\mathbf{q}_i = \mathbf{w}_{\pi(i)}$. Then $S_{i,\pi(i)} = \langle \mathbf{w}_{\pi(i)}, \mathbf{w}_{\pi(i)} \rangle = d_k$, while for $j \neq \pi(i)$, Bernstein's inequality shows $S_{ij} = \langle \mathbf{w}_{\pi(i)}, \mathbf{w}_j \rangle = O(\sqrt{d_k})$ w.h.p. Hence choosing $d_k = \Theta(\log m)$ and threshold $\tau = d_k/2$ yields simultaneous separation over all pairs $(i,j)$ by a union bound, so a single head recovers all edges w.h.p. Details are in App. E. The following shows this implies correctness over all contexts:

**Monotonicity under context restriction.** If $S_{i,\pi(i)}^{\max} > \tau$ and $S_{ij}^{\max} < \tau$ for all $j \neq \pi(i)$ over the full vertex set $V$, then the same inequalities hold for any context $\mathcal{C} \subseteq V$ since restricting from $V$ to $\mathcal{C}$ only removes distractor targets.

**Construction II: Permutations Under Compressive Embeddings** We now extend the permutation case to the compressive regime $d_{\text{model}} \ll m$ under a *Gaussian unit-norm* embedding:[5] Each item $v_i$ is embedded as a fixed vector $\mathbf{x}_i \in \mathbb{R}^{d_{\text{model}}}$ drawn i.i.d. as $\tilde{\mathbf{x}}_i \sim \mathcal{N}(0, I/d_{\text{model}})$ and then $L_2$-normalized, i.e., $\mathbf{x}_i = \tilde{\mathbf{x}}_i / \|\tilde{\mathbf{x}}_i\|_2$. Write $X \in \mathbb{R}^{m \times d_{\text{model}}}$ for the matrix with $i$-th row $\mathbf{x}_i^\top$. Given such an embedding and permutation $\pi$, our goal is to construct attention parameters that recognize $G$.

**Multi-Head Algorithmic Construction.** The fundamental challenge with embeddings is that the input $\mathbf{x}_i$ is a superposed representation of the node's identity. Our construction first approximately inverts the embedding process, projecting the $d_{\text{model}}$-dimensional vector $\mathbf{x}_i$ back into the $m$-dimensional one-hot space using the transpose of the embedding matrix. We then apply the logic from the one-hot case. However, doing this with a single head yields too much noise due to the inversion being only approximate. We mitigate this noise by using multiple attention heads, where each is responsible for recognizing the outgoing edges from a disjoint subset of sources. This results in smaller individual heads, and thus less noise. Fixed-point sources are placed in smaller blocks of size $\lfloor d_{\text{model}}/4 \rfloor$, which keeps $h = O(m/d_{\text{model}})$ while controlling the diagonal score; details and the general case are in App. E. The proof of the following theorem appears in Appendix E, where we also show how to extend these results to more general embeddings and graphs.

---

**Algorithm 1** Permutation Graphs with $d_{\text{model}} < m$

1: **Input:** Permutation graph $G = (V, E)$ with $\pi : V \to V$; embedding matrix $X \in \mathbb{R}^{m \times d_{\text{model}}}$.

2: **Parameters:** $h \leq 5m/d_{\text{model}} + 2$; $d_k = C \log m$ for sufficiently large constant $C$.

3: **Set Threshold:** $\tau = \frac{1}{2} d_k$.

4: **Partition sources and targets.** Let $F = \{i : \pi(i) = i\}$. Partition the non-fixed sources $V \setminus F$ into blocks of size $\leq d_{\text{model}}$ and the fixed points $F$ into blocks of size $\leq \lfloor d_{\text{model}}/4 \rfloor$. For each block $V_k$ set $T_k := \pi(V_k)$ (so $T_k = V_k$ on a fixed-point block); the $\{T_k\}$ partition $V$. Head $k$ handles sources in $V_k$ and targets in $T_k$.

5: **Random signatures:** Draw $W_{\text{sig}} \in \{\pm 1\}^{m \times d_k}$ with i.i.d. Rademacher entries; let $\mathbf{w}_j$ be its $j$-th row.

6: **Ideal one-hot-space templates (for each head $k$):**

7:     **Query Matrix:** $W'_{Q,(k)} \in \mathbb{R}^{m \times d_k}$ with row $i$ equal to $\mathbf{w}_{\pi(i)}$ if $i \in V_k$, and $\mathbf{0}$ otherwise.

8:     **Key Matrix:** $W'_{K,(k)} \in \mathbb{R}^{m \times d_k}$ with row $j$ equal to $\mathbf{w}_j$ if $j \in T_k$, and $\mathbf{0}$ otherwise.

9: **Approximate de-embedding:** $W_Q^{(k)} = X^\top W'_{Q,(k)}$, $W_K^{(k)} = X^\top W'_{K,(k)}$.

---

**Theorem 4.1** (Multi-head recognition under Gaussian

---

[4]Our model is scoped to a single self-attention layer; multi-layer iterative routing is outside our abstraction.

[5]Spherical codes give clean separation thresholds; see App. E for why this geometry is natural and its practical relevance.

unit-norm embeddings, max-over-heads). *Assume Gaussian unit-norm embeddings with $d_{\text{model}} \geq c_0 \log m$ for a sufficiently large absolute constant $c_0$. Let $h \leq 5m/d_{\text{model}} + 2 = O(m/d_{\text{model}})$ heads, per-head dimension $d_k = C \log m$ for a sufficiently large absolute constant $C$, and threshold $\tau = \frac{1}{2}d_k$. Construct $\{(W_Q^{(k)}, W_K^{(k)})\}_{k=1}^h$ as in Algorithm 1 and let $k(i)$ denote the unique head index such that $i \in V_{k(i)}$. Then with probability at least $1 - m^{-3}$ over the draw of $(X, W_{\text{sig}})$, simultaneously for all $i \in V$:*

$$S_{i,\pi(i)}^{(k(i))} > \tau \qquad and \qquad \max_{k \in [h]} \max_{j \neq \pi(i)} S_{ij}^{(k)} < \tau.$$

*Consequently, $\forall j \neq \pi(i)$, $S_{i,\pi(i)}^{\max} > \tau > S_{ij}^{\max}$, so max-over-heads recovers all edges, and the total key budget satisfies $D_K = h\, d_k = \Theta\!\left(\frac{m \log m}{d_{\text{model}}}\right)$.*

**Consequence.** By Theorem 4.1 we have global separation under max-over-heads, and by monotonicity under context restriction the same parameters recognize $E|_{\mathcal{C}}$ for every context $\mathcal{C} \subseteq V$ and every context length. The total key budget matches our lower bound for permutation graphs up to constant factors. The same separation-based approach extends to softmax aggregation (App. C). The de-embedding + signature structure here qualitatively mirrors what mechanistic interpretability finds in trained heads—projection into a subspace where particular relationships produce high QK dot products and others are suppressed (Kamath et al., 2025).

## 5. The power of multiple heads

With no compression (Construction I), a single head suffices: queries and keys can coincide exactly on true edges and be nearly orthogonal otherwise, yielding true-edge scores $\Theta(d_k)$ and non-edge scores concentrated near 0. In the compressive setting (Construction II), we first approximately de-embed $\mathbf{u}_i := \mathbf{x}_i X^\top = \mathbf{e}_i + \boldsymbol{\delta}_i$, so each source carries a small *leakage* vector $\boldsymbol{\delta}_i$ that spreads mass across many coordinates. With Rademacher signatures (see §4) the head-$k$ score decomposes into a signal term—$\Theta(d_k)$ for true edges and concentrated near 0 for non-edges—and a noise term controlled by the leakage. The dominant component of this noise, denoted $N_3$ in Appendix E, scales with the *block size* $B := |T_k|$ served by a head. Intuitively, if the block size is too large, there is too much noise, and so multiple heads are required to keep the block size small.

$$N_3(B) \;\asymp\; \frac{B}{d_{\text{model}}} \sqrt{d_k \log m}. \tag{1}$$

To guarantee (w.h.p.) a fixed margin between the true target and all non-targets, it must be that, for constants $c_1, c_2 > 0$,

$$N_3(B) \leq c_1 d_k \implies d_k \geq c_2 \frac{B^2}{d_{\text{model}}^2} \log m. \tag{2}$$

**Single head with compression.** If one head serves all items, then $B = m$ and $D_K = d_k$. Applying (2) implies

$$D_K = \Omega\left(\frac{m^2}{d_{\text{model}}^2} \log m\right).$$

**Multiple heads with compression.** Construction II partitions the items into $\approx \frac{m}{d_{\text{model}}}$ heads with $B = d_{\text{model}}$ per head. Plugging $B = d_{\text{model}}$ into (1) yields $N_3(B) = \Theta(\sqrt{d_k \log m})$. Taking $d_k = c_3 \log m$ with $c_3$ larger than the constant in (1) ensures $N_3 \leq c_1 d_k$ w.h.p., and the total key dimension is

$$D_K \;=\; h\, d_k \;=\; O\!\left(\frac{m}{d_{\text{model}}} \log m\right).$$

**Consequence.** The additional $\frac{m}{d_{\text{model}}}$ term for a single head implies that if $m = \omega(d_{\text{model}})$ (the compressive regime), the single head requirement above implies asymptotically larger $D_k$ than the multihead construction. Or, equivalently, for a fixed $D_k$ budget, multiple heads can handle more edges (relationships) than a single head, even in a permutation graph. Multiple heads do not boost per-head expressivity; they *localize* de-embedding noise by reducing block size $B$, so that each head aggregates leakage over fewer coordinates, bringing the noise to a manageable level. Note that this is not a lower bound for *all* conceivable single-head designs, but it shows that within the de-embedding to signature template we use, a single head cannot perform as well as multiple heads.

## 6. Experiments

We conduct experiments mirroring our theoretical model, testing several predictions. First, we compare the empirical minimum total key dimension, $\hat{D}_K^\star$, to the predicted theoretical scaling law of $\Theta\left(\frac{m \log m}{d_{\text{model}}}\right)$, noting that optimization may fail to find a solution matching the theoretical constructive bound. Second, we test head-count predictions: whether permutation graphs exhibit a multi-head advantage and how the empirically optimal head count tracks the theory. Permutation graphs ensure each query has at most one correct in-context target, so improvements from multiple heads cannot be attributed to attending to multiple true neighbors.

**Experimental implementation** We instantiate our idealized attention layer with two learned projections $W_Q, W_K \in \mathbb{R}^{d_{\text{model}} \times D_K}$ partitioned into $h$ heads ($d_k = D_K/h$). For a context matrix $X_{\mathcal{C}}$, head $k$ computes $S^{(k)} = Q^{(k)}(K^{(k)})^\top$ with $Q^{(k)} = X_{\mathcal{C}} W_Q^{(k)}$ and $K^{(k)} = X_{\mathcal{C}} W_K^{(k)}$; scores are combined by an elementwise max $S_{\max} = \max_k S^{(k)}$, and we predict an edge $(p \to q)$ iff $S_{\max}(p, q) > \tau$ for a single learned global threshold $\tau$.

There is no $1/\sqrt{d_k}$ scaling, softmax, or value pathway, so capacity is purely key–query. Tasks are permutation graphs on $m$ items (one out/in-edge per node). Node embeddings $x_i \sim \mathcal{N}(0, I/d_{\text{model}})$ are $L_2$-normalized and frozen, making $D_K$ the sole capacity knob. Contexts of length $\ell$ (default $\ell=16$) are sampled with target-in-context rate $\rho=0.5$.

We train $W_Q, W_K, \tau$ with AdamW (lr $10^{-3}$, weight decay 0) using a weighted logistic loss over all ordered pairs within a context (positive weight $\ell-1$; logit sharpness $\alpha=10$), one context per step. For each run, a single permutation $\pi$ and embedding matrix are fixed by seed; training contexts are drawn on-the-fly, with 500 validation and 2,000 held-out test contexts from the same $(\ell, \rho)$ distribution. Early stopping checks validation micro-F1 every 500 steps and halts after five consecutive checks above 0.995. We report micro-F1 on the fixed test set with the single learned $\tau$; the "minimum $D_K$" is the smallest $D_K$ achieving mean test micro-F1 $\geq 0.99$ for at least one head count $h$. Full details appear in App. B.1.

### 6.1. Results and comparison to theory

We probe capacity on permutation graphs with $m \in \{64, 128, 256, 512\}$ and $d_{\text{model}} \in \{16, 32, 64\}$. For each $(m, d_{\text{model}})$ we sweep head counts $h \in \{1, 2, 4, 8, 16, 32, 64\}$ and several total key sizes $D_K = h\, d_k$ (multiple $D_K$ per $h$). Each configuration is trained from 10 seeds with the protocol described above. We evaluate average test micro–F1 on a fixed held-out set and define the empirical threshold $D_K^\star = \min\{ D_K : \exists h \text{ s.t. mean test micro-F1} \geq 0.99 \}$. We denote by $h^\star$ a head count that attains $D_K^\star$. Full grids and per-config step limits are in App. B.2.

To isolate sequence-length effects at fixed embedding compression, we also traverse the diagonal $r \overset{\text{def}}{=} m/d_{\text{model}}=8$ with $(m, d_{\text{model}}) \in \{(128, 16), (256, 32), (512, 64), (1024, 128), (2048, 256), (4096, 512)\}$, using 3 seeds for the largest points and increased budgets (App. B.2). Finally, to probe extreme compression we include a second $r=32$ point $(1024, 32)$ (in addition to $(512, 16)$).

**Qualitative phenomena.** We observe: (i) a *sharp* F1 transition in a narrow $D_K$ window (capacity threshold) across all $(m, d_{\text{model}}, h)$ (Fig. 1 here and Fig. 10 in App. B.2); (ii) a pronounced *multi-head advantage* for many $(m, d_{\text{model}})$, even though each query has a single target—splitting a fixed $D_K$ across more heads reduces interference from superposition (Fig. 2); and (iii) the *optimal* head count increases with compression $\lambda = m/d_{\text{model}}$ (Fig. 2), while per-head width at the threshold is modest.

**Empirical thresholds on the base grid.** The minimum $D_K^\star$ grows rapidly with $m$ and falls with $d_{\text{model}}$; exact

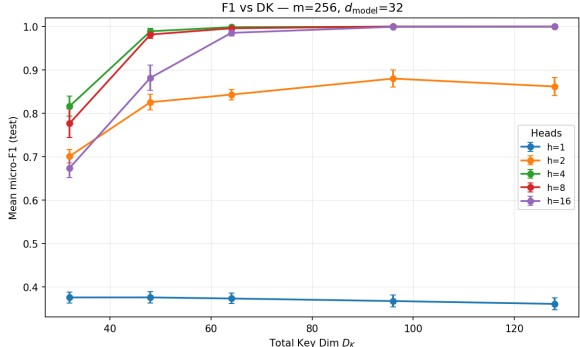

*Figure 1.* **Example F1–$D_K$ curves.** Lines are a fixed number of heads for $m = 256, d_{\text{model}} = 32$. A single head fails to separate the signal from superposition noise, and performs much worse than multiple heads. Error bars are 95% CIs over 10 runs.

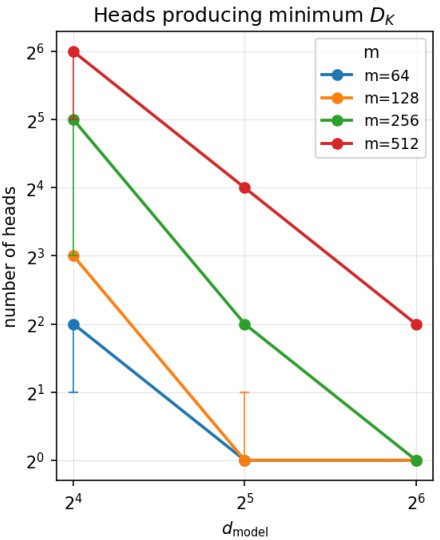

*Figure 2.* **$h^\star$ minimizing $D_K$.** More heads are needed as $m$ grows or $d_{\text{model}}$ shrinks. See App. B.2 for error bar description.

values appear in Figure 9 (App. B.2). A single head often fails to reach 0.99 F1 within the scanned $D_K$ (e.g., $(m, d_{\text{model}}) \in \{(512, 64), (256, 32)\}$, Fig. 10), whereas several small heads pass at much smaller $D_K$.

**Scaling laws** Plotting $D_K^\star$ against $\frac{m \log m}{d_{\text{model}}}$ yields a tight linear relation, $D_K^\star \approx \mathbf{1.19}\, \frac{m \log m}{d_{\text{model}}}$ with $R^2 = \mathbf{0.944}$ (Fig. 3). We see small deviations when $d_{\text{model}}$ is too small relative to $\log m$; this is consistent with our theoretical results. Excluding the three $(d_{\text{model}}=16, m>64)$ points (above the line in Fig 3)—which violate the precondition $d_{\text{model}} \gtrsim c_0 \log m$ used by our constructions—gives slope 0.966 with $R^2 = \mathbf{0.992}$. Thus, the empirical capacity closely matches the theoretical $\Theta\!\left(\frac{m \log m}{d_{\text{model}}}\right)$ rate. The head count that attains $D_K^\star$ scales approximately linearly with compression, $h^\star \approx \mathbf{1.65}\, \frac{m}{d_{\text{model}}} - \mathbf{6.64}$ ($R^2 = \mathbf{0.824}$,

Fig. 11 in the Appendix). At $D_K^\star$, per-head widths are small: $d_k^\star \in [5, 24]$ on the base grid (median 11), indicating gains come from *adding heads* rather than making each head wide (Table 1; App. B.2).

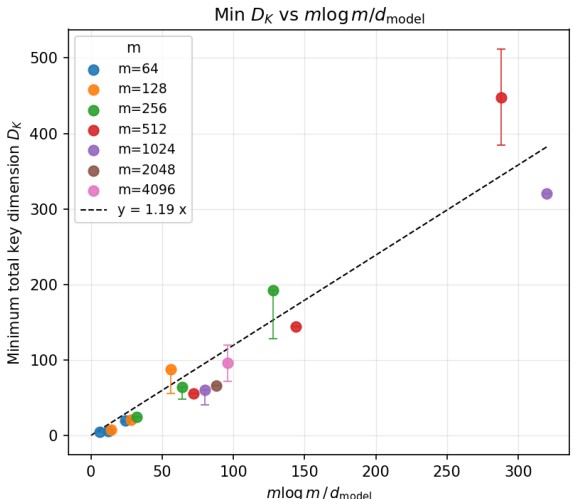

*Figure 3.* **Empirical Validation of Capacity Law.** The minimum key dimension $D_K^\star$ tracks the theoretical prediction $\frac{m \log m}{d_{\text{model}}}$ tightly ($R^2 = 0.944$). See App. B.2 for error bar description.

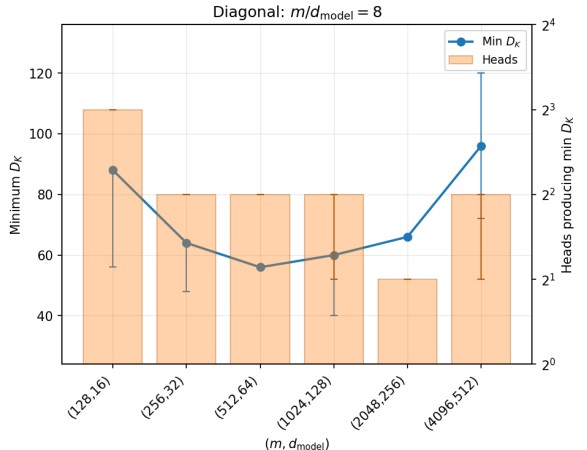

*Figure 4.* **Empirical validation of optimal headcount.** Fixed compression diagonal with $\lambda = 8$. Line (left axis): minimum $D_K^\star$ achieving F1$\geq .99$. Bars (right axis): $h$ achieving that minimum. See App. B.2 for error bar description.

**Fixed-compression diagonal** ($\lambda$=8). Holding $\lambda$ constant collapses the prediction to $D_K^\star \propto \lambda \log m$, so the dependence on $m$ should be logarithmic. Along $(128, 16) \to (4096, 512)$ we observe roughly this behavior from $m \geq 512$ onward (Fig. 4): $D_K^\star$ grows slowly while $m$ grows exponentially, matching the $\log m$ factor. The first two points are slightly conservative (smaller $d_{\text{model}}$) and align with the same $d_{\text{model}} \gtrsim \log m$ finite-size effect. We also expect the

optimal head count to be proportional to $\lambda$; the observed results align well with this expectation.

**Takeaways.** Empirical thresholds align closely with the $m \log m / d_{model}$ capacity law and expose a clear *multi-head advantage* even for one-target graphs. Discrepancies appear exactly where theory anticipates stronger superposition (small $d_{\text{model}}$ and very large $m$). Overall, allocating key–query budget across *more heads with modest width* is the efficient path to capacity in compressed embeddings.

### 6.2. Extensions and robustness

The model studied so far isolates the key–query channel with a max-over-heads aggregator, frozen near-orthogonal embeddings, and permutation graphs. We now stress-test each of these assumptions. Softmax aggregation and an added value channel preserve all qualitative findings; full details are in App. C, A.1, and A.2. A denser graph family and a full Transformer block with frozen GPT-2 embeddings bring our predictions closer to practice and we present them in more detail.

**Softmax and a value channel.** On permutation graphs, replacing the max aggregator with scaled-softmax with sum-over-heads preserves the sharp F1 transition and the multi-head advantage across $(m, d_{\text{model}})$ settings (App. A.1, Fig. 7); the empirical threshold shifts right, consistent with the extra logarithmic factors in our softmax construction (Thm. C.1, App. C). Formally, the single-head total key dimension is asymptotically larger than the multi-head budget by a factor of $\Omega(m/(d_{\text{model}}(\log m)^2))$ in the compressive regime (Prop. C.5), so the noise-suppression advantage of partitioning is not specific to the max surrogate. Adding a fixed random value pathway and training on message retrieval (MSE against the neighbor's value vector) rather than thresholded edge classification yields the same picture: in the compressed regime, a single head fails to drive the retrieval error down while multi-head models succeed (App. A.2). The capacity bottleneck is neither an artifact of the max surrogate nor the binary edge objective.

**Beyond permutation graphs.** We re-run the protocol on $r$-regular graphs ($r \in \{1, 2, 4, 8, 16\}$, $m$=128, $d_{\text{model}}$=32). All three qualitative phenomena persist (App. B.3): the F1–$D_K$ transition stays sharp and shifts right with $r$; multi-head models clear the threshold at smaller $D_K$ than single-head models; and the optimal head count rises with density, tracking $m'/d_{\text{model}}$ up to a small constant (App. Fig. 13). Quantitatively, plotting $D_K^\star$ against the edge-normalized variable $x = (rm/d_{\text{model}}) \log_2 m$ collapses all five densities onto a single line $D_K^\star \approx 0.46 \, x$ (Fig. 5). The binding constraint is therefore the number of *edges* $m'$, not the vertex count $m$. Theoretically, a general construction

(Algorithm 5, Thm. E.10) achieves $D_K = O\left(\frac{m' \log m}{d_{\text{model}}} + \Delta \log m\right)$ for any directed graph with maximum degree $\Delta$. For graphs with bounded skew $\Delta/d_{\text{avg}} \leq m/d_{\text{model}}$—which includes every $r$-regular graph—this matches the information-theoretic lower bound up to constants whenever $m' = O(m^{2-\epsilon})$.

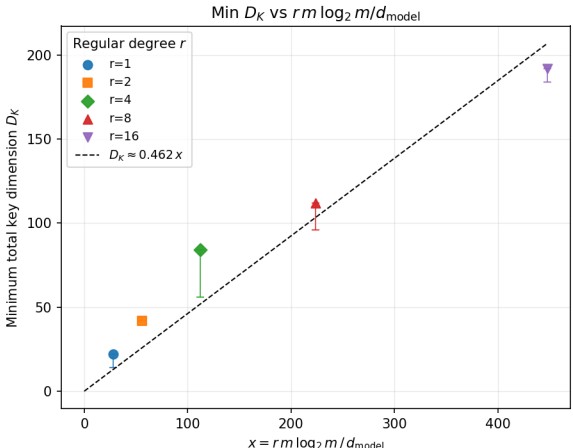

*Figure 5.* **Capacity scales with edges, not vertices.** Empirical $D_K^\star$ for $r$-regular graphs ($r \in \{1, 2, 4, 8, 16\}$, $m=128$, $d_{\text{model}}=32$) versus the edge-normalized variable $x = (rm/d_{\text{model}}) \log_2 m$. All densities collapse onto a single line $D_K^\star \approx 0.46\, x$.

**A full single-layer Transformer block with frozen GPT-2 embeddings.** We finally validate the capacity-based account in our most realistic setting: a full pre-LN Transformer block (MHA + MLP + residuals) trained end-to-end with frozen GPT-2-small token embeddings ($d_{\text{model}}=768$, $m=6144$ items drawn from WikiText-103 unigram-frequency tokens). We design an induction-style retrieval task: each example places a query token $q$, a latent target $s = \pi(q)$ under a fixed permutation $\pi$, and an in-context matching $M$ over a length-$\ell$ context ($\ell=16$); the label is $M(s)$. Crucially, every pair $(a_i, b_i)$ in the matching is encoded into a *single superposed slot* $x_i = E[a_i] + E[b_i]$, so all information needed by the query is concentrated at one position—attending to multiple positions is never useful, and the conventional "parallel attention" rationale for multi-head attention cannot apply. We sweep $D_K \in \{64, 128, 256, 512\}$, partition each $D_K$ across $h \in \{1, 2, 4, 8\}$ heads ($d_k = D_K/h$, $d_v = d_{\text{model}}/h$), keep MLP width $d_{\text{ff}} = 16$ so the bottleneck remains in the QK channel, and train end-to-end with 7 seeds per configuration. Full setup is in App. A.3.

All three core phenomena recur (Fig. 6). (i) For multi-head models ($h \geq 2$), accuracy increases smoothly with $D_K$ and approaches 0.98 by $D_K=512$. (ii) At every fixed $D_K$, splitting the budget across multiple heads strictly improves accuracy; the best partitions use $h \in \{4, 8\}$. (iii)

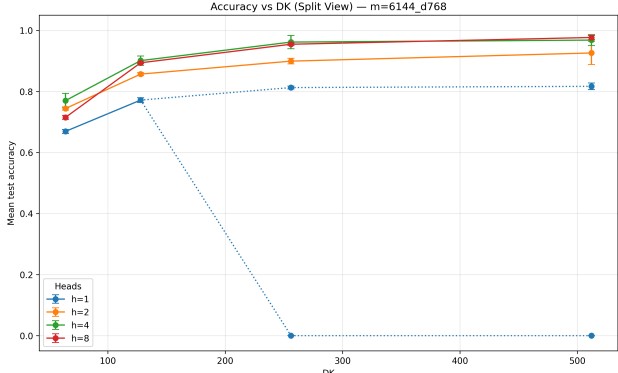

*Figure 6.* **Multi-head advantage in a full Transformer block with frozen GPT-2 embeddings.** Mean test accuracy vs. $D_K$ for head partitions $(h, d_k)$ at fixed $D_K$, with $m=6144$, $d_{\text{model}}=768$, $\ell=16$. Error bars are 95% CIs over 7 seeds. For $h=1$ at $D_K \geq 256$ training is bimodal: a fraction of seeds collapse to the NULL baseline (lower dotted, near 0), while the remaining runs saturate near 0.81 (upper dotted) and do not improve with larger $D_K$.

Single-head models exhibit a sharp failure mode: for $D_K \geq 256$ training becomes bimodal, with a substantial fraction of seeds collapsing to the NULL baseline (accuracy $\approx 0$) while the rest saturate near 0.81 and do not improve with larger $D_K$. Because these comparisons are at fixed $D_K$ and all information a query needs sits in one context slot, the multi-head advantage here cannot be attributed to parameter count or attending to multiple positions; it is the capacity-based effect predicted by our theory, surviving non-orthogonal real-text embeddings, MLP and residual pathways, and end-to-end training.

## 7. Limitations

Our $D_K$ guarantees are not uniformly tight: for max-over-heads we asymptotically match the information-theoretic lower bound for permutation graphs and broad sparse regimes, but gaps remain for very dense, highly degree-imbalanced graphs, and for softmax our sufficient condition incurs extra logarithmic factors of unclear necessity. Since a central motivation is to show the multi-head advantage does *not* rely on "attending to multiple targets at once," permutation graphs are the most diagnostic setting, where our bounds and evidence are strongest. Our analysis assumes a controlled embedding geometry to quantify interference; we partially bridge this with our frozen-GPT-2 Transformer block, and expect realistic embeddings to create *more* (not less) interference via concentration. Learned embeddings could improve constants, but our lower bounds (which still apply to them) and near-matching upper bounds limit how much they can help. Empirically, we focus on controlled, largely synthetic single-layer tasks, so we do not model multi-layer iterative routing or end-to-end training dynamics.

## Impact Statement

This work is primarily theoretical and aims to improve our understanding of how attention mechanisms allocate limited representational capacity. Better understanding of attention capacity may enable more compute- and parameter-efficient Transformer designs, potentially reducing the energy and cost of training and deployment. As with many advances in model efficiency and capability, these insights could indirectly support the development of more capable systems; we encourage pairing architectural improvements with appropriate evaluation, robustness, and safety practices. We do not foresee unique, domain-specific negative impacts beyond these general considerations.

## Acknowledgements

We thank Jonathan Kelner for pointing out that the constructive upper bounds must treat the fixed points of the permutation—sources whose target is themselves—separately; this observation motivated the smaller-block handling of fixed-point sources used throughout our constructions.

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

# A. Robustness to Model Extensions

We next report on experiments run in progressively more realistic model variants, and see that even with these extensions, the same core phenomena predicted by the simpler analysis continue to govern performance. We study the effects of the softmax aggregation rule, adding a value channel, and a full Transformer block with realistic frozen embeddings.

## A.1. Experiments incorporating softmax

We also conducted experiments with the softmax version of the idealized model. These experiments mirror our experiments in the base model, with only the following changes:

- Scores are subject to a softmax along the context dimension.

- Scores are scaled by $\sqrt{d_k}$.

- Scores are aggregated over heads via a sum (instead of max).

- Training was allowed to run for up to 80,000 steps.

We depict several examples of the resulting F1-$D_K$ curves in Figure 7. As in the model with max over heads (and no softmax), we here see a distinct multi-head advantage when in the compressed embedding range of $m \gg d_{\text{model}}$. We also see that the required $D_K$ to reach an F1 of 0.99 is shifted to the right from the experiments run in our core model - consistent with the additional logarithmic factors in our constructions.

## A.2. Extension to a value channel

We next study the empirical impact of adding a value channel to our model. All aspects of the data generation and optimization protocol remain in our core model (permutation graphs on $m$ nodes, frozen normalized embeddings $x_i \in \mathbb{R}^{d_{\text{model}}}$, contexts of length $\ell$ with target-in-context rate $\rho$, and AdamW with the same learning rate and regularization), except that we equip the layer with a value pathway and change the learning objective from edge classification to message retrieval.

Concretely, the key–query path is unchanged: we retain the same learned projections $W_Q, W_K \in \mathbb{R}^{d_{\text{model}} \times D_K}$, partitioned into $h$ heads with per-head width $d_k = D_K/h$, and we continue to aggregate scores by an elementwise max over heads $S_{\text{max}} = \max_k Q^{(k)}(K^{(k)})^\top$ without $1/\sqrt{d_k}$ scaling or softmax. The only architectural addition is a *random value map* $W_V \in \mathbb{R}^{d_{\text{model}} \times d_{\text{msg}}}$, drawn once at initialization and then frozen. Each node $i$ carries a "message" $y_i = x_i W_V \in \mathbb{R}^{d_{\text{msg}}}$ that lies in the span of the original embeddings, matching the theoretical assumption of a fixed random value channel.

Given a context $\mathcal{C}$ with embedding matrix $X_{\mathcal{C}}$, we reuse the same attention scores $S_{\text{max}} \in \mathbb{R}^{\ell \times \ell}$ to mix value messages. Let $V_{\mathcal{C}} = X_{\mathcal{C}} W_V$ collect the in-context messages. For each position $i$ whose outgoing permutation neighbor $\pi(i)$ also appears in the context, we form a predicted neighbor message

$$\hat{y}_i = \left(S_{\text{max}} V_{\mathcal{C}}\right)_i \tag{3}$$

using the raw (un-normalized) scores in $S_{\text{max}}$ as mixing weights, and we supervise it toward the true neighbor message $y_{\pi(i)} = x_{\pi(i)} W_V$. Training minimizes the mean squared error between $\hat{y}_i$ and $y_{\pi(i)}$ over all such "valid" positions in the batch, i.e.,

$$\mathcal{L}_{\text{MSE}} = \frac{1}{|\mathcal{I}_{\text{valid}}|} \sum_{i \in \mathcal{I}_{\text{valid}}} \left\|\hat{y}_i - y_{\pi(i)}\right\|_2^2, \tag{4}$$

where $\mathcal{I}_{\text{valid}}$ is the set of indices whose permutation neighbor lies in the same context. Early stopping now operates on validation MSE for this message-retrieval task (with the same check interval and patience as before), and we report test-set MSE as the primary metric. Micro-F1 and score margins induced by the learned $W_Q, W_K$ no longer play any role in optimization or stopping.

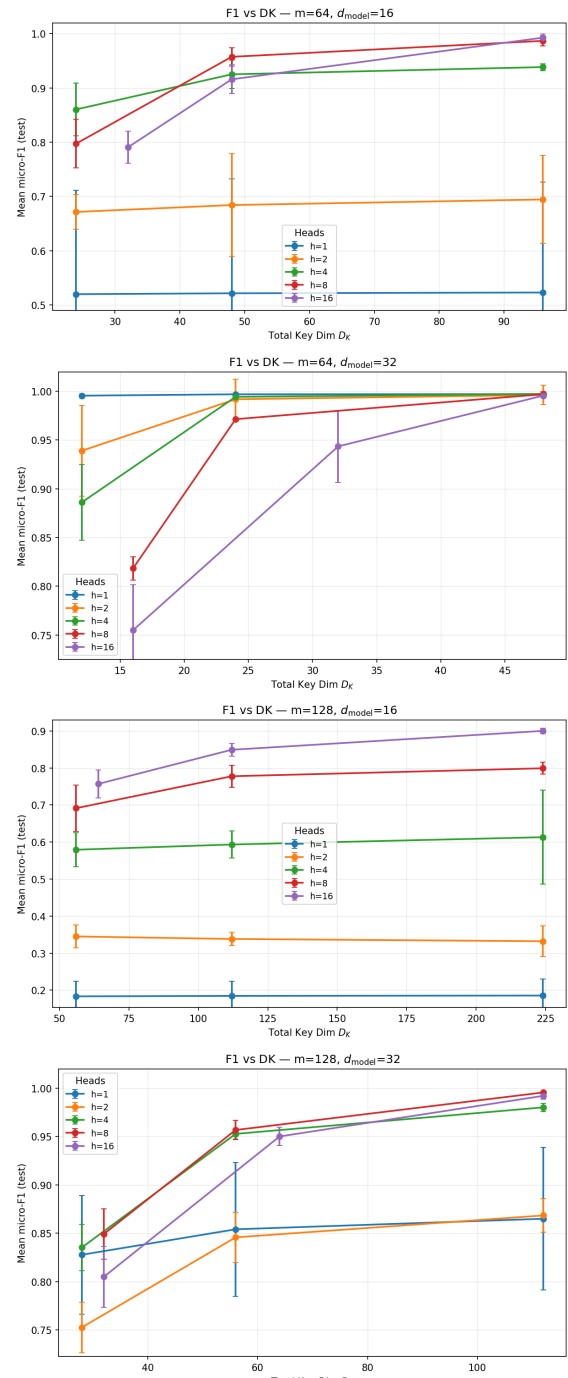

*Figure 7.* **Example F1–$D_K$ curves with softmax incorporated.** Each panel fixes $(m, d_{model})$ and sweeps heads $h$ and $D_K = h\, d_k$; markers show mean test micro–F1 and error bars are 95% CIs over 3 runs. We see a distinct multi-head advantage when in the compressed embedding regime - i.e., all cases except $m = 64$, $d_{model} = 32$.

**Findings (Value Retrieval).** We observe that the multi-head advantage found with edge classification transfers directly to the message-retrieval task. Figure 8 displays the

test MSE as a function of total key dimension $D_K$ for $m = 64$ with embedding dimensions $d_{model} = 16$ and $d_{model} = 32$.

**Multi-head advantage in compressed regimes.** In the compressed regime where $d_{model} \ll m$ (Figure 8, left, $d_{model} = 16$), we observe a significant separation between single-head and multi-head performance. The single-head model ($h = 1$) fails to reduce MSE significantly even as $D_K$ increases, plateauing at a high error rate. In contrast, models with $h \geq 4$ are able to leverage the key budget effectively, driving the MSE down sharply. This confirms that even when the task is soft message retrieval rather than hard binary classification, the geometric bottleneck of identifying the correct neighbor requires the noise suppression impact of multiple heads.

**Behavior in less compressed regimes.** When the embedding dimension is relaxed to $d_{model} = 32$ (Figure 8, right), the necessity for multiple heads diminishes, consistent with our theoretical predictions. Here, a single head ($h = 1$) is competitive, and in some low-$D_K$ settings even outperforms highly fragmented architectures (e.g., $h = 16$, where $d_k$ becomes very small). However, intermediate head counts ($h = 4, 8$) still achieve the lowest ultimate MSE.

**Takeaway.** These results demonstrate that our core findings are not an artifact of the thresholded classification objective. The attention mechanism's ability to route information—specifically, to select the correct value vector to mix—is governed by the same geometric capacity constraints derived for the edge-existence problem.

### A.3. Full transformer layer with frozen GPT-2 embeddings

We next turn to a much more complex and realistic scenario: a full single layer Transformer block used to solve a controlled retrieval problem similar to induction-head pointer indirection. We use a fairly standard structure: pre-LN attention plus MLP block with residuals, and use frozen GPT-2 token embeddings with a tied unembedding.

**Frozen vocabulary and embedding space.** Let $W_{TE} \in \mathbb{R}^{|\mathcal{V}| \times d_{model}}$ be GPT-2-small's token embedding matrix ($d_{model} = 768$). We restrict to an item set of size $m$, using the $m$ tokens with the highest unigram frequency in the WikiText-103 training corpus (estimated from the first 10 million tokens using the GPT-2 tokenizer), excluding the EOS token. This yields a frozen table

$$E \in \mathbb{R}^{(m+1) \times d_{model}},$$

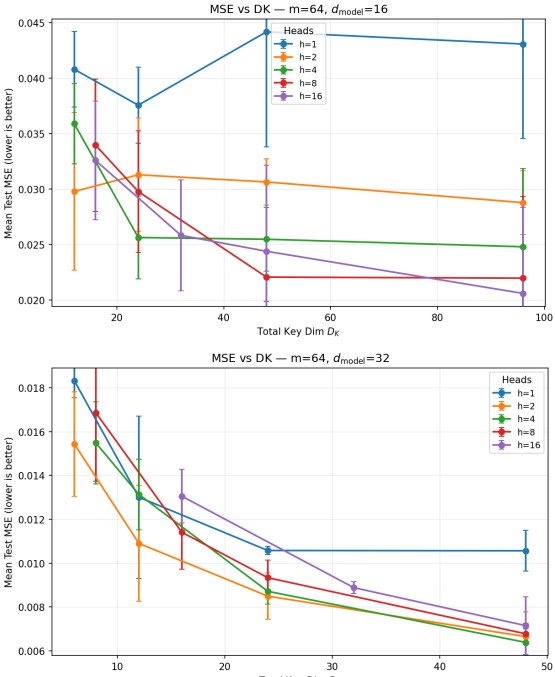

*Figure 8.* **Value retrieval MSE mirrors capacity transitions.** Mean test MSE vs. total key dimension $D_K$ for a value-retrieval task with frozen random value mappings ($m$=64). **Top** ($d_{\text{model}}$=16): In the compressed regime, single-head attention (blue) fails to retrieve the correct message, while multi-head attention succeeds. **Bottom** ($d_{\text{model}}$=32): With less compression pressure, the gap narrows, though multi-head models still attain the lowest final MSE. Error bars are 95% CIs over 3 runs

containing the $m$ item embeddings plus $E[\text{NULL}] = W_{\text{TE}}[\text{EOS}]$, preventing the model from "hiding" structure in embeddings and keeping the bottleneck aligned with QK capacity.

**Task: permutation→matching retrieval (symbolic IOI / induction-style indirection).** For each run (fixed $(m, \ell, D_K, h, \text{seed})$), we sample a random permutation $\pi : [m] \to [m]$ and keep it fixed across train/val/test. Each example samples a query $q$ and finds its *successor* $s = \pi(q)$ (rejecting $s = q$). We then create a sample $S$ of $2\ell - 1$ items, where the items of $S$ are distinct from each other, as well as from $s$ and $q$. The items in $S \cup \{s\}$ are grouped into $\ell$ disjoint pairs, defining a perfect matching $M$ over the $2\ell$ items. The label is $y = M(s)$, where $M(s)$ is the item matched by $M$ with the target of the query $q$.

This implements a two-step pointer computation: a *stored* relation $\pi$ (compiled into parameters) followed by a *prompt-defined* binding $M$ (recovered from the context). It is directly analogous to (i) *induction-head* pointer indirection and (ii) *indirect-object-identification*-style compositional retrieval of "the other entity." However, we use an input encoding that ensures that all the information needed by

a query is contained in a single context location, and thus "paying attention to multiple locations simultaneously" is never required.

**Input encoding (superposition; set-like context).** Each pair $(a_i, b_i)$ is encoded as a single *superposed* slot

$$x_i = E[a_i] + E[b_i], \qquad i = 1, \dots, \ell.$$

The sequence is

$$X = [x_1, \dots, x_\ell, \ E[\text{NULL}], \ E[q]] \in \mathbb{R}^{(\ell+2) \times d_{\text{model}}}.$$

No positional embeddings are used, making the $\ell$ context slots exchangeable; the model is evaluated only at the final (query) position.

**Model: one pre-LN Transformer block with explicit QK budget.** We train a single GPT-style pre-LN block (MHA + MLP + residuals) and read out only the last token. The key experimental lever is the total key/query width

$$D_K = h \, d_k,$$

swept across head partitions $(h, d_k)$ at fixed $D_K$. Values use $d_v = d_{\text{model}}/h$. We train $\{W_Q, W_K, W_V, W_O\}$ (no bias) plus a small 2-layer GELU MLP of fixed width (e.g. $d_{\text{ff}} = 16$) to keep the primary bottleneck in the QK channel. Output uses weight tying:

$$\text{logits} = h_{\text{last}} E^\top \in \mathbb{R}^{m+1}.$$

**Training and capacity measurement.** Data are generated on the fly; the objective is cross-entropy on the final-position prediction. We optimize with AdamW (weight decay 0.01; constant learning rate; mixed precision for throughput) and train for a fixed budget of updates, recording the best validation accuracy achieved and reporting the corresponding held-out test accuracy.[6] For each $(m, \ell, D_K, h, \text{seed})$ we train from scratch and report held-out accuracy. We compare head partitions $(h, d_k)$ at fixed $D_K$ to determine if there is a multi-head advantage.

**Results: multi-head attention remains strongly beneficial in a single layer Transformer and frozen GPT-2 embeddings.** We evaluate the setting $m = 6144$, $\ell = 16$, $d_{\text{model}} = 768$ over $D_K \in \{64, 128, 256, 512\}$ and head counts $h \in \{1, 2, 4, 8\}$ (so $d_k = D_K/h$), using 7 random seeds per configuration, and a maximum of 100,000 training steps per configuration/seed. Figure 6 summarizes the results.

Three qualitative phenomena stand out.

---

[6]In this task the label is never NULL, so the trivial baseline that always predicts NULL has test accuracy 0.

**(1) Increasing $D_K$ improves performance, but the attainable accuracy depends heavily on how $D_K$ is partitioned into heads.** For multi-head models ($h \geq 2$), accuracy increases smoothly with $D_K$ and begins to saturate by $D_K \approx 512$. Thus, once $D_K$ is moderately large, the model can approach near-perfect two-step retrieval—but only for sufficiently multi-headed partitions.

**(2) A pronounced multi-head advantage at fixed $D_K$, despite the task never requiring attention to multiple locations.** At every $D_K$, allocating the same total key budget across multiple heads yields substantially higher test accuracy. Because these comparisons hold at fixed $D_K$ (and thus fixed $W_Q, W_K$ parameter shapes), they isolate a structural advantage of multi-head computation rather than a parameter-count effect. Crucially, this advantage arises even though each query's information is contained in a *single* context slot and there is never a need to attend to multiple slots simultaneously.

**(3) Single-head models exhibit a sharp failure mode and saturate far below multi-head performance.** For $h = 1$, performance improves from $D_K = 64$ to 128, but for $D_K \geq 256$ training becomes bimodal: a substantial fraction of seeds collapse to the NULL baseline (test accuracy $\approx 0$), while the remaining runs converge to a partial solution near $0.81$ and do not improve meaningfully with larger $D_K$. No such complete collapse is observed for $h \geq 2$. This indicates that simply increasing total key width is not sufficient in the single-head setting: splitting the QK budget into multiple heads is critical both for representational performance *and* optimization stability.

**Takeaway.** Even with a full Transformer block, real (non-orthogonal) frozen GPT-2 embeddings, and a task that never requires attending to multiple context positions, we observe a strong and robust multi-head advantage at fixed total key dimension. The best-performing partitions use several heads (typically $h \in \{4, 8\}$ in this regime), while overly many heads at very small $D_K$ can be counterproductive (e.g. $h = 8$ at $D_K = 64$, where $d_k = 8$). Overall, these findings mirror the capacity-based prediction from Section 5: when embeddings are compressed and superposed, distributing a fixed QK budget across multiple heads can substantially reduce interference and improve retrieval accuracy.

## B. More details on our experiments

### B.1. Experimental implementation details

This appendix reproduces the full experimental protocol (model specification, context sampling procedure, loss, optimization, early stopping, and evaluation criteria) described in Section 6.

Our experiments instantiate the upper-bound model from Section 3 as follows. The parameters are two learned projections $W_Q, W_K \in \mathbb{R}^{d_{\text{model}} \times D_K}$ and a *single global scalar threshold* $\tau$. We conceptualize $W_Q, W_K$ as $h$ head blocks of width $d_k = D_K/h$. Scoring is done for a context matrix $X_{\mathcal{C}} \in \mathbb{R}^{\ell \times d_{\text{model}}}$, where head $k$ produces $S^{(k)} = Q^{(k)}(K^{(k)})^{\top} \in \mathbb{R}^{\ell \times \ell}$ with $Q^{(k)} = X_{\mathcal{C}} W_Q^{(k)}$ and $K^{(k)} = X_{\mathcal{C}} W_K^{(k)}$. Scores are aggregated by elementwise max across heads: $S_{\max} = \max_k S^{(k)}$. At evaluation time we predict an edge ($p \to q$) iff $S_{\max}(p, q) > \tau$. This matches the theoretical mechanism exactly: there is *no* $1/\sqrt{d_k}$ scaling, *no* softmax, and *no* value pathway—so capacity is purely key–query driven.

Our primary testbed is the family of permutation graphs $\{(V, E_{\pi})\}$ with $|V| = m$ and $E_{\pi} = \{(i, \pi(i)) : i \in V\}$, where $\pi$ is a uniformly random permutation. This realizes the $m' = m$ constructive case used in our upper bound and isolates the single-target setting in which head specialization is most interpretable. Consistent with our constructions, each node $i \in V$ has a fixed embedding $x_i \in \mathbb{R}^{d_{\text{model}}}$ drawn i.i.d. from $\mathcal{N}(0, I/d_{\text{model}})$ and then $L_2$-normalized. Embeddings are frozen throughout training and evaluation. This both aligns with the random (nearly orthogonal) embedding assumption in our proofs and makes $D_K$ the sole capacity knob. An example is a context $\mathcal{C}$ of length $\ell$ (baseline $\ell = 16$). To prevent degenerate class imbalance when $\pi(i)$ often falls outside $\mathcal{C}$, we enforce a target per-context positive rate $\rho \in (0, 1)$ as follows:

1. Sample a set $S \subseteq V$ of $\ell$ distinct nodes uniformly.

2. Sample $b \sim \text{Binomial}(\ell, \rho)$ and choose $U \subseteq S$ with $|U| = b$.

3. For each $i \in U$, if $\pi(i) \notin S$ replace a random $j \in S \setminus \{i\}$ by $\pi(i)$, preserving $|S| = \ell$ and distinctness.

All of our experiments use $\rho = 0.5$. This preserves the RGR semantics (positives remain exactly those $(i, \pi(i))$ that land in the same context) while reducing the time required to train.

For each experiment we sample one permutation $\pi$ and one embedding matrix $X$ using a fixed seed. We then generate a validation set of 500 contexts and a held-out test set of 2,000 contexts with the same $(\ell, \rho)$ distribution. We then draw training contexts on the fly from the same generator (one context per optimization step).

We train $W_Q, W_K, \tau$ by minimizing a weighted logistic loss on all ordered pairs within a context:

$$z_{pq} = \alpha\big(S_{\max}(p, q) - \tau\big),$$

$$\mathcal{L} \;=\; \frac{1}{|\mathcal{C}|^2} \sum_{p,q} \Big[ \underbrace{\text{softplus}(-z_{pq}) y_{pq}}_{\text{positive term}} \cdot \underbrace{\text{pos}_{\text{weight}}}_{=\ell-1}$$
$$+\; \text{softplus}(z_{pq})(1 - y_{pq}) \Big],$$

where $y_{pq} = 1$ iff $(v_{i_p}, v_{i_q}) \in E$. The weighting $\text{pos}_{\text{weight}} = \ell - 1$ reflects that each source has at most one positive among $\ell$ candidates.

We use AdamW with learning rate $10^{-3}$ and weight decay 0. Parameters are initialized with $W_Q, W_K \sim \mathcal{N}(0, 1/\sqrt{d_{\text{model}}})$ and $\tau = 0$. The logit sharpness is $\alpha = 10$. We train for a number of steps with early stopping: every 500 steps we compute validation micro-F1; if it exceeds 0.995 for 5 consecutive checks, training halts. The number of steps increases with problem complexity. We use one context per step (contexts are small and independent), which keeps the implementation close to the theoretical algorithm and avoids artifacts from large mini-batches.

All evaluation is conducted on the fixed held-out test set of 2,000 contexts using the *single learned* threshold $\tau$ shared across all contexts. Our metric is *Micro-F1* over all ordered pairs across all test contexts. This directly measures correctness of binary edge recognition per the RGR objective. While the *stopping rule* uses validation F1 $> 0.995$, the *minimum $D_K$* we report below is extracted on the *test* set using a looser criterion: the smallest $D_K$ achieving mean micro–F1 $\geq 0.99$ for at least one $h$. We use 0.99 to keep a margin from the stopping rule. All tables and statements about minimum $D_K$ are based on this 0.99 test criterion.

## B.2. Result details

We provide additional detail on the results, the configurations used to obtain them, and the methodology for error bars.

METHODOLOGY FOR ERROR INTERVAL
CONSTRUCTION

**Display CIs for F1 curves.** Unless otherwise noted, error bars are 95% $t$-intervals across seeds: $\bar{F}_1 \pm t_{0.975,\,n-1}\, s/\sqrt{n}$, where $n$ is the number of runs and $s$ their sample standard deviation. Intervals reflect training-run variability with a fixed test set.

**Minimum key dimension $D_K^\star$.** The error interval for the minimum total key dimension, $D_K^\star$, is designed to reflect the uncertainty in the F1 score. For any given model configuration, we determine a central estimate along with an optimistic lower bound and a conservative upper bound, all based on a required F1 score of at least 0.99.

Let the mean F1 score from a set of trials be $\bar{F}_1$, with its corresponding 95% confidence interval being

$[F_{1,\text{low}}, F_{1,\text{high}}]$. The three reported values for $D_K^\star$ are defined as follows:

- **Central Estimate:** The primary value reported. It's the minimum $D_K$ found for which the **mean F1 score** meets the performance threshold ($\bar{F}_1 \geq 0.99$).

- **Conservative Upper Bound:** This is the minimum $D_K$ for which the **lower bound of the F1 confidence interval** meets the threshold ($F_{1,\text{low}} \geq 0.99$). This stricter condition identifies the $D_K$ needed to be 95% confident that the true performance is sufficient.

- **Optimistic Lower Bound:** This is the minimum $D_K$ for which the **upper bound of the F1 confidence interval** meets the threshold ($F_{1,\text{high}} \geq 0.99$). This looser condition identifies the $D_K$ for which it is merely plausible that the true performance is sufficient.

**Optimal number of heads.** Let $h^\star$ be the head count achieving $D_K^\star$ (ties broken by larger $\bar{F}_1$). We form a candidate pool of head counts whose tested $D_K$ lies within 10% of $D_K^\star$. Each candidate is compared to $h^\star$ using a *paired* two-sided $t$-test on per-seed F1; candidates with $p > 0.05$ are labeled "not significantly different" and retained.[7] The reported interval spans the minimum and maximum head counts retained.

FURTHER DETAILS ON RESULTS

Figure 9 lists $D_K^\star$, the minimum total key dimension, found for each configuration of $m$ and $d_{\text{model}}$ we tested. We use our minimum key dimension $D_K^\star$ intervals methodology, with the upper right corner being the upper bound and the lower left corner being the lower bound. The $d_K^\star$ (per head key size) used to achieve these $D_K^\star$s are shown in Table 1, for numbers in the main sweep.

| $m$ | $d_{\text{model}}$ | | |
|---|---|---|---|
| | 16 | 32 | 64 |
| 64 | 5 | 6 | 5 |
| 128 | 11 | 21 | 7 |
| 256 | 6 | 16 | 24 |
| 512 | 7 | 9 | 14 |

*Table 1.* **Per-head key dimension $d_k$ from the main sweep.**

These are found using the training step upper bounds shown in Table 2, where we increase the steps as the problem size and complexity increases.

---

[7]We do not interpret $p > 0.05$ as proof of equivalence; it only indicates insufficient evidence of a difference at $\alpha = 0.05$.

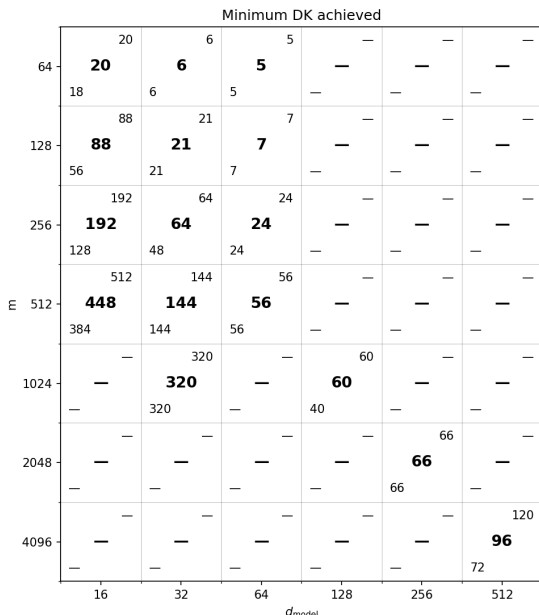

*Figure 9.* **Minimum total key dimension** $D_K^\star$. Upper right and lower left numbers represent confidence range; methodology described in the text.

Also, we provide additional examples of our findings from the main sweep of configurations in Fig. 10.

In Fig. 11, we plot the number of heads used in the optimal found configuration versus the compression $m/d_{\text{model}}$.

### B.3. Experiments for Denser graphs

We next consider denser graphs in the same max-over-heads model variant. Our theoretical results demonstrate tight or nearly tight asymptotic bounds for a broad class of denser graphs; this section seeks to validate and refine those results through experiments in a setting that mirrors our theoretical model. In particular, we examine $r$ regular graphs, and study the impact of scaling $r$.

Unless noted, the architecture, training loop, optimizer, early-stopping protocol, evaluation metric, and data split are identical to the permutation-graph experiments. The only substantive differences are:

- **Task / graph family.** We replace the permutation graph (out-degree = 1) with a directed $r$-regular graph on $m$ nodes (every node has exactly $r$ out-neighbors and $r$ in-neighbors; no self-loops, no multi-edges). Concretely, the graph is sampled as the union of $r$ random perfect matchings ("layers"), each built by a randomized greedy permutation under "forbidden" constraints to avoid self-loops and duplicates; we verify that all in-degrees equal $r$.

| $m$ | $d_{\text{model}}$ | Training step cutoff |
|---|---|---|
| 64 | 16, 32, 64 | 20,000 |
| 128 | 16, 32, 64 | 20,000 |
| 256 | 32, 64 | 20,000 |
| 256 | 16 | 30,000 |
| 512 | 32, 64 | 20,000 |
| 512 | 16 | 80,000 |
| 1024 | 128 | 80,000 |
| 2048 | 256 | 80,000 |
| 4096 | 512 | 200,000 |

*Table 2.* **Training step cutoffs by configuration.** Default cutoff is 20,000 steps, with extended budgets for larger problem sizes.

- **Labeling.** For a context $X_{\mathcal{C}}$, an ordered pair $(p \to q)$ is positive iff $q \in N^+(p)$ and both $p, q \in \mathcal{C}$. Thus each row can contain up to $r$ positives (vs. at most one in the permutation case). Prediction still uses the per-head scores $S^{(k)}$, an elementwise max across heads $S_{\max}$, and a single learned global threshold $\tau$.

- **Context construction (fixed target-in-context rate).** We retain the same target-in-context rate $\rho$, but now implement it over sources: we draw $b \sim \text{Binom}(\ell, \rho)$ sources from the sampled context and ensure that for each selected source $u$ at least one out-neighbor of $u$ is included in the context. When $r=1$ this reduces to the permutation procedure.

- **Class-imbalance weighting.** Because the number of positives per context grows with $r$, we replace the fixed positive weight $(\ell-1)$ used for permutations with a batch-adaptive weight

$$w_+ = \frac{\#\text{negatives}}{\#\text{positives}} \tag{5}$$

computed per context inside the same logistic loss on logits $\alpha(S_{\max} - \tau)$ (with the same $\alpha$).

Everything else (frozen, normalized Gaussian node embeddings; idealized attention with no softmax/value path; elementwise max across heads; single learned $\tau$; AdamW with the same hyperparameters; validation/test protocol; and the micro-F1 reporting criterion) is as in the baseline/core permutation-graph experiments. We also here test the specific case of $m = 128$ and $d_{\text{model}} = 32$ and scale $r$ over $\{1, 2, 4, 8, 16\}$. Given the consistency of our results across different values of $m$ and $d_{\text{model}}$, we believe that this serves as a good proxy for more general behavior.

**Findings and impact (denser graphs).** Moving from permutation graphs ($r=1$) to denser, $r$-regular graphs preserves the qualitative behavior of the capacity transition

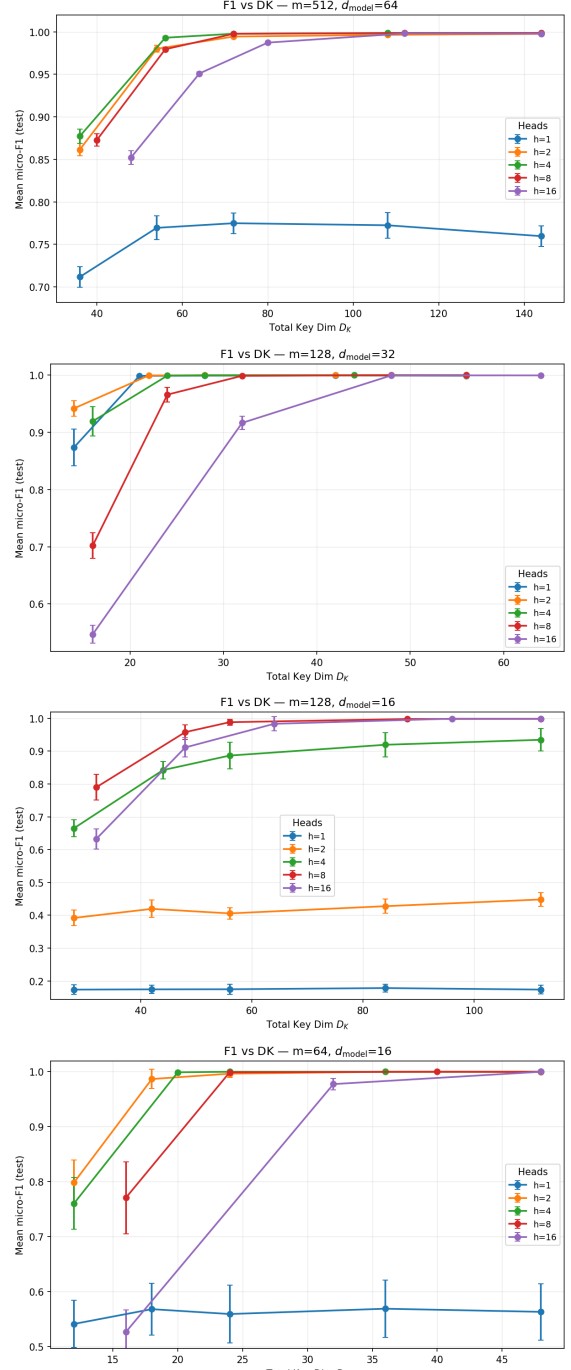

Figure 10. **Example F1–$D_K$ curves.** Each panel fixes $(m, d_{\text{model}})$ and sweeps heads $h$ and $D_K = h\,d_k$; markers show mean test micro–F1 and error bars are 95% CIs over 10 runs. The transition from failure to success occurs at a configuration-specific $D_K$ threshold which is dependent on $h$.

and sharpens several quantitative predictions about how the key–query budget should scale.

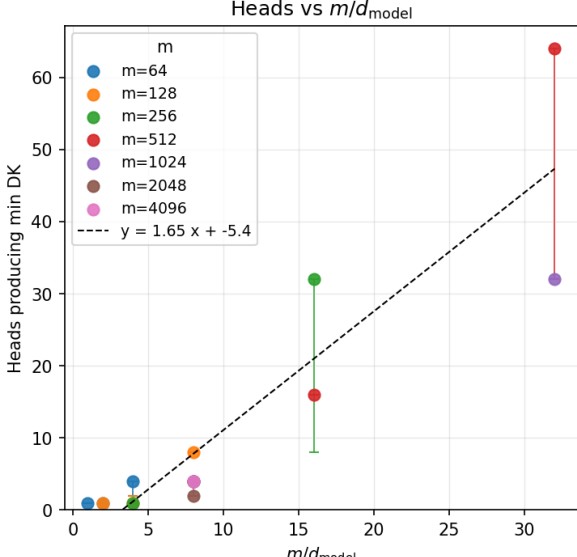

Figure 11. The number of heads needed grows approximately linearly with compression; the dashed line shows a least-squares fit. See text for a description of the error bars. We do not tie the line to the origin, since heads are clipped at $h \geq 1$.

**Sharp capacity transition persists with density.** For fixed $m$ and $d_{\text{model}}$, micro-F1 as a function of the total key budget $D_K$ remains almost step-like: performance stays low until a small window in $D_K$ where it rapidly approaches perfect recovery, and multi-head models cross the threshold at smaller $D_K$ than single-head models. This is visible for $r=2$ and $r=4$ in the F1–vs–$D_K$ sweeps in Figure 12: 1–2 heads plateau well below perfect accuracy, whereas 4–8 heads exhibit a sudden jump to micro-F1 $\approx 1$. Increasing graph density does not blur the phase transition; it simply shifts it to the right, as expected from the larger number of edges that must be separated.

**Capacity is governed by edges, not vertices.** Normalizing by the *edge* budget

$$x = \frac{m'}{d_{\text{model}}} \log_2 m = \frac{r\,m}{d_{\text{model}}} \log_2 m, \qquad (6)$$

the empirically minimal $D_K$ required for high accuracy collapses to a single linear trend across degrees $r \in \{1, 2, 4, 8, 16\}$ (Figure 5). A single proportionality constant fits all densities:

$$D_K^{\star} \approx 0.46\,x, \qquad (7)$$

so, as in our theoretical results, the total key dimension tracks $m'$ much more tightly than $m$. Intuitively, the model's *where-to-attend* budget must scale with the number of *encoded relationships*; increasing vertices without adding edges exerts far less pressure on $D_K$.

**Head count scales with edge density.** The number of heads at the empirical threshold grows with graph density and is well predicted (up to a small constant factor) by the edge-normalized ratio $m'/d_{\text{model}}$ (Figure 13). In our runs with $m{=}128$ and $d_{\text{model}}{=}32$, the head count that achieves the smallest passing $D_K$ increases roughly linearly with $r$ and stays close (within a factor of $\sim 2$) to the theoretical target $h^\star \propto m'/d_{\text{model}} = (r\,m)/d_{\text{model}}$. This reinforces a capacity-based rationale for multi-head attention: as more edges are superposed in the compressed embedding space, distributing the key–query budget across more, narrower heads reduces interference and lowers the required $D_K$.

**Relation to bounds.** For denser graphs (Figure 5, $r =$ 16, the empirical thresholds lie slightly below our constructive upper-bound designs—i.e., we need slightly less total key dimension than the construction would guarantee—suggesting either slack in the analysis or other factors the model exploits during training. At the same time, as $r$ grows the gap between the constructive upper bound and the information-theoretic lower bound grows. Together, these trends indicate the true optimum is closer to the lower-bound scaling and that there is room to tighten (or redesign) dense-graph constructions.

**Takeaway.** Across densities, capacity remains a threshold phenomenon; the threshold is controlled by the *number of edges* $m'$ rather than the vertices $m$; and the optimal head count scales with $m'/d_{\text{model}}$. Practically, when budgeting attention for relational workloads, counting *relationships* is a more reliable guide than counting vocabulary size. The denser-graph experiments therefore extend the capacity law beyond permutations and provide further evidence for a principled multi-head advantage that grows with graph density.

## C. Extension to Softmax

We now instantiate the QK-only model of §3 with standard scaled dot-product attention: each head applies a *softmax over the items in the context* (with the usual $d_k^{-1/2}$ scaling inside the logits). As in §3, we omit the value channel and treat attention weights as per-pair scores. We aggregate across heads by *summation* and then threshold the aggregated score. We view summation as the natural aggregation computation, given the lack of a value channel. In App. D, we prove that this model variant requires $D_K = \Omega\!\left(\frac{m'}{d_{\text{model}}}\log\frac{m^2}{m'}\right)$, and we give a construction approaching this bound.

**Explicit Construction.** We use the same Gaussian unit-norm embedding and the same random-signature de-embedding scheme as in Algorithm 1 and Theorem 4.1. Given a context $\mathcal{C} \subseteq V$ of length $\ell$, for head $k$ we form

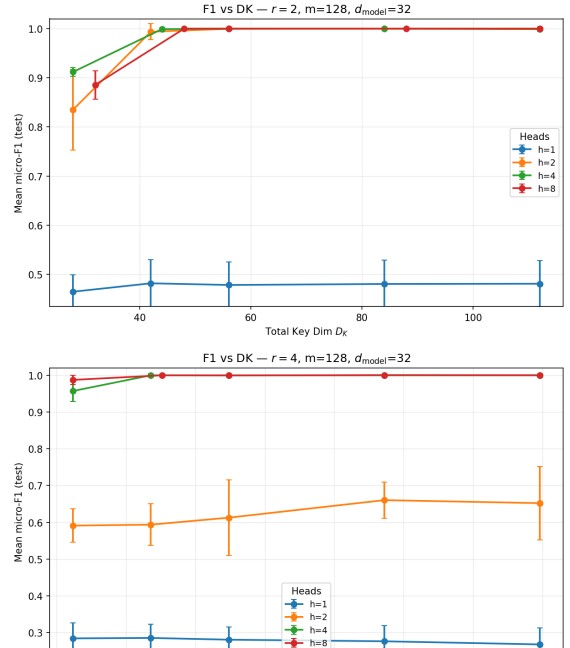

Figure 12. **Sharp transition persists and shifts right for $r \in \{2, 4\}$.** Mean test micro-F1 vs. total key dimension $D_K$ for $m{=}128, d_{\text{model}}{=}32$. More heads reach perfect recovery at smaller $D_K$; for $r = 2$, a single head plateaus well below 1.0, while for $r = 4$ 2 heads has the same property. More heads exhibit a sudden jump to $\approx 1.0$.

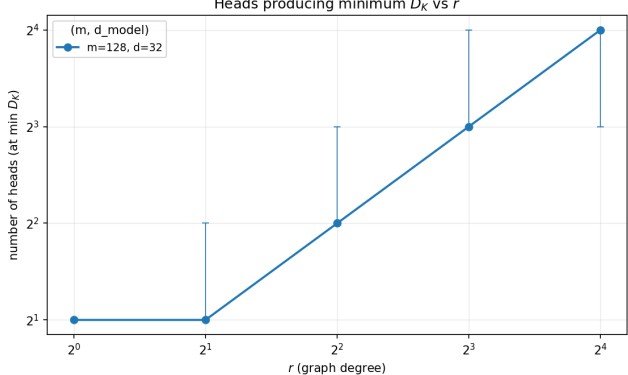

Figure 13. **Optimal head count rises with density.** Number of heads that achieves the smallest passing $D_K$ versus graph degree $r$ (here $m{=}128, d_{\text{model}}{=}32$). The trend tracks $m'/d_{\text{model}} = (r\,m)/d_{\text{model}}$ up to a constant factor and shows increasing benefit from additional heads as the graph becomes denser. Error bars indicate neighboring head counts that tied for the minimum within the measurement resolution (using the same methodology as described in Appendix B.2).

the usual (scaled) logits and per-head probabilities

$$L_{ij}^{(k)} := \frac{S_{ij}^{(k)}}{\sqrt{d_k}}, \quad p_{ij}^{(k)} := \frac{\exp\!\left(L_{ij}^{(k)}\right)}{\sum_{t \in \mathcal{C}} \exp\!\left(L_{it}^{(k)}\right)} \quad (j \in \mathcal{C}). \tag{8}$$

**Algorithm 2** Softmax Construction for Permutations with Compressive Embeddings

1: **Input:** Permutation graph $G = (V, E)$ with $\pi : V \to V$; Gaussian unit-norm embedding matrix $X \in \mathbb{R}^{m \times d_{\text{model}}}$ (rows $\mathbf{x}_i^\top$).
2: **Parameters:** per-head width $d_k = C(\log m)^2$; uniform block size $b = \lfloor \alpha d_{\text{model}}/\sqrt{d_k} \rfloor$ for a small absolute constant $\alpha > 0$; $h = O(m/b) = O(m \log m/d_{\text{model}})$ heads; random Rademacher signatures as in Construction II; **threshold** $\tau = \frac{1}{2}$.
3: **Partition sources/targets:** Split $V$ into disjoint blocks $V_1, \ldots, V_h$ of size $|V_k| \le b$ and $T_k := \{\pi(s) : s \in V_k\}$.
4: **Templates and de-embedding:** As in Algorithm 1, build one-hot-space templates $W'_{Q,(k)}, W'_{K,(k)}$ and set $W_Q^{(k)} = X^\top W'_{Q,(k)}, W_K^{(k)} = X^\top W'_{K,(k)}$.
5: **Per-head scores:** For each head $k$ and $(i, j)$, compute $S_{ij}^{(k)} = \langle \mathbf{q}_i^{(k)}, \mathbf{k}_j^{(k)} \rangle$, then logits $L_{ij}^{(k)} = S_{ij}^{(k)}/\sqrt{d_k}$ and softmax $p_{ij}^{(k)}$ over $j \in \mathcal{C}$.
6: **Aggregate across heads:** $A_{ij} = \sum_{k=1}^h p_{ij}^{(k)}$; declare edge $(i, j)$ present iff $A_{ij} > \tau$.

The *aggregated score* we threshold is the sum over heads

$$A_{ij} := \sum_{k=1}^h p_{ij}^{(k)}. \tag{9}$$

We declare that $i$ has an edge to $j$ iff $A_{ij} > \tau$ with $\tau := \frac{1}{2}$.

The only substantive changes from Construction II are (i) computing the per-head softmax (with scaling) over $j \in \mathcal{C}$ and (ii) replacing $\max_k$ aggregation over heads by a sum over $k$.

## C.1. Main guarantee and efficiency

**Theorem C.1** (Softmax analogue of Construction II). *Assume the setup of Algorithm 2 with uniform block size $b = \lfloor \alpha d_{\text{model}}/\sqrt{d_k} \rfloor$ (so $h = O(m \log m/d_{\text{model}})$) and Gaussian unit-norm embeddings, where $d_{\text{model}} \ge c_0(\log m)^2$ for a sufficiently large absolute constant $c_0$. There exist absolute constants $C, c_0, C_\ell, \alpha, \gamma > 0$ such that if $d_k = C(\log m)^2$ and $\ell \ge C_\ell h$, then for all sufficiently large $m$, with probability at least $1 - m^{-3}$ over the draw of $(X, \text{signatures})$, the following holds: for any fixed context $\mathcal{C} \subseteq V$ of length $\ell$, simultaneously for all sources $i$ and all $j \in \mathcal{C}$,*

$$A_{i,\pi(i)} > \frac{1}{2} \quad \text{and} \quad A_{ij} < \frac{1}{2} \quad (j \ne \pi(i)).$$

*Consequently, for every source $i$, if the unique target $\pi(i)$ is in the context, then using the fixed threshold $\tau = \frac{1}{2}$ recovers that target within the context, and if $\pi(i)$ is not in*

*the context, then no edge is identified. Finally, the total key dimension satisfies*

$$\boxed{D_K = h\, d_k = \Theta\!\left( \frac{m}{d_{\text{model}}} (\log m)^3 \right).}$$

Before the proof we record two deterministic per-head bounds. Recall $L_{ij}^{(k)} = S_{ij}^{(k)}/\sqrt{d_k}$, and for $i \notin V_k$ write $\mathbf{q}_i = \sum_{s \in V_k} \delta_{i,s} \mathbf{w}_{\pi(s)}$.

**Lemma C.2** (Small-block non-owner logits). *With $d_k = C(\log m)^2$, $d_{model} \ge c_0(\log m)^2$, uniform block size $b = \lfloor \alpha d_{model}/\sqrt{d_k} \rfloor$, and any fixed $A > 0$, there are constants $K_0, K_1$ such that w.p. $\ge 1 - m^{-A}$, simultaneously for every head $k$, source $i \notin V_k$, and target $j$,*

$$|L_{ij}^{(k)}| \le K_0 \quad (j \notin T_k),$$
$$|L_{ij}^{(k)}| \le K_0 + K_1 \sqrt{\tfrac{d_k \log m}{d_{model}}} \quad (j \in T_k).$$

*Proof.* Since $i \notin V_k$, $S_{ij}^{(k)} = \mathbb{I}[j \in T_k] (\mathbf{q}_i \cdot \mathbf{w}_j) + \mathbf{q}_i \cdot \sum_{t \in T_k} \delta_{j,t} \mathbf{w}_t$. The second ($N_3$-type) term has conditional mean $d_k \sum_{t \in T_k} \delta_{i,\pi^{-1}(t)} \delta_{j,t}$ and fluctuation $\le K \|\boldsymbol{\delta}_{i,V_k}\|_2 \|\boldsymbol{\delta}_{j,T_k}\|_2 \sqrt{d_k \log m}$. By item 2 at block size $b$, $\|\boldsymbol{\delta}_{i,V_k}\|_2^2, \|\boldsymbol{\delta}_{j,T_k}\|_2^2 \le b/d_{\text{model}} + o(1) \le 2\alpha/\sqrt{d_k}$, so the fluctuation over $\sqrt{d_k}$ is $o(1)$; the mean, via item 3 at $|T_k| \le b$ (with $F = \{v : \pi(v) = v\}$), is $\sqrt{d_k} \cdot [\mathbb{I}[i = j]|T_k \cap F|/d_{\text{model}} + C_3(\sqrt{b \log m}/d_{\text{model}} + \log m/d_{\text{model}})] = O(1)$ (using $b \le \alpha d_{\text{model}}/\sqrt{d_k}$, $d_k = C(\log m)^2$, $d_{\text{model}} \ge c_0(\log m)^2$; the first term is $\le \sqrt{d_k}\, b/d_{\text{model}} \le \alpha$). Thus $|L_{ij}^{(k)}| \le K_0$ for $j \notin T_k$. For $j \in T_k$ with $r = \pi^{-1}(j) \in V_k$, $\mathbf{q}_i \cdot \mathbf{w}_j = d_k \delta_{i,r} + (\text{Rademacher sum})$, contributing $\sqrt{d_k} \delta_{i,r} = O(\sqrt{d_k \log m}/d_{\text{model}})$ plus an $o(1)$ fluctuation, giving the in-block bound. Union bound over $O(m^3)$ triples. $\square$

**Lemma C.3** (Uniform denominator lower bound). *For suitable $c, c_0, C, \alpha, C_\ell > 0$ and any $A > 0$, w.p. $\ge 1 - m^{-A}$, simultaneously for every head $k$, source $i \notin V_k$, and context $\mathcal{C} \ni i$ with $\ell \ge C_\ell h$, $\sum_{j \in \mathcal{C} \setminus \{i\}} e^{L_{ij}^{(k)}} \ge c\ell$.*

*Proof.* On the event of Lemma C.2 the bound is deterministic. If $d_{\text{model}} \ge d_k \log m$ then $\sqrt{d_k \log m/d_{\text{model}}} \le 1$, so every $L_{ij}^{(k)} \ge -(K_0 + K_1)$ and the sum is $\ge (\ell - 1)e^{-(K_0+K_1)} \ge c\ell$. Else $d_{\text{model}} < d_k \log m$: every source block has size $\le b$, so $h \ge m/b$ and $\ell \ge C_\ell m/b$, giving $|T_k|/\ell \le b^2/(C_\ell m) \le \alpha^2 C(\log m)^4/(C_\ell m) \to 0$; thus $|T_k| + 1 \le \ell/2$ and the $\ge \ell/2$ out-of-block targets ($L_{ij}^{(k)} \ge -K_0$) give $\sum \ge \frac{\ell}{2} e^{-K_0} \ge c\ell$. $\square$

*Proof of Theorem C.1.* We demonstrate this using the following steps:

**Step 1: Raw-score separation (owner head).** Let $k^\star$ be the unique head with $i \in V_{k^\star}$. The signal/noise separation in the proof of Theorem 4.1 holds verbatim for any $d_k \geq C \log m$ and any block size $\leq d_{\text{model}}$ (the estimates only improve as the block shrinks and $d_k$ grows), so it applies at our block size $b$ and $d_k = C(\log m)^2$, yielding absolute constants such that, simultaneously for all $i$ and all $j$,

$$S_{i,\pi(i)}^{(k^\star)} \geq \tfrac{3}{4} d_k, \qquad S_{ij}^{(k^\star)} \leq \tfrac{1}{4} d_k \quad (j \neq \pi(i)). \quad (10)$$

Dividing by $\sqrt{d_k}$ gives the corresponding logit gap

$$L_{i,\pi(i)}^{(k^\star)} \geq \tfrac{3}{4}\sqrt{d_k}, \qquad L_{ij}^{(k^\star)} \leq \tfrac{1}{4}\sqrt{d_k} \quad (j \neq \pi(i)). \quad (11)$$

**Step 2: Softmax calibration within the owner head.** We use the standard calibration lemma.

**Lemma C.4** (Softmax calibration). *If $z_\star \geq a$ and $z_t \leq b$ for all $t \neq \star$ in a set of size $\ell$, then*

$$\frac{e^{z_\star}}{\sum_t e^{z_t}} \geq \frac{1}{1 + (\ell-1)e^{-(a-b)}}.$$

Applying Lemma C.4 to head $k^\star$ with $a = \tfrac{3}{4}\sqrt{d_k}$ and $b = \tfrac{1}{4}\sqrt{d_k}$ from (11) gives

$$p_{i,\pi(i)}^{(k^\star)} \geq \frac{1}{1 + (\ell-1)e^{-\frac{1}{2}\sqrt{d_k}}}. \quad (12)$$

Using $\frac{1}{1+u} \geq 1-u$ for $u \geq 0$ and $\ell \leq m$, we further obtain

$$p_{i,\pi(i)}^{(k^\star)} \geq 1 - (\ell-1)e^{-\frac{1}{2}\sqrt{d_k}} \geq 1 - m\,e^{-\frac{1}{2}\sqrt{d_k}}.$$

If $d_k \geq C(\log m)^2$, then $e^{-\frac{1}{2}\sqrt{d_k}} \leq e^{-\frac{1}{2}\sqrt{C}\log m} = m^{-\frac{1}{2}\sqrt{C}}$, so

$$m\,e^{-\frac{1}{2}\sqrt{d_k}} \leq m^{1-\frac{1}{2}\sqrt{C}} = m^{-\gamma} \text{ with } \gamma := \tfrac{1}{2}\sqrt{C} - 1.$$

Choosing $C$ large enough ensures $\gamma > 0$, and we conclude that

$$p_{i,\pi(i)}^{(k^\star)} \geq 1 - m^{-\gamma}. \quad (13)$$

Similarly, for any $j \neq \pi(i)$ we have

$$p_{ij}^{(k^\star)} \leq \frac{e^{\frac{1}{4}\sqrt{d_k}}}{e^{\frac{3}{4}\sqrt{d_k}}} = e^{-\frac{1}{2}\sqrt{d_k}} \leq m^{-\gamma}, \quad (14)$$

after possibly decreasing $\gamma$ by an absolute factor (absorbed by increasing $C$).

**Step 3: Non-edge aggregate via deterministic per-head bounds.** Steps 1–2 give the owner head $p_{i,\pi(i)}^{(k^\star)} \geq 1 - m^{-\gamma}$

and, for $j \neq \pi(i)$, $p_{ij}^{(k^\star)} \leq m^{-\gamma}$; the owner separation (10) holds at block size $b \leq d_{\text{model}}$ since it only improves with smaller blocks and larger $d_k$. Fix a non-edge $j \neq \pi(i)$ and split $A_{ij} = \sum_k p_{ij}^{(k)}$ into three groups. (a) The owner head contributes $\leq m^{-\gamma}$. (b) Target blocks $\{T_k\}$ partition $V$, so at most one non-owner head $k_j$ has $j \in T_{k_j}$; by Lemma C.2 its numerator is $e^{L_{ij}^{(k_j)}} \leq \exp(K_0 + K_1\sqrt{d_k \log m/d_{\text{model}}}) = m^{o(1)}$, and by Lemma C.3 its denominator is $\geq c\ell \geq cC_\ell m\sqrt{d_k}/(\alpha d_{\text{model}})$, so $p_{ij}^{(k_j)} = o(1)$ (both $d_{\text{model}} \leq (\log m)^4$ and $> (\log m)^4$ give $o(1)$). (c) Every other non-owner head has $j \notin T_k$, so $L_{ij}^{(k)} \leq K_0$ and $p_{ij}^{(k)} \leq e^{K_0}/(c\ell)$; summing over $\leq h$ heads with $\ell \geq C_\ell h$ gives $\leq e^{K_0}/(cC_\ell)$. Hence $A_{ij} \leq m^{-\gamma} + o(1) + e^{K_0}/(cC_\ell) < \tfrac{1}{2}$ for $C_\ell$ large and $m$ large, while $A_{i,\pi(i)} \geq p_{i,\pi(i)}^{(k^\star)} > \tfrac{1}{2}$.

**Step 4: Budget.** $h = O(m/b) = O(m\sqrt{d_k}/d_{\text{model}}) = O(m \log m/d_{\text{model}})$, so $D_K = h\,d_k = \Theta\big(\frac{m}{d_{\text{model}}}(\log m)^3\big)$. $\qquad\square$

## C.2. Multiple heads remain advantageous under softmax

We next show that the multi-head advantage we saw in the max over heads model variant carries over to softmax, albeit with a slightly smaller ratio. We saw above that softmax normalization introduces a new—and unavoidable—per-head requirement: even with perfect separation of raw scores, converting an additive margin into a probability $1-\varepsilon$ *uniformly over contexts of length $\ell$* forces $d_k = \Omega((\log \ell)^2)$ under the standard $d_k^{-1/2}$ scaling. This cost is orthogonal to the compressive de-embedding noise that drove the multi-head advantage in §5. In the softmax model, the two constraints simply stack: (i) the owner head must achieve a raw score gap large enough to survive de-embedding leakage (as in (1)–(2)), and (ii) that gap must be at least $\Omega(\sqrt{d_k})$ so that Lemma C.4 pushes the owner-head probability to $1 - \varepsilon$ against $\ell$ distractors. The first favors *more* heads (to shrink the per-head block size $B$), while the second imposes a universal $(\log \ell)^2$ floor on $d_k$.

**Proposition C.5** (Multi-head vs. single head with softmax, within the compressive template). Under the setup of Algorithm 2 and Construction II (Gaussian unit-norm embeddings; Rademacher signatures; de-embedding noise scaling (1)), any design that succeeds w.h.p. uniformly over all contexts $\mathcal{C}$ of length $\ell \leq m$ that *contain* the true target must satisfy the combined per-head width condition

$$d_k \geq \max\Big\{ C_1(\log \ell)^2, \ C_2\frac{B^2}{d_{\text{model}}^2}\log m \Big\}, \quad (15)$$

where $B$ is the number of targets served by the head (the block size). Consequently:

**Single head** ($B = m$):

$$D_K^{\text{single}} \geq \Omega\left(\frac{m^2}{d_{\text{model}}^2} \log m + (\log m)^2\right), \quad (16)$$

**Multi-head** ($B = \Theta(d_{\text{model}}/\log m)$):

$$D_K^{\text{multi}} = \Theta\left(\frac{m}{d_{\text{model}}}(\log m)^3\right). \quad (17)$$

Hence, in the compressive regime $m \gg d_{\text{model}}$, the single-head total key dimension is asymptotically larger by at least

$$\frac{D_K^{\text{single}}}{D_K^{\text{multi}}} \gtrsim \frac{(m^2/d_{\text{model}}^2) \log m}{(m/d_{\text{model}})(\log m)^3} = \frac{m/d_{\text{model}}}{(\log m)^2}. \quad (18)$$

*Proof sketch.* The leakage term $N_3(B) \asymp \frac{B}{d_{\text{model}}}\sqrt{d_k \log m}$ from (1) enforces $N_3 \leq c_1 d_k$, which is equivalent to $d_k \geq C_2 \frac{B^2}{d_{\text{model}}^2} \log m$ (Eq. (2)). This guarantees the raw owner-head separation used in Step 1 of the softmax proof (inequalities (10)–(11)). Lemma C.4 then turns an $\Omega(\sqrt{d_k})$ logit gap into owner-head mass $1 - O(m^{-\gamma})$ provided $d_k \geq C_1(\log \ell)^2$. Summing across heads, each non-owner head contributes $O(1/\ell)$ background mass, summed over $\leq h$ heads to $< \frac{1}{2}$ whenever $\ell \geq C_\ell h$, by the three-group denominator argument of Theorem C.1 (Step 3); hence the fixed threshold $\tau = \frac{1}{2}$ separates the aggregated scores. For the single-head case ($h = 1$) the background term vanishes, but the leakage constraint uses $B = m$, giving the stated lower bound on $d_k$; for the multi-head choice we have $B = \Theta(d_{\text{model}}/\log m)$, so the leakage term becomes $O(1)$ and the softmax floor $C_1(\log m)^2$ dominates, yielding the claimed $d_k^{\text{multi}} = \Theta((\log m)^2)$ and $D_K^{\text{multi}} = \Theta(\frac{m}{d_{\text{model}}}(\log m)^3)$. $\square$

# D. Lower Bounds on Relational Graph Recognition

We begin with a simple bit-budget argument that already forces a large key dimension when each parameter has $O(1)$ effective bits. We then prove a precision-independent lower bound, based on a metric-entropy argument, that holds provided there is a fixed margin separation between positive and negative decisions, as well as a bound on the scale of the parameter and embedding norms. All of our upper bounds satisfy these assumptions.

**Warm-up: fixed-precision (bit-budget) lower bound.** Let $N = m(m-1)$ be the number of ordered, loop-free pairs. Suppose each real parameter in $\{W_Q^{(k)}, W_K^{(k)}, \tau\}$ is represented with $b = \Theta(1)$ effective bits. Counting only the QK channel (and the global threshold), the parameter budget is

$$B = b(2 d_{\text{model}} D_K + 1) \quad \text{bits.}$$

Hence the model can realize at most $2^B$ distinct edge labelings of the $N$ ordered pairs. Uniform recovery of *every* graph with $m'$ edges requires at least $\binom{N}{m'}$ distinct labelings, so $2^B \geq \binom{N}{m'}$. Rearranging gives

$$d_{\text{model}} D_K \geq \frac{1}{2b} \log\binom{m(m-1)}{m'} - O(1),$$

or equivalently,

$$D_K = \Omega\left(\frac{\log\binom{m(m-1)}{m'}}{d_{\text{model}}}\right) = \Omega\left(\frac{m'}{d_{\text{model}}} \log \frac{m^2}{m'}\right).$$

This argument is independent of the aggregator, and thus holds specifically for both our max-over-heads variant and our softmax variant. It holds for any context length $\ell \geq 2$, and does not assume a constant margin between the scores of present and missing edges.

**Precision-agnostic lower bound assuming constant margin.** We now prove the same result without assuming discretization or bit precision: as long as there is a fixed positive margin $\gamma$ separating "yes" from "no" decisions and a fixed bound on the scale of the parameters, the same $\Omega\left(\frac{m'}{d_{\text{model}}} \log \frac{m^2}{m'}\right)$ lower bound holds for real-valued parameters. All of our upper bounds achieve constant margins under the same model, so the comparison is apples-to-apples.

**Definition D.1** (Constant-margin recovery). A parameter choice $\left(\{U_k, V_k\}_{k=1}^h, \tau\right)$ *recovers* a graph $G = (V, E)$ with margin $\gamma > 0$ if, for every ordered pair $(u, v)$ with $u \neq v$,

$$(u, v) \in E \Rightarrow S(u, v) \geq \tau + \gamma,$$

$$(u, v) \notin E \Rightarrow S(u, v) \leq \tau - \gamma,$$

with $S$ given by (21).

**Theorem D.2** (Description-length lower bound for QK). *Fix $m \in \mathbb{N}$ and $m' \in \{0, \ldots, m(m-1)\}$. Suppose a single self-attention QK channel with total key dimension $D_K = \sum_{k=1}^h d_k$ and embeddings $\mathbf{x}_v, v \in V$ can recover every graph in $\mathcal{G}_{m,m'}$ with margin $\gamma \in (0, 1)$ on contexts of any length $\ell \geq 2$. Further, assume that the recovery is achievable with parameters and embeddings whose norms are bounded by a universal constant independent of $m$. Then there exists a constant $c(\gamma) > 0$ such that*

$$d_{model} D_K \geq c(\gamma) \log\binom{m(m-1)}{m'} - O(1), \quad (19)$$

*or equivalently*

$$D_K = \Omega\left(\frac{\log\binom{m(m-1)}{m'}}{d_{model}}\right) = \Omega\left(\frac{m'}{d_{model}} \log \frac{m^2}{m'}\right). \quad (20)$$

*Proof.* We first point out that we can focus only on length-2 contexts. Uniform correctness for RGR requires that, for *every* ordered pair $(u, v) \in V \times V$, the decision "$(u, v) \in E$?" is the same in every context containing $u$ and $v$. In particular it must be correct in the length-2 context $\mathcal{C} = (u, v)$. Hence any lower bound proved using only length-2 contexts applies to the full problem.

For head $k$, write the per-pair score on a length-2 context as

$$s_k(u, v) = \mathbf{x}_u^\top A_k \, \mathbf{x}_v,$$

$$A_k = W_Q^{(k)} W_K^{(k)\top}, \quad \mathrm{rank}(A_k) \le d_k.$$

Define the final score $S(u, v) = \max_k s_k(u, v)$, and decide

$$(u, v) \in E \iff S(u, v) > \tau. \tag{21}$$

We factor each $A_k = U_k V_k^\top$ with $U_k, V_k \in \mathbb{R}^{d_{\mathrm{model}} \times d_k}$, and concatenate $U = [U_1 | \cdots | U_h] \in \mathbb{R}^{d_{\mathrm{model}} \times D_K}$, $V = [V_1 | \cdots | V_h] \in \mathbb{R}^{d_{\mathrm{model}} \times D_K}$. The following normalization assumption is without loss of generality by scale homogeneity of the decision rule (simultaneously scaling $(U, V)$ and $\tau$), and our assumption of bounded norms of our embeddings and parameters:

$$\|U\|_F \le 1, \quad \|V\|_F \le 1,$$
$$|\tau| \le 1, \quad \|\mathbf{x}_v\|_2 \le 1 \quad \forall v \in V. \tag{22}$$

Note that without the bounded norms assumption, this normalization step could introduce a dependence on $m$ into the margin $\gamma$, which in turn would carry into the final lower bound.

**Lemma D.3** (Lipschitz property of the Decision Function). *Under the normalization assumptions in* (22)*, for any two parameter settings* $(U, V, \tau)$ *and* $(\tilde{U}, \tilde{V}, \tilde{\tau})$*, the following holds for the decision function:*

$$\left| (S(u, v) - \tau) - (\tilde{S}(u, v) - \tilde{\tau}) \right|$$
$$\le \|U - \tilde{U}\|_F + \|V - \tilde{V}\|_F + |\tau - \tilde{\tau}|. \tag{23}$$

*Sketch.* For each head, using $\|\mathbf{x}_u\|_2, \|\mathbf{x}_v\|_2 \le 1$ and $\|U_k\|_F, \|V_k\|_F \le 1$ (implied by (22)),

$$\left| s_k - \tilde{s}_k \right| = \left| \mathbf{x}_u^\top (U_k V_k^\top - \tilde{U}_k \tilde{V}_k^\top) \mathbf{x}_v \right|$$
$$\le \|U_k - \tilde{U}_k\|_F + \|V_k - \tilde{V}_k\|_F.$$

With our max aggregation rule, we have $\left| S - \tilde{S} \right| \le \max_k \lfloor s_k - \tilde{s}_k \rfloor \le \max_k \|U_k - \tilde{U}_k\|_F + \max_k \|V_k - \tilde{V}_k\|_F \le \|U - \tilde{U}\|_F + \|V - \tilde{V}\|_F$. Finally, $|\tau - \tilde{\tau}|$ adds linearly. $\square$

We also note that the same Lemma holds (using a similar proof) if we instead use either a log-sum-exp aggregation rule or a summation aggregation rule: our lower bounds hold for those cases as well.

We also note that the same covering-number argument applies to other common aggregations (including $\sum_k$ and log-sum-exp across heads), and specifically to our more standard softmax model in which each head applies a softmax over the context and we sum the resulting attention weights across heads before thresholding; see the paragraph "Softmax attention" below.

From our normalization assumption (22), every parameterization lies in the compact ball

$$\mathbb{B} := \left\{ (U, V, \tau) : \|U\|_F \le 1, \|V\|_F \le 1, |\tau| \le 1 \right\},$$

whose radius is at most 1. Therefore, under the normalization (22) and Lemma D.3, the parameter set admits an $\varepsilon$-net (in Frobenius metric for $U, V$ and absolute value for $\tau$) of radius $\varepsilon = \gamma/4$ and size at most

$$N_{\mathrm{cov}}(\varepsilon) \le \left( \frac{C}{\varepsilon} \right)^{2 \, d_{\mathrm{model}} D_K + 1} = \left( \frac{C'}{\gamma} \right)^{2 \, d_{\mathrm{model}} D_K + 1}$$

for absolute constants $C, C' > 0$. Each net point induces a *unique* labeling of the $N = m(m - 1)$ ordered pairs by the $\gamma$-margin, because any perturbation of radius $\varepsilon$ preserves all signs by (23). Therefore at most $N_{\mathrm{cov}}(\varepsilon)$ distinct edge sets can be realized at margin $\gamma$. Since the mechanism must realize all $\binom{N}{m'}$ edge sets of size $m'$, we obtain $\binom{N}{m'} \le N_{\mathrm{cov}}(\varepsilon)$, which rearranges to (19)–(20). $\square$

**Softmax attention.** Theorem D.2 also applies to the softmax version of our model. Uniform recovery requires the decision for a fixed ordered pair $(i, j)$ to be identical in every context containing $i$ and $j$, so it suffices to analyze the length-2 context $\mathcal{C} = (i, j)$. For $\ell = 2$ each head reduces to a two-way softmax:

$$p_{ij}^{(k)} = \frac{\exp(L_{ij}^{(k)})}{\exp(L_{ii}^{(k)}) + \exp(L_{ij}^{(k)})} \tag{24}$$
$$= \sigma\big(L_{ij}^{(k)} - L_{ii}^{(k)}\big), \qquad \|\sigma'\|_\infty \le \tfrac{1}{4},$$

where $\sigma(t) = 1/(1 + e^{-t})$. Using (22) and $L_{ij}^{(k)} = S_{ij}^{(k)}/\sqrt{d_k}$, the mean-value theorem and Cauchy–Schwarz give, for any two settings $(U, V)$ and $(\tilde{U}, \tilde{V})$,

$$|A_{ij} - \tilde{A}_{ij}| \le \tfrac{1}{2}\big(\|U - \tilde{U}\|_F + \|V - \tilde{V}\|_F\big). \tag{25}$$

Thus the decision map remains $O(1)$-Lipschitz, so the same $\varepsilon$-net/margin-stability argument as in the proof of Theorem D.2 yields the identical lower bound (20).

**Bounds for specific regimes.** The theorem implies:

- Exactly $m' = m$ edges (e.g., permutation graphs): $D_K = \Omega\big(\frac{m' \log m}{d_{\text{model}}}\big)$.

- Dense regime with $m' = \Theta(m^2)$: $D_K = \Omega\big(\frac{m'}{d_{\text{model}}}\big)$.

- Sparse regime with $m' = O(m^{2-\epsilon})$ for some $\epsilon > 0$: $D_K = \Omega\big(\frac{m' \log m}{d_{\text{model}}}\big)$.

# E. Further Details on Our Explicit Constructions

We here provide additional details on our explicit constructions from Section 4. The computational complexity of all the constructive algorithms we describe in this paper is $O(m d_{model} \log m)$. This follows fairly directly from the constructions themselves, although a small bit of care is needed to multiply the matrices together in the right order. This is significantly lower than the computation required to train an attention mechanism using gradient descent. We start with the easiest case - permutation graphs with no embedding. This result is subsumed by the second construction, so is only included as a warm up for the more general case.

## E.1. Construction I: Permutation Graphs with One-Hot Embeddings

**Setup.** We start with the following two assumptions: (i) $G$ is a permutation on $m$ items, defined by a function $\pi : V \to V$, where the edges are $E = \{(v_i, v_{\pi(i)}) \mid v_i \in V\}$, and thus $m' = m$. (ii) Node $v_i$ is represented by the one-hot vector $\mathbf{x}_i = \mathbf{e}_i \in \mathbb{R}^m$, setting the model dimension $d_{\text{model}} = m$.

With these assumptions, a single attention head ($h = 1$) suffices, so the total key dimension is $D_K = d_k$. Our goal is to define $W_K, W_Q$, and a global threshold $\tau$ such that the score $S_{ij} = (\mathbf{x}_i W_Q) \cdot (\mathbf{x}_j W_K)$ exceeds $\tau$ iff $j = \pi(i)$. Our construction works for all vertices $V$ of the graph $G$, independent of the current context in our model; we formalize how this applies to specific contexts below.

**Algorithmic Construction (one-hot case)** The core idea is to assign each node $v_j$ a random "signature" via its key vector $\mathbf{k}_j$. The query vector $\mathbf{q}_i$ for $v_i$ is the signature of its target, $v_{\pi(i)}$. The dot product between vectors is maximized when the query signature matches the key signature.

**Theorem E.1** (Single-head recognition under one-hot inputs). *Under the construction above, for $d_k = C \log m$ with $C$ sufficiently large (depending only on $p$), we have with probability at least $1 - m^{-3}$ over the draw of $W_K$ that*

$$S_{i,\pi(i)} > \tau \quad \text{and} \quad S_{ij} < \tau \text{ for all } i \in V, j \neq \pi(i).$$

**Algorithm 3** Construction for Permutation Graphs with One-Hot Inputs

1: **Input:** Graph $G = (V, E)$ defined by permutation $\pi$.
2: **Setup:** Choose a probability $p \in (0, 1/2)$, e.g., $p = 1/4$, and dimension $d_k = C \log m$, for sufficiently large constant $C$.
3: **Construct Key Matrix:** Draw $W_K \in \mathbb{R}^{m \times d_k}$ with i.i.d. entries $(W_K)_{jl} \sim \text{Bernoulli}(p)$.
4:     For each node $v_j$, the key is $\mathbf{k}_j = \mathbf{e}_j W_K$.
5: **Construct Query Matrix:** For each node $v_i$, set its query $\mathbf{q}_i = \mathbf{k}_{\pi(i)}$.
6:     This is equivalent to setting the $i$-th row of $W_Q$ to be the $\pi(i)$-th row of $W_K$.
7: **Set Threshold:** $\tau = \frac{p+p^2}{2} d_k$.

*Hence a single attention head correctly identifies all edges of $G$.*

*Proof.* For $j = \pi(i)$,

$$S_{i,\pi(i)} = \mathbf{k}_{\pi(i)} \cdot \mathbf{k}_{\pi(i)} \sim \text{Binomial}(d_k, p)$$

with mean $\mu_1 = d_k p$. For $j \neq \pi(i)$,

$$S_{ij} = \mathbf{k}_{\pi(i)} \cdot \mathbf{k}_j \sim \text{Binomial}(d_k, p^2)$$

with mean $\mu_2 = d_k p^2$. Take $\tau = \frac{\mu_1 + \mu_2}{2} = \frac{p+p^2}{2} d_k$.

For the (lower) tail at the true edge, the Chernoff bound gives

$$\Pr\big[S_{i,\pi(i)} \leq \tau\big] \leq \exp\Big(-\frac{\mu_1 \delta_1^2}{2}\Big)$$

where

$$\delta_1 = 1 - \frac{\tau}{\mu_1} = \frac{1-p}{2},$$

so $\Pr[S_{i,\pi(i)} \leq \tau] \leq \exp\big(-\frac{d_k p(1-p)^2}{8}\big)$. For the (upper) tail at non-edges, the Chernoff bound yields

$$\Pr\big[S_{ij} \geq \tau\big] \leq \exp\Big(-\frac{\mu_2 \delta_2^2}{2+\delta_2}\Big)$$

where

$$\delta_2 = \frac{\tau}{\mu_2} - 1 = \frac{1-p}{2p},$$

hence $\Pr[S_{ij} \geq \tau] \leq \exp\big(-\frac{d_k\, p(1-p)^2}{2(1+3p)}\big)$. A union bound over the $m$ target pairs and the $m(m-1)$ non-target pairs gives a total failure probability

$$m\, e^{-c_1 d_k} + m^2 e^{-c_2 d_k} \quad \text{with}$$
$$c_1 = \frac{p(1-p)^2}{8}, \ c_2 = \frac{p(1-p)^2}{2(1+3p)}.$$

Choosing $d_k = C \log m$ with $C > \max\{3/c_1, 2/c_2\}$ makes this at most $m^{-3}$, establishing the simultaneous separation $S_{i,\pi(i)} > \tau > S_{ij}$ and correctness. $\square$

Monotonicity under context restriction immediately now yields correctness for *every* context, independent of context length. Our lower bound from Section D for this case is $\Omega(\frac{m'}{d_{\text{model}}} \log m) = \Omega(\log m)$. Our construction achieves an upper bound of $d_k = O(\log m)$, demonstrating that the bound is tight for this class of problems. Also note that the threshold proof is identical if softmax is used; see Appendix F.

### E.2. Construction II: Permutations Under Compressive Embeddings

**Why Gaussian unit-norm (spherical) embeddings.** Random spherical codes let us obtain clean dot-product thresholds in our separation arguments; the general case of arbitrary embeddings is handled in Construction III (Sec. E.3). This cosine geometry is also standard and effective in practice: many systems explicitly constrain features to a hypersphere (e.g., NormFace and ArcFace in face recognition; Spherical Text Embedding in NLP; spherical objectives in metric learning). See also (Wang et al., 2017; Deng et al., 2019; Meng et al., 2019; Zhang et al., 2020).

We next prove the correctness of Construction II, which follows from Theorem 4.1, restated here for convenience.

**Theorem E.2** (Multi-head recognition under Gaussian unit-norm embeddings, max-over-heads). *Assume Gaussian unit-norm embeddings with $d_{\text{model}} \geq c_0 \log m$ for a sufficiently large absolute constant $c_0$. Let $h \leq 5m/d_{\text{model}} + 2 = O(m/d_{\text{model}})$ heads, per-head dimension $d_k = C \log m$ for a sufficiently large absolute constant $C$, and threshold $\tau = \frac{1}{2} d_k$. Construct $\{(W_Q^{(k)}, W_K^{(k)})\}_{k=1}^{h}$ as in Algorithm 1 and let $k(i)$ denote the unique head index such that $i \in V_{k(i)}$.*

*Then with probability at least $1 - m^{-3}$ over the draw of $(X, W_{\text{sig}})$, simultaneously for all $i \in V$:*

$$S_{i,\pi(i)}^{(k(i))} > \tau \qquad and \qquad \max_{k \in [h]} \max_{j \neq \pi(i)} S_{ij}^{(k)} < \tau.$$

*Consequently, $\forall j \neq \pi(i), S_{i,\pi(i)}^{\max} > \tau > S_{ij}^{\max}$, so max-over-heads recovers all edges, and the total key budget satisfies $D_K = h\, d_k = \Theta\left(\frac{m \log m}{d_{\text{model}}}\right)$.*

*Proof.* Let $\mathbf{u}_i := \mathbf{x}_i X^\top = \mathbf{e}_i (XX^\top)$ and $\boldsymbol{\delta}_i := \mathbf{u}_i - \mathbf{e}_i \in \mathbb{R}^m$. Thus $\mathbf{u}_i$ is the $i$-th row of the Gram matrix $G := XX^\top$; it satisfies $\mathbf{u}_i(i) = 1$ and, for $j \neq i$, $\mathbf{u}_i(j) = \langle \mathbf{x}_i, \mathbf{x}_j \rangle$. Let $F := \{v : \pi(v) = v\}$ be the fixed points of $\pi$; by the partition (Algorithm 1) every head's block $V_k$ is either fixed-point-free ($V_k \subseteq V \setminus F$, $|V_k| \leq d_{\text{model}}$) or a block of fixed points ($V_k = T_k \subseteq F$, $|V_k| \leq \lfloor d_{\text{model}}/4 \rfloor$). Fix a head $k$ and a source $i \in V_k$. Analogously to Construction I, write

$$\mathbf{q}_i^{(k)} = \mathbf{u}_i W_{Q,(k)}', \qquad \mathbf{k}_j^{(k)} = \mathbf{u}_j W_{K,(k)}'.$$

Using $\mathbf{u}_t = \mathbf{e}_t + \boldsymbol{\delta}_t$ and the definitions of $W_{Q,(k)}'$ and $W_{K,(k)}'$, decompose, for any $j$,

$$
\begin{aligned}
S_{ij}^{(k)} &= (\mathbf{u}_i W_{Q,(k)}') \cdot (\mathbf{u}_j W_{K,(k)}') \\
&= \underbrace{\mathbf{w}_{\pi(i)} \cdot \mathbf{w}_j \cdot \mathbb{I}(j \in T_k)}_{\text{Signal}} + \underbrace{\mathbf{w}_{\pi(i)} \cdot \sum_{t \in T_k} \delta_{j,t} \mathbf{w}_t}_{N_1} \\
&\quad + \underbrace{\Big( \sum_{s \in V_k} \delta_{i,s} \mathbf{w}_{\pi(s)} \Big) \cdot \mathbf{w}_j \cdot \mathbb{I}(j \in T_k)}_{N_2} \\
&\quad + \underbrace{\Big( \sum_{s \in V_k} \delta_{i,s} \mathbf{w}_{\pi(s)} \Big) \cdot \Big( \sum_{t \in T_k} \delta_{j,t} \mathbf{w}_t \Big)}_{N_3}.
\end{aligned}
$$

Signal here means the contribution that would remain under a perfect inverse (i.e., if $XX^\top = I$): $\mathbf{w}_{\pi(i)} \cdot \mathbf{w}_j \cdot \mathbb{I}(j \in T_k)$. The Noise terms $N_1, N_2, N_3$ arise solely from the leakage vectors $\boldsymbol{\delta}_i, \boldsymbol{\delta}_j$ due to approximate de-embedding. For $j \in T_k \setminus \{\pi(i)\}$ the cross-inner product $\mathbf{w}_{\pi(i)} \cdot \mathbf{w}_j$ is *not* counted as noise (it is intrinsic signature cross-correlation) and is bounded separately. To bound the Noise terms, we next quantify properties of the approximate inverse $XX^\top$ for unit-norm Gaussian rows.

**Lemma E.3** (Concentration of the approximate inverse). *Let $X$ be as above and $d_{\text{model}} \geq c_0 \log m$ for a sufficiently large constant $c_0$. With probability at least $1 - m^{-4}$, simultaneously for all $i \in [m]$ and heads $k \in [h]$:*

1. $\mathbf{u}_i(i) = 1$ *(deterministically).*

2. *(Leakage $L_2$-mass, size-proportional) For every $i$ and every $S \subseteq [m]$ with $|S| \leq d_{model}$,*

$$
\begin{aligned}
\|\boldsymbol{\delta}_{i,S}\|_2^2 &= \sum_{s \in S,\, s \neq i} \langle \mathbf{x}_i, \mathbf{x}_s \rangle^2 \\
&\leq \frac{|S \setminus \{i\}|}{d_{model}} + C_2 \Big( \frac{\sqrt{|S| \log m}}{d_{model}} + \frac{\log m}{d_{model}} \Big)
\end{aligned}
$$

*for an absolute constant $C_2$. In particular $\|\boldsymbol{\delta}_{i,S}\|_2^2 \leq 2$ for $|S| \leq d_{model}$, and $\|\boldsymbol{\delta}_{i,S}\|_2^2 \leq \frac{1}{4} + o(1)$ for $|S| \leq \lfloor d_{model}/4 \rfloor$.*

3. *(Cross-correlations, block-size sensitive) Let $F^* := T_k \cap F$. For all $i, j$,*

$$
\begin{aligned}
\Big| \sum_{a \in T_k} \delta_{i,\pi^{-1}(a)}\, \delta_{j,a} \Big| \\
\leq \mathbb{I}[i = j] \frac{|F^* \setminus \{i\}|}{d_{model}} \\
+ C_3 \Big( \frac{\sqrt{|T_k| \log m}}{d_{model}} + \frac{\log m}{d_{model}} \Big)
\end{aligned}
$$

*for an absolute constant $C_3$ (absorbing the $(1 + O(1/d_{model}))$ corrections and a single $O(1/d_{model})$ predecessor term). The diagonal term is present only when $T_k$ contains fixed points of $\pi$; for a fixed-point-free head $F^* = \varnothing$, and bounding $|T_k| \leq d_{model}$ recovers the coarse $C_3 \sqrt{\log m / d_{model}}$.*

*Proof.* For Item 2: each $\langle \mathbf{x}_i, \mathbf{x}_s \rangle^2$ $(s \neq i)$ is sub-exponential with $\psi_1$-norm $\Theta(1/d_{model})$ and mean $1/d_{model} + O(1/d_{model}^2)$, and $\{\langle \mathbf{x}_i, \mathbf{x}_s \rangle^2\}_{s \in S}$ are independent given $\mathbf{x}_i$. Bernstein over $|S|$ terms gives the displayed deviation $C_2(\sqrt{|S| \log m}/d_{model} + \log m/d_{model})$ around the mean $|S \setminus \{i\}|/d_{model}$, simultaneously over all $i$ and the $O(m/d_{model})$ blocks $S$ by a union bound.

For Item 3, write $Z_a := \delta_{i, \pi^{-1}(a)} \, \delta_{j,a}$; by $\delta_{\ell, \ell} = 0$ the product vanishes when $\pi^{-1}(a) = i$ or $a = j$. Split $\sum_a Z_a = \sum_a \mathbb{E}[Z_a] + \sum_a (Z_a - \mathbb{E}[Z_a])$. *Mean.* For $p \neq q$, $r \neq s$, $\mathbb{E}[\langle \mathbf{x}_p, \mathbf{x}_q \rangle \langle \mathbf{x}_r, \mathbf{x}_s \rangle] = 0$ unless $\{p, q\} = \{r, s\}$. Applied to $\{i, \pi^{-1}(a)\}, \{j, a\}$, the only nonzero means are (a) $i = j$ with $\pi^{-1}(a) = a$ (a fixed point $a \in F^* \setminus \{i\}$), contributing $1/d_{model} + O(1/d_{model}^2)$ each, and (b) the single index $a = i$ with $j = \pi^{-1}(i) \neq i$, of size $O(1/d_{model})$; hence $\sum_a \mathbb{E}[Z_a] = \mathbb{I}[i = j] \, |F^* \setminus \{i\}|/d_{model} \, (1 + O(1/d_{model})) + O(1/d_{model})$. *Centered part.* Each $Z_a - \mathbb{E}[Z_a]$ is sub-exponential $(\psi_1 = \Theta(1/d_{model}))$. Since $Z_a$ depends on $(\mathbf{x}_a, \mathbf{x}_{\pi^{-1}(a)})$, the dependency graph on $T_k$ with $a \sim a' \iff \{a, \pi^{-1}(a)\} \cap \{a', \pi^{-1}(a')\} \neq \varnothing$ has max degree $\leq 2$ (as $a' \in \{\pi(a), \pi^{-1}(a)\}$), hence is 3-colorable; within a color class the $\{a, \pi^{-1}(a)\}$ are disjoint, so conditional on $(\mathbf{x}_i, \mathbf{x}_j)$ the class is independent. The $\leq 4$ boundary indices $a \in \{i, j, \pi(i), \pi(j)\} \cap T_k$ are handled directly: $a = j$ and $a = \pi(i)$ give $Z_a = 0$ (by $\delta_{\ell, \ell} = 0$); $a = i$ and $a = \pi(j)$ give $|Z_a| = O(\log m/d_{model})$ (one factor a fixed inner product of size $O(\sqrt{\log m/d_{model}})$, the other generic of the same size). For the generic indices, Bernstein per color class (conditional on $(\mathbf{x}_i, \mathbf{x}_j)$) and $|C_\ell| \leq |T_k|$ give $O(\sqrt{|T_k| \log m}/d_{model} + \log m/d_{model})$; the conditional-vs-unconditional mean gap is 0 for non-fixed $a$, vanishes for $i = j$, and for $i \neq j$ is $\leq (|T_k|/d_{model}) O(\sqrt{\log m/d_{model}}) = O(\sqrt{|T_k| \log m}/d_{model})$. A union bound over $(i, j)$ and combining parts gives the bound, with $C_3$ absorbing constants. Item 1 is immediate from unit-norm rows. $\square$

**Signal.** If $j = \pi(i)$, then Signal $= \|\mathbf{w}_{\pi(i)}\|_2^2 = d_k$ (exactly). If $j \in T_k$ and $j \neq \pi(i)$, then Signal $= \mathbf{w}_{\pi(i)} \cdot \mathbf{w}_j$ is a sum of $d_k$ i.i.d. Rademacher variables and thus sub-Gaussian with mean 0 and variance $d_k$. By a union bound over all $(i, j, k)$, with probability at least $1 - m^{-5}$,

$$|\text{Signal}| \leq C_\star \sqrt{d_k \log m} \quad \text{for all } (i, j \in T_k \setminus \{\pi(i)\}, k),$$

for an absolute constant $C_\star$.

**Noise.** Condition on $X$ and apply Lemma E.3. For $N_1$,

$$N_1 = \sum_{r=1}^{d_k} \Big( \sum_{t \in T_k} \delta_{j,t} \, w_t[r] \Big) w_{\pi(i)}[r]$$

is a sum of $d_k$ i.i.d. mean-zero sub-Gaussian variables with variance proxy $\|\boldsymbol{\delta}_{j, T_k}\|_2^2 \leq C_2$. Hence, by Bernstein/Hoeffding and a union bound over $(i, j, k)$,

$$|N_1| \leq C_4 \sqrt{C_2 \, d_k \, \log m}$$

holds w.h.p. for an absolute constant $C_4$. The same bound holds for $N_2$ with $\|\boldsymbol{\delta}_{j, V_k}\|_2^2 \leq C_2$.

For $N_3$, write for each column $r$,

$$X_r := \sum_{s \in V_k} \delta_{i,s} \, w_{\pi(s)}[r], \qquad Y_r := \sum_{t \in T_k} \delta_{j,t} \, w_t[r].$$

Then $N_3 = \sum_{r=1}^{d_k} X_r Y_r$. Conditional on $X$, $\{(X_r, Y_r)\}_{r=1}^{d_k}$ are i.i.d.; each $X_r$ and $Y_r$ is mean-zero sub-Gaussian with parameters $\lesssim \|\boldsymbol{\delta}_{i, V_k}\|_2 \leq \sqrt{C_2}$ and $\lesssim \|\boldsymbol{\delta}_{j, T_k}\|_2 \leq \sqrt{C_2}$, respectively. Thus $X_r Y_r$ is mean $\langle \boldsymbol{\delta}_{i, \pi^{-1}(T_k)}, \boldsymbol{\delta}_{j, T_k} \rangle$ and sub-exponential with $\psi_1$-norm $\lesssim C_2$. Consequently,

$$\mathbb{E}[N_3 \mid X] = d_k \langle \boldsymbol{\delta}_{i, \pi^{-1}(T_k)}, \boldsymbol{\delta}_{j, T_k} \rangle,$$

and, by Bernstein plus a union bound,

$$\big| N_3 - \mathbb{E}[N_3 \mid X] \big| \leq C_5 \, C_2 \sqrt{d_k \log m}$$

w.h.p. for an absolute constant $C_5$. Using Lemma E.3(3) and $|T_k| \leq d_{model}$,

$$\big| \mathbb{E}[N_3 \mid X] \big| \leq \mathbb{I}[i = j] \frac{d_k \, |F^* \setminus \{i\}|}{d_{model}} + d_k \, C_3 \sqrt{\frac{\log m}{d_{model}}},$$

where $F^* = T_k \cap F$; on a fixed-point-free head $F^* = \varnothing$ and the diagonal term vanishes.

**Separation.** Fix $C, c_0$ large enough that every signature-fluctuation term $O(\sqrt{d_k \log m})$ and every de-embedding term $O(d_k \sqrt{\log m/d_{model}})$ is at most $d_k/16$ (feasible as $d_k = C \log m$, $d_{model} \geq c_0 \log m$); any sum of a bounded number of them is then $\leq d_k/4$.

*Target edges.* If $i \in V \setminus F$, let $k$ be its head $(F^* = \varnothing)$: Signal $= d_k$ exactly, $|\mathbb{E}[N_3 \mid X]| \leq C_3 d_k \sqrt{\log m/d_{model}}$ by item 3, and $|N_1|, |N_2|, |N_3 - \mathbb{E}[N_3 \mid X]| = O(\sqrt{d_k \log m})$, so $S_{i, \pi(i)}^{(k)} \geq \frac{3}{4} d_k > \tau$. If $i \in F$, its head has $\pi(i) = i \in V_k = T_k$, Signal $= \|\mathbf{w}_i\|_2^2 = d_k$, and the diagonal $N_3$ mean $d_k \|\boldsymbol{\delta}_{i, V_k}\|_2^2 \geq 0$ only strengthens the edge; again $S_{ii}^{(k)} \geq \frac{3}{4} d_k > \tau$.

*Non-edges $j \neq \pi(i)$.* On a fixed-point-free head $(F^* = \varnothing, |T_k| \leq d_{model})$, Signal $\in \{0\} \cup \{O(\sqrt{d_k \log m})\}$ and item

3 gives $|\mathbb{E}[N_3 \mid X]| \leq C_3 d_k \sqrt{\log m / d_{\text{model}}}$, so $|S_{ij}^{(k)}| < \frac{1}{4} d_k < \tau$. The remaining case is a fixed-point head $V_k = T_k \subseteq F$, $|V_k| \leq \lfloor d_{\text{model}}/4 \rfloor$, evaluated at a source $i \notin V_k$ on the diagonal $j = i$: then (using $\pi(s) = s$ for $s \in V_k \subseteq F$) $\mathbf{q}_i^{(k)} = \sum_{s \in V_k} \delta_{i,s} \mathbf{w}_s = \mathbf{k}_i^{(k)}$, so

$$S_{ii}^{(k)} = \Big\| \sum_{s \in V_k} \delta_{i,s} \mathbf{w}_s \Big\|_2^2,$$

$$\mathbb{E}\big[ S_{ii}^{(k)} \mid X \big] = d_k \|\boldsymbol{\delta}_{i,V_k}\|_2^2 \leq d_k \big( \tfrac{1}{4} + o(1) \big) \leq \tfrac{5}{16} d_k$$

by item 2 at block size $\leq d_{\text{model}}/4$, with fluctuation $O(\sqrt{d_k \log m})$; hence $S_{ii}^{(k)} \leq \frac{5}{16} d_k + \frac{1}{16} d_k < \frac{1}{2} d_k = \tau$. For $j \neq i$ on a fixed-point head the $\mathbb{I}[i = j]$ term in item 3 vanishes and the bound reduces to the fixed-point-free estimate, again $< \frac{1}{4} d_k$. A union bound over all $(i, j, k)$ completes the proof. $\square$

Thus, with Gaussian unit-norm embeddings and Rademacher signatures, our construction recognizes the entire graph using a total key dimension

$$D_K = h \cdot d_k = O\left( \frac{m \log m}{d_{\text{model}}} \right),$$

This bound is asymptotically optimal, matching our lower bound within a constant factor, since $m' = m$ in the case of permutation graphs. In the proof of Theorem 4.1, the non-edge bounds hold uniformly over all $(i, j, k)$ (we union bound over $(i, j, k)$), so for any $j \neq \pi(i)$ we have $S_{ij}^{(k)} < \tau$ for *all* heads $k$ and hence $S_{ij}^{\max} < \tau$, while the target edge satisfies $S_{i,\pi(i)}^{\max} > \tau$. Monotonicity under context restriction then yields correctness on $E|_{\mathcal{C}}$ for every context $\mathcal{C}$.

### E.3. Construction III: More General Embeddings

The analysis of Construction II (Gaussian unit–norm) ultimately used only two facts about the Gram matrix $XX^\top$: (i) diagonals concentrate around a common scale, and (ii) for any *small* subset of indices the off–diagonal leakage has bounded $\ell_2$ mass, with a mild control on a corresponding cross–leakage term. We package these into a reusable, block–level notion that subsumes the usual pairwise incoherence and is tight enough to cover sparse/binary compressive embeddings.

**Definition E.4** (Restricted self-incoherence, relative to a block family). Fix a permutation $\pi$ and the two-block-size partition; let $\mathcal{B}$ be the family of source/target blocks $\{V_k, T_k = \pi(V_k)\}$ together with the fixed-point sub-blocks $\{F_{k'}\}$ — a fixed collection of $O(m/B)$ sets of size $\leq B$, determined by $\pi$ and the partition (not by $X$) — with induced twists $\sigma = \pi^{-1}|_{T_k}$ (and $\text{id}_{F_{k'}}$ on loop blocks). With $X_{\text{inv}} = \frac{1}{\mu} X^\top$, $\mathbf{u}_i = \mathbf{x}_i X_{\text{inv}}$, $\boldsymbol{\delta}_i = \mathbf{u}_i - \mathbf{e}_i$, say $X$ is $(\mu, \varepsilon_d, B; \rho, \gamma, \eta)$-*restricted self-incoherent relative to* $\mathcal{B}$ if:

1. **Diagonal stability:** $\mathbf{u}_i(i) \in [1 - \varepsilon_d, 1 + \varepsilon_d]$ for all $i$.

2. **Restricted leakage (size-proportional, with small-block term):** for every $i$ and every $S \in \mathcal{B}$, $\|\boldsymbol{\delta}_{i,S}\|_2^2 \leq \rho |S|/B + \eta$.

3. **Twisted restricted cross-leakage:** for every $i, j$ and every $S \in \mathcal{B}$ with induced twist $\sigma$, *provided $i \neq j$ or $\sigma$ has no fixed point in $S$*, $\big| \sum_{a \in S} \delta_{i,\sigma(a)} \delta_{j,a} \big| \leq \gamma$.

The conditions are imposed only on the $O(m)$ blocks of $\mathcal{B}$, which is all the construction evaluates; this is what the concrete embeddings below satisfy.

---

**Algorithm 4** Construction for Generalized Embeddings

1: **Input:** Embedding matrix $X \in \mathbb{R}^{m \times d_{\text{model}}}$; permutation graph $G = (V, E)$ with $\pi : V \to V$.

2: **Parameters:** Signature sparsity $p \in (0, 1/20]$; per–head width $d_k = C \log m$ for a sufficiently large absolute constant $C$.

3: **Random signatures:** Draw $W_{\text{sig}} \in \{0, 1\}^{m \times d_k}$ with i.i.d. Bernoulli$(p)$ entries; let $\mathbf{w}_j$ denote its $j$-th row.

4: **Set Threshold:** $\tau := \frac{p + p^2}{2} d_k$.

5: **Choose block size and partition:** Pick a block size $B$ (specified per embedding family below). Let $F = \{i : \pi(i) = i\}$; partition the non-fixed sources $V \setminus F$ into blocks of size $\leq B$ and the fixed points $F$ into blocks of size $\leq \lfloor B/4 \rfloor$, giving $h \leq 5m/B + 2$ blocks. For each head $k$, define its target set

$$T_k := \{\pi(s) : s \in V_k\}.$$

6: **Define one-hot–space templates (for each head $k$):**

$$W'_{Q,(k)}(i, :) = \begin{cases} \mathbf{w}_{\pi(i)} & i \in V_k \\ 0 & \text{else} \end{cases}$$

$$W'_{K,(k)}(j, :) = \begin{cases} \mathbf{w}_j & j \in T_k \\ 0 & \text{else} \end{cases}.$$

7: **Realize parameters via approximate inverse:**

$$W_Q^{(k)} = X_{\text{inv}} W'_{Q,(k)}, \qquad W_K^{(k)} = X_{\text{inv}} W'_{K,(k)}.$$

---

**Theorem E.5** (Recognition under restricted self-incoherence). *Let $X$ be $(\mu, \varepsilon_d, B; \rho, \gamma, \eta)$-restricted self-incoherent relative to $\mathcal{B}$. Take $d_k = C \log m$ ($C$ a large absolute constant) and $\tau = \frac{1}{2} d_k$ (Rademacher signatures). There is a small absolute constant $c$ and absolute constants $C_1, C_2, C_3$ such that if*

$$C_1 \sqrt{\rho + \eta} + C_2 \gamma + C_3 \varepsilon_d < c,$$

*then with probability at least $1 - m^{-3}$ over $W_{\text{sig}}$ (conditionally on $X$, as (R1)–(R3) are deterministic in $X$), the two-block-size construction (block size $B$ for non-fixed-point sources, $\lfloor B/4 \rfloor$ for fixed points) recovers all edges of $\pi$, and $D_K = h\, d_k = \Theta((m \log m)/B)$.*

*Proof sketch.* The seven-case analysis of Theorem 4.1 carries over with item 2 of Lemma E.3 replaced by (R2) and item 3 by the twisted form (R3). On non-fixed-point heads the twist $\sigma = \pi^{-1}|_{T_k}$ has no fixed points, so (R3) bounds the $N_3$ mean by $d_k \gamma$ for all $(i, j)$. On a fixed-point head ($\sigma = \text{id}$), (R3) supplies the off-diagonal bound, while the diagonal $N_3$ mean is $d_k \|\boldsymbol{\delta}_{i,F_{k'}}\|_2^2 \le d_k(\rho/4 + \eta)$ by (R2) at block size $\lfloor B/4 \rfloor$. The deterministic $N_1, N_2$ leakage terms are single coordinates of size $d_k |\delta_{j,\pi(i)}| \le d_k \sqrt{\rho + \eta}$ (a coordinate is bounded by the block norm), and the signature fluctuations are $O(\sqrt{d_k \log m})$. The hypothesis $C_1 \sqrt{\rho + \eta} + C_2 \gamma + C_3 \varepsilon_d < c$ keeps every non-edge contribution below $\frac{1}{4} d_k$ and the edge above $\frac{3}{4} d_k$, so $\tau = \frac{1}{2} d_k$ separates. With $h \le 5m/B + 2$, $D_K = \Theta((m \log m)/B)$. $\square$

**How to pick $B$.** The theorem asks only that the margin condition $C_1 \sqrt{\rho + \eta} + C_2 \gamma + C_3 \varepsilon_d < c$ hold at the chosen block size $B$. Different embedding families admit different $(\rho, \gamma, \eta)$–vs–$B$ trade–offs; plugging the corresponding $B$ into $D_K = \Theta((m/B) \log m)$ yields the budget.

COROLLARIES FOR COMMON EMBEDDING MODELS

**Corollary E.6** (Gaussian unit-norm). *For GUN embeddings, verify (R1)–(R3) per block over the $O(m/B)$ sets $S \in \mathcal{B}$ and union-bound. Lemma E.3 gives $\|\boldsymbol{\delta}_{i,S}\|_2^2 \le \rho|S|/B + \eta$ with $\rho \lesssim B/d_{model}$ and $\eta = O(\log m/d_{model})$ (splitting on $|S| \gtrless \log m$); the multiset argument gives (R3) with $\gamma \lesssim \sqrt{\log m/d_{model}}$ over the admissible cases. Since $\rho \asymp B/d_{model}$ is controlled by the ratio $B/d_{model}$ (not by $d_{model}$), take $B = c_B d_{model}$ with $c_B$ a small absolute constant, so $\sqrt{\rho + \eta} \lesssim \sqrt{c_B} + o(1)$ meets the margin while $B = \Theta(d_{model})$. Then $D_K = \Theta((m \log m)/d_{model})$. The coordinate leakage $|\langle \mathbf{x}_j, \mathbf{x}_{\pi(i)} \rangle|$ is in fact $O(\sqrt{\log m/d_{model}})$, sharper than the abstract $\sqrt{\rho + \eta}$, so the $N_1, N_2$ terms are controlled directly for $c_0$ large.*

**Corollary E.7** (Random binary compressive embeddings, centered). *Let $X$ be centered/signed sparse: each coordinate is 0 w.p. $1 - p_B$ and $\pm 1/\sqrt{d_{model} p_B}$ otherwise, $p_B = \Theta(\log m/d_{model})$, so $\mathbb{E}\langle \mathbf{x}_i, \mathbf{x}_s \rangle = 0$ for $i \ne s$ (centering makes the multiset mean-zero argument applicable; nonnegative binary features have positive-mean off-diagonals and are not covered). Per block: (R2) holds with $\rho \lesssim B p_B$, $\eta = O(p_B)$; (R3) with $\gamma \lesssim B p_B^2$. Taking $B = c_B d_{model}/\log m$ gives $\rho = \Theta(c_B)$, $\gamma = O(\log m/d_{model})$, and $D_K = \Theta((m \log^2 m)/d_{model})$. The coordinate leakage*

$|\langle \mathbf{x}_j, \mathbf{x}_{\pi(i)} \rangle|$ *is $O(p_B)$, sharper than the abstract $\sqrt{\rho + \eta}$, so the $N_1, N_2$ terms are controlled directly for $c_0$ large.*

**Signature family.** We stated the construction with Bernoulli($p$) signatures because the thresholding analysis naturally separates $j = \pi(i)$ from $j \in T_k \setminus \{\pi(i)\}$ at means $p d_k$ vs. $p^2 d_k$. One can equivalently use Rademacher $\{\pm 1\}$ signatures with threshold $\tau = \frac{1}{2} d_k$; all bounds above translate verbatim with the same $B$ and $d_k = \Theta(\log m)$.

**Takeaways.** Definition E.4 abstracts the only geometric inputs needed by the attention construction. Plugging in model–specific $(\rho, \gamma, \eta)$–vs–$B$ trade–offs yields the head count $h = \Theta(m/B)$ and total key budget $D_K = \Theta((m/B) \log m)$. For Gaussian unit–norm embeddings one recovers the $D_K = \Theta(m \log m/d_{model})$ guarantee; for sparse random binary compressive embeddings one obtains $D_K = \Theta(m \log^2 m/d_{model})$.

### E.4. Construction IV: General Graphs

We now extend the permutation constructions to general directed graphs $G = (V, E)$ with $|V| = m$ vertices and $|E| = m'$ edges. In this case, our information theoretic lower bound on total key dimension is $D_K = \Omega\left(\frac{m'}{d_{model}} \log(m^2/m')\right)$. We here provide a general upper bound for any graph, and show that for graphs that have a mild skew condition (the maximum degree is not too much larger than the average degree), it asymptotically matches this lower bound for all but the densest graphs (which match within a log factor). As before we use max aggregation over heads with a global scalar threshold $\tau$, and we work under the *Gaussian unit-norm embedding* model from Construction II: the row vectors of $X \in \mathbb{R}^{m \times d_{model}}$ are i.i.d. isotropic Gaussian followed by $L_2$-normalization. All probabilities are over the draw of $X$ and of the (head-shared) random signature matrix.

**Packing edges into matchings.** The analysis in Theorem 4.1 operates on blocks in which each source has exactly one outgoing edge *and* targets are distinct within the block. Equivalently, each head should see a *matching* (a partial permutation) between a set of sources and a set of targets.

We will use a simple decomposition of the edge set into matchings of size $d_{model}$ which will be our block size. Write $d_{out}(i)$ and $d_{in}(i)$ for the out-/in-degree of $v_i$. Let $\Delta_{out} := \max_i d_{out}(i)$ and $\Delta_{in} := \max_i d_{in}(i)$ denote the maximum out- and in-degrees, and write $\Delta := \max\{\Delta_{out}, \Delta_{in}\}$.

**Lemma E.8** (Coloring-and-batching decomposition). *Let $G = (V, E)$ be any directed graph on $m$ vertices and $m'$ edges, and let $H := \left\lceil \frac{m'}{d_{model}} \right\rceil + \Delta$. Then there exists a partition of $E$ into $H$ disjoint sets $M_1, \ldots, M_H$ such that*

*for every $k$: (i) $M_k$ is a matching (no two edges in $M_k$ share a source or a target); (ii) $|M_k| \leq d_{model}$.*

*Proof.* Identify $G$ with its bipartite incidence graph $\mathcal{B} = (V_L \cup V_R, E)$ where each directed edge $(i, j)$ becomes an undirected edge between $i \in V_L$ and $j \in V_R$. Then $\Delta(\mathcal{B}) = \Delta$. By Kőnig's line-coloring theorem, $E = F_1 \cup \cdots \cup F_\Delta$ with each $F_c$ a matching. Split each $F_c$ into blocks of size at most $d_{\text{model}}$; since $\sum_{c=1}^{\Delta} \lceil |F_c|/d_{\text{model}} \rceil \leq \left\lceil \frac{\sum_c |F_c|}{d_{\text{model}}} \right\rceil + \Delta = \left\lceil \frac{m'}{d_{\text{model}}} \right\rceil + \Delta = H$, we obtain $H$ matchings $M_k$ each of size at most $d_{\text{model}}$. $\qquad\square$

Thus, after packing via Lemma E.8 head $k$ will operate on the matching $M_k$. Let $V_k \subseteq V$ and $T_k \subseteq V$ denote the sources and targets incident to $M_k$ and write $\pi_k : V_k \to T_k$ for the bijection defined by $M_k$.

We pre-separate self-loops. Let $L = \{v : (v, v) \in E\}$ and split $E = E_{\text{loop}} \cup E_{\text{non-loop}}$ with $E_{\text{loop}} = \{(v, v) : v \in L\}$. Batch $E_{\text{non-loop}}$ into matchings of size $\leq d_{\text{model}}$ ($H_1 \leq \lceil (m' - |L|)/d_{\text{model}} \rceil + \Delta$), and (since $E_{\text{loop}}$ is itself a matching) chunk it into "loop heads" of size $\leq \lfloor d_{\text{model}}/4 \rfloor$ ($H_2 \leq 5|L|/d_{\text{model}} + 1$, using $\lfloor d_{\text{model}}/4 \rfloor \geq d_{\text{model}}/5$). A loop head has $V_k = T_k = F_k \subseteq L$, $\pi_k = \text{id}$, exactly the fixed-point-head structure. With $|L| \leq m'$ the total head count is $H \leq \lceil (m' - |L|)/d_{\text{model}} \rceil + \Delta + \lceil |L|/\lfloor d_{\text{model}}/4 \rfloor \rceil = O(m'/d_{\text{model}} + |L|/d_{\text{model}} + \Delta)$.

**Construction.** We reuse the compressive permutation machinery head-by-head.

**Lemma E.9** (Cross-correlation for partial injections). *Let $M_k$ be a partial matching with sources $V_k$, targets $T_k = \pi_k(V_k)$, $|T_k| \leq d_{model}$, and let $L$ be the self-loop set. Then w.p. $\geq 1 - m^{-5}$, simultaneously for all $i, j$,*

$$\left| \sum_{a \in T_k} \delta_{i, \pi_k^{-1}(a)} \delta_{j,a} \right| \leq \mathbb{I}[i = j] \frac{|T_k \cap L|}{d_{model}}$$
$$+ C_3 \left( \frac{\sqrt{|T_k| \log m}}{d_{model}} + \frac{\log m}{d_{model}} \right).$$

*Proof.* For each target $a \in T_k$, $\pi_k^{-1}(a)$ is well-defined (one in-edge per target in a matching), so $Z_a = \delta_{i, \pi_k^{-1}(a)} \delta_{j,a}$ depends only on the pair $P_a = \{a, \pi_k^{-1}(a)\}$ ($\pi_k(a)$ need not be defined). Put $a \sim a'$ iff $P_a \cap P_{a'} \neq \varnothing$; as $a \mapsto \pi_k^{-1}(a)$ is injective, each index lies in $\leq 2$ pairs, so the dependency graph has max degree $\leq 2$ and is 3-colorable. The proof of item 3 of Lemma E.3 then applies verbatim with this graph, the fixed points of $\pi_k$ in $T_k$ being exactly $T_k \cap L$. $\qquad\square$

**Theorem E.10** (General graphs). *Assume $d_{model} \geq c_0 \log m$ for a sufficiently large constant $c_0$. For any fixed directed graph $G$ and fixed batching, with the construction*

**Algorithm 5** Construction for General Graphs

1: **Input:** Directed graph $G = (V, E)$ with $|V| = m$, $|E| = m'$; embedding matrix $X \in \mathbb{R}^{m \times d_{\text{model}}}$ with Gaussian unit-norm rows.

2: **Parameters:** Number of heads $h = H = H_1 + H_2 = O(m'/d_{\text{model}} + |L|/d_{\text{model}} + \Delta)$; per-head key/query dimension $d_k = C \log m$ for a sufficiently large absolute constant $C$. Non-loop heads use block size $d_{\text{model}}$, loop heads block size $\lfloor d_{\text{model}}/4 \rfloor$.

3: **Pack edges into matchings:** Pre-separate self-loops $E_{\text{loop}} = \{(v, v) : v \in L\}$ from $E_{\text{non-loop}}$. Decompose $E_{\text{non-loop}}$ into disjoint matchings of size $\leq d_{\text{model}}$ via Lemma E.8, and chunk $E_{\text{loop}}$ into loop heads of size $\leq \lfloor d_{\text{model}}/4 \rfloor$. For each head $k$, let $V_k$ and $T_k$ be the sources and targets incident to $M_k$ and write $\pi_k : V_k \to T_k$ for the associated bijection (loop heads have $V_k = T_k$, $\pi_k = \text{id}$).

4: **Random signatures:** Draw a shared Rademacher matrix $W_{\text{sig}} \in \{\pm 1\}^{m \times d_k}$ with i.i.d. entries; let $\mathbf{w}_j$ denote its $j$-th row.

5: **Per-head "ideal" matrices:**

$$\left( W'_{Q,(k)} \right)_{i,\cdot} := \begin{cases} \mathbf{w}_{\pi_k(i)} & i \in V_k \\ \mathbf{0} & \text{otherwise} \end{cases},$$

$$\left( W'_{K,(k)} \right)_{j,\cdot} := \begin{cases} \mathbf{w}_j & j \in T_k \\ \mathbf{0} & \text{otherwise} \end{cases},$$

where $\mathbf{w}_j$ is the $j$-th row of $W_{\text{sig}}$.

6: **Final projections (approximate de-embedding):** As in Construction II, use the approximate inverse $X^\top$:

$$W_Q^{(k)} = X^\top W'_{Q,(k)}, \qquad W_K^{(k)} = X^\top W'_{K,(k)}.$$

7: **Set Threshold:** $\tau = \frac{1}{2} d_k$.

---

*above, there is a universal $C$ such that, with probability at least $1 - m^{-3}$ over the draw of $(X, W_{\text{sig}})$ (pointwise in $(G, \text{batching})$), simultaneously for all ordered pairs $(i, j)$,*

$$S_{ij}^{\max} = \max_{1 \leq k \leq H} S_{ij}^{(k)} \begin{cases} > \tau & \text{if } (i, j) \in E, \\ < \tau & \text{if } (i, j) \notin E. \end{cases}$$

*Consequently, $D_K = O\left( \frac{m' \log m}{d_{model}} + \frac{|L| \log m}{d_{model}} + \Delta \log m \right)$, which under bounded skew $\Delta \leq m'/d_{model}$ (and $|L| \leq m'$) simplifies to $O(m' \log m/d_{model})$.*

*Proof sketch.* By Lemma E.8 and the loop pre-separation, each head $k$ sees a partial injection $\pi_k : V_k \to T_k$ with $|T_k| \leq d_{\text{model}}$. Within head $k$, the score decomposition is that of Theorem 4.1, and the cross-correlation bound extends to the partial injection $\pi_k$ (Lemma E.9);

non-loop heads have no fixed points so the diagonal term vanishes, and loop heads are the fixed-point (diagonal) case of Theorem 4.1. For $(i, j) = (i, \pi_k(i))$ the Signal term equals $d_k$ and the three Noise terms $(N_1, N_2, N_3)$ are controlled using Lemma E.3 and Lemma E.9, since all leakage sets ($V_k \setminus \{i\}$ and $T_k$) have size $\leq d_{\text{model}}$. For $(i, j) \neq (i, \pi_k(i))$, Signal is a sum of i.i.d. Rademachers with variance $d_k$, while the same leakage bounds control $N_1, N_2, N_3$. Choosing $C$ and $c_0$ as in Theorem 4.1 yields, within each head, $S^{(k)}_{i, \pi_k(i)} > \tau$ and $|S^{(k)}_{ij}| < \tau$ for all $j \neq \pi_k(i)$ simultaneously with probability $1 - m^{-4}$.

A union bound over all heads and all pairs in those heads costs only a $\log$ factor absorbed by $d_k = C \log m$: using $|M_k| \leq d_{\text{model}}$ and $\sum_k |M_k| = m'$, we have $\sum_k |M_k|^2 \leq d_{\text{model}} \sum_k |M_k| = d_{\text{model}} m' = O(m' d_{\text{model}})$ events in total. Finally, max pooling across heads preserves separation (non-edges are below $\tau$ *in every head*, and each true edge belongs to exactly one $M_k$), and monotonicity under context restriction yields context-robustness for arbitrary subsets $\mathcal{C} \subseteq V$. $\qquad\square$

**Degree skew and tightness.** Let $d_{\text{avg}} = \frac{m'}{m}$. Define the *skew factor* to be $\Delta / d_{\text{avg}}$ and consider the condition

$$\frac{\Delta}{d_{\text{avg}}} \leq \frac{m}{d_{\text{model}}}. \qquad (26)$$

In other words, the ratio of the maximum degree to the average degree is no larger than the compression of the embedding, or equivalently, $\Delta \leq \frac{m'}{d_{\text{model}}}$. This condition automatically holds for all $d$-regular graphs (since $\Delta = d_{\text{avg}}$).

**Corollary E.11** (Bounded Skew). *Assume $d_{model} \geq c_0 \log m$ and* (26). *Then the construction with $h_0 = \lceil m'/d_{model} \rceil$ heads and $d_k = C \log m$ achieves the same separation guarantee as Theorem E.10, and $D_K = \Theta\big(\frac{m' \log m'}{d_{model}}\big)$.*

This is immediate from Theorem E.10 and asymptotically matches the lower bound from Section D for this class of graphs, provided $m' = O(m^{2-\epsilon})$ for some positive constant $\epsilon$.

## F. Additional Justification: Computational Footprint of the Model (Section 3)

While it is natural to consider the number of heads ($h$) and the per-head key/query dimension ($d_k$) as two separate resources, we argue that the most relevant complexity measure is their product

$$D_K := hd_k,$$

since standard implementations perform multi-head attention using batched dense linear algebra whose leading costs depend on $D_K$.

Concretely, let $\mathbf{X} \in \mathbb{R}^{\ell \times d_{\text{model}}}$ stack the context embeddings. With

$$W_Q^{\text{cat}} = [W_Q^{(1)} | \cdots | W_Q^{(h)}] \in \mathbb{R}^{d_{\text{model}} \times (hd_k)}$$

and

$$W_K^{\text{cat}} = [W_K^{(1)} | \cdots | W_K^{(h)}] \in \mathbb{R}^{d_{\text{model}} \times (hd_k)}$$

formed by concatenating the head weights, the total queries/keys are computed as

$$\mathbf{Q}_{\text{total}} = \mathbf{X} W_Q^{\text{cat}} \qquad \text{and} \qquad \mathbf{K}_{\text{total}} = \mathbf{X} W_K^{\text{cat}}.$$

Thus, both flops and parameter/memory cost for forming $\mathbf{Q}_{\text{total}}$ and $\mathbf{K}_{\text{total}}$ scale as

$$O(\ell \, d_{\text{model}} \, hd_k) = O(\ell \, d_{\text{model}} \, D_K)$$

and

$$O(d_{\text{model}} \, hd_k) = O(d_{\text{model}} \, D_K),$$

respectively, motivating $D_K$ as the appropriate budget.

This same dependence on $D_K$ persists in the subsequent computation of per-head logits. Reshape $\mathbf{Q}_{\text{total}}, \mathbf{K}_{\text{total}}$ into $\mathbf{Q} \in \mathbb{R}^{h \times \ell \times d_k}$ and $\mathbf{K} \in \mathbb{R}^{h \times \ell \times d_k}$, and form per-head score matrices

$$\mathbf{S}^{(t)} = \mathbf{Q}^{(t)} \big(\mathbf{K}^{(t)}\big)^\top \in \mathbb{R}^{\ell \times \ell}, \qquad t \in [h],$$

(with the usual $d_k^{-1/2}$ scaling in the softmax variant, which does not change asymptotic cost). The total work to compute all $\{\mathbf{S}^{(t)}\}_{t=1}^h$ scales as

$$O(h \, \ell^2 \, d_k) = O(\ell^2 \, D_K).$$

While sub-cubic matrix multiplication algorithms could theoretically make one large head asymptotically faster than several smaller ones, this effect is absent in practice. The dominant kernels here are highly optimized batched matrix multiplications (and, when applicable, softmax/normalization), and their observed throughput typically tracks the total matrix sizes involved. Consequently, at fixed $D_K = hd_k$, varying the split between $h$ and $d_k$ has little effect on the leading computational footprint of the $QK$ channel.

## G. Related Work

We survey work most relevant to our capacity-centric view of self-attention and position our **Relational Graph Recognition** (RGR) results in that landscape. The central distinction we draw is between *what* attention can compute in principle (expressivity), *how* architectural resources govern this power (capacity), and *which* parts of the Transformer carry the binding/addressing load (keys/queries vs. other channels).

**Memorization Capacity and Parameter–Dependent Bounds**

A growing body of work quantifies how many input–label associations Transformers—and, more narrowly, the attention mechanism—can memorize. As described in Section 2, the bounds from the memorization setting do not directly imply bounds on RGR, nor the other way around. For the attention module itself, (Mahdavi et al., 2024) prove that a single MHA layer with $h$ heads can memorize $\Omega(h \min\{\ell, d_k\})$ examples under a linear-independence assumption on the inputs, highlighting linear scaling in $h$ and the role of the per-head key/query width $d_k$. Complementary analyses bound attention's memory depth and clarify depth–capacity trade-offs (Madden et al., 2025). Moving to full Transformers, constructive results show that (under token-wise $(r, \delta)$-separated inputs) a stack of $2\ell$ self-attention layers suffices to memorize $N$ sequences with $\tilde{O}(\ell + \sqrt{\ell N})$ parameters (Kim et al., 2023); even a *single-layer, single-head* Transformer has nontrivial capacity under the same separatedness assumption, whereas replacing softmax by hardmax breaks memorization (Kajitsuka & Sato, 2024).

Beyond construction-style bounds, (Madden et al., 2025) give general upper and lower bounds for next-token prediction that scale as $\Theta(\omega N)$ in the presence of positional encodings and a vocabulary of size $\omega$, and (Chen & Zou, 2024) show that a *single-layer* Transformer can memorize when sequences are sufficiently zero-padded (though not in a parameter-optimal way). Classical results for ReLU networks connect parameter counts to memorization thresholds and VC-style capacity (Vardi et al., 2021). More closely related to our focus on resource efficiency, (Kajitsuka & Sato, 2025) establish nearly matching upper/lower bounds on the *minimal parameter count* needed for memorization in Transformers: $\tilde{O}(\sqrt{N})$ parameters are sufficient (and necessary up to logs) for next-token prediction, and $\tilde{O}(\sqrt{\ell N})$ for sequence-to-sequence, under token-wise separatedness; they further suggest that self-attention effectively *identifies* sequences while the feed-forward network can become the bottleneck when *associating* labels.

**Superposition, Constructive Designs, and Depth Separation**

A concurrent line of work analyzes how networks compute many features in *superposition*, with lower and upper bounds for narrow MLPs and constructive designs for multi-feature computation (Adler & Shavit, 2024; Adler et al., 2025; Hänni et al., 2024). Recent work further connects superposition to robust scaling behavior, providing additional evidence that feature packing can underwrite smooth scaling trends across model sizes (Liu et al., 2025). The capacity limits shown in this line are complementary

to ours in terms of architectures: their focus is on MLPs while ours is on attention and, in particular, on the key–query channel.

Foundational depth-separation and minimal-width universality results motivate the proof template we adopt—information-theoretic lower bounds matched by explicit constructions (Telgarsky, 2016; Hanin & Sellke, 2019; Kidger & Lyons, 2020; Cybenko, 1989). In attention, constructive correspondences also explain how multi-head architectures partition pattern spaces; e.g., with relative positions, $s^2$ heads can realize any $s \times s$ convolution (Cordonnier et al., 2020b). Our constructions similarly partition relational signal across heads to mitigate interference when $d_{\text{model}} \ll m$, explaining the empirical advantage of many small heads and clarifying when too-small $d_k$ triggers low-rank failure (Bhojanapalli et al., 2020).

**Dimension-, Rank-, and Resource–Driven Expressivity**

A growing body of theory isolates how *dimensional* resources govern attention's representational power. Universality guarantees establish that sufficiently resourced Transformers can approximate sequence-to-sequence functions (Yun et al., 2020a), while more refined results show task-dependent strengths and weaknesses (Sanford et al., 2023). Focusing on the attention map, (Likhosherstov et al., 2021) prove that with fixed error and sparsity, self-attention can approximate dynamic sparse right-stochastic matrices using only $O(\log \ell)$ hidden dimensions (for context length $\ell$), echoing the role of near-orthogonality we exploit in our constructions. Conversely, (Bhojanapalli et al., 2020) identify a per-head *low-rank bottleneck*: when $d_k < \ell$, a head cannot realize arbitrary $\ell \times \ell$ stochastic attention matrices. This clarifies a trade-off inside the total key/query budget $D_K = h \, d_k$: pushing $D_K$ into many tiny heads can induce head-wise rank limits.

Beyond these, several works develop structural and inductive-bias characterizations of self-attention. (Dong et al., 2021) show that *pure* attention without mixing loses rank doubly-exponentially with depth, explaining failure modes in deep attention stacks and underscoring the role of residual mixing. (Edelman et al., 2022) analyze *variable creation* and sparsity patterns induced by softmax, while (Sahiner et al., 2022) use convex duality to give optimization- and geometry-based interpretations of ViT attention. For sample complexity and approximation, (Li et al., 2023a) study learning and generalization of shallow ViTs; rates and approximation guarantees have been developed for Transformer encoders and sequence models (Gurevych et al., 2022; Takakura & Suzuki, 2023; Jiang & Li, 2024). Recent generalization bounds that are (largely) sequence-length independent sharpen this picture (Trauger & Tewari, 2024). Finally, theory has also pinpointed

sparsity-oriented inductive biases: Transformers provably learn *sparse token selection* that FCNs cannot (Wang et al., 2024), and exhibit a simplicity bias for sparse Boolean functions (Bhattamishra et al., 2023). Recent results show that when embeddings are learned, attention can provably focus on informative tokens under suitable conditions (Wu et al., 2025).

Empirical observations likewise single out the key/query channel as an operative budget. Our results formalize this perspective for a concrete relational family (RGR), deriving matching lower and constructive upper bounds in terms of $D_K$ and the number of relations.

### Formal-Language Limits, Compositionality, and Universality

Formal-language analyses delimit what fixed-size attention can recognize. Beyond general universality (Yun et al., 2020a), there are sharp impossibility results for periodic and hierarchical languages (Hahn, 2020; Bhattamishra et al., 2020; Yang et al., 2024). Recent work uses communication-complexity arguments to show single-layer self-attention struggles with *function composition* at fixed embedding/heads, e.g., "grandparent-of" requires resources that scale with domain size (Peng et al., 2024). Complementing these, (Luo et al., 2022) identify additional structural constraints on what Transformers can compute under realistic resource regimes. We view these results as orthogonal to RGR: they characterize *classes of computations*, whereas we fix a relational family and ask *how much key/query budget* is necessary and sufficient to represent its edges across arbitrary contexts.

### In-Context Learning and Algorithmic Views of Self-Attention

A complementary line of theory frames Transformers—and attention in particular—as executing *algorithms* over the context. (Li et al., 2023b) analyze generalization and implicit model selection in in-context learning; (von Oswald et al., 2022) give evidence that Transformers can implement gradient-descent-like updates in context; and (Garg et al., 2022) characterize which simple function classes are learnable in context. Recent theory also isolates the role of positional information in algorithmic execution: Back de Luca et al. (2025) study *positional attention*, where attention weights depend only on positional encodings, and show such models can retain strong expressivity for parallel algorithmic computation (with depth/sample-complexity tradeoffs). These works clarify how attention can implement algorithmic behaviors, while our RGR focus quantifies the *key–query capacity* required to retrieve relational edges reliably.

### Connectivity Patterns vs. Capacity in the Key/Query Channel

An alternative way to constrain attention is by controlling the connectivity pattern of the attention graph. Even $O(\ell)$-sparse patterns can be universal under appropriate designs (Yun et al., 2020b), and systematic pruning of dense patterns maps out cost–performance frontiers (Wang et al., 2022). Our analysis treats connectivity as *not* the bottleneck: given the ability to attend broadly, the limiting factor for RGR is how much relational information can be encoded and separated in keys/queries as $m$ and $m'$ grow.

### Emergent Sparsity and Transition Phenomena in Attention

Beyond hard-wiring sparsity, several recent works study how sparse (or structured) attention patterns *emerge* during training and how these transitions depend on data statistics. For example, (Zucchet et al., 2025) analyze when sparse attention arises and how repetition and distributional structure can accelerate its emergence. This line is complementary to our setting: we characterize *budget-driven* transitions in relational recovery as $D_K$ varies under controlled relational structure (RGR), whereas emergence work focuses on *training dynamics* and when attention becomes sparse or selective.

### Head Specialization, Pruning, and Information Bottlenecks

Mechanistic interpretability consistently finds that specific heads specialize to linguistic relations (Clark et al., 2019; Vig & Belinkov, 2019). At the same time, many trained heads can be pruned with small accuracy loss (Michel et al., 2019; Voita et al., 2019), indicating redundancy. Information-bottleneck analyses at the head/layer level quantify such redundancy and attribution in both language and vision models (Qian et al., 2025; Hong et al., 2025), and architectural proposals target representation bottlenecks (Gerasimov et al., 2025). More recently, causal attribution frameworks explicitly learn gates over heads to characterize facilitating vs. interfering roles and head interactions, reinforcing that head effects can be context-dependent and non-modular (Nam et al., 2025). Our results supply a capacity-theoretic backbone for these observations: for RGR, performance transitions are governed primarily by $D_K$; distributing $D_K$ across heads reduces interference between superposed relations, but overly small $d_k$ per head incurs rank limits—predicting both specialization and safe pruning regimes.

## Attention as Associative Memory vs. Relational Addressing

Modern Hopfield networks are equivalent, in a precise sense, to attention updates and can *store* exponentially many patterns in the associative dimension with single-step retrieval (Ramsauer et al., 2021). FFN layers in Transformers have also been interpreted as *key–value memories* (Geva et al., 2021). Our results complement this memory-centric view by isolating the *addressing* budget: how much key/query capacity is required to select the correct neighbors (edges) for arbitrary contexts. Together these views separate storage capacity from the cost of accurate retrieval/selection in the key–query channel.

A recent synthesis by (Zhong et al., 2025) frames both self-attention and FFNs as instances of kernelized associative memory, and proposes a retrieval signal-to-noise ratio (SNR) to quantify recall fidelity as the number of stored key–value pairs grows. Their analysis highlights how the exponential kernel underlying softmax attention can dramatically improve retrieval SNR relative to linear (and ReLU-like) kernels, and connects kernel choice to a precision–superposition tradeoff that influences polysemanticity. This viewpoint is complementary to ours: whereas their capacity proxy is recall accuracy for stored associations under distributional assumptions, our RGR formulation studies worst-case *relational addressing* across all contexts and all graphs in a family, yielding necessary and sufficient bounds in terms of the total key dimension $D_K$.

## Graph Transformers and Structural Encodings

Expressivity of graph Transformers is shaped by structural encodings and higher-order tokenization. SEG-WL analyses show that structural features (e.g., SPIS encodings) set the attainable expressivity ceiling and can be matched by simple Transformer variants (Zhu et al., 2023). Recent work has also aimed to bridge the theory–practice gap in graph Transformers by unifying architectural choices around attention and positional/structural encodings and validating them at scale (Stoll et al., 2025). Higher-order graph Transformers reach (or fall short of) $t$-WL power depending on whether explicit tuple indices and structural signals are provided (Zhou et al., 2024). In the vision setting, (Jelassi et al., 2022) prove that ViTs can learn spatial structure under appropriate conditions, resonating with our assumptions that near-orthogonal embeddings and structural signals determine how efficiently edges can be packed and recovered; given such signals, our $D_K$-based bounds become tight predictors of success.

**Linear / softmax-free attention.** A large line of work replaces or approximates softmax attention to reduce the quadratic cost in sequence length, often by expressing attention as a kernel feature-map product that can be evaluated associatively (so-called *linear attention*) (Katharopoulos et al., 2020). Follow-up methods improve the fidelity of softmax approximations via random-feature estimators (Choromanski et al., 2021; Peng et al., 2021) or exploit low-rank structure of the attention map (Wang et al., 2020), while other work proposes alternative normalizers / reweightings that aim to recover some of softmax's concentration behavior with linear-time computation (Qin et al., 2022). Recent work continues to probe the softmax/linear divide from several angles: (i) gating-based operators reintroduce nonlinearity and sparsity while mitigating attention pathologies (e.g., attention sinks), highlighting the functional role of competitive nonlinearities in routing (Qiu et al., 2025); (ii) feature-efficiency analyses ask how many linear-attention features are needed to distill or approximate softmax well under compute constraints (Nishikawa et al., 2025); (iii) simplified statistical models and mean-field analyses provide theoretical accounts of when softmax enjoys intrinsic advantages and how it behaves during learning (Duranthon et al., 2025; Dohmatob, 2025); and (iv) asymptotic results identify regimes where softmax attention approaches linear behavior, clarifying when the distinction can blur (Boursier & Boyer, 2025). From a unifying associative-memory perspective, (Zhong et al., 2025) analyze softmax vs. linear attention via retrieval-SNR and propose update-rule variants (e.g., delta-rule hybrids) that reintroduce selective overwrite/forgetting while retaining high-fidelity kernelized retrieval. Our focus is complementary: rather than proposing a new efficient operator, we use RGR to isolate how *the presence and type of nonlinearity in the QK routing computation* affects relational capacity and multi-head interference reduction, clarifying when purely linear score aggregation can lose the multi-head benefit and why even simple nonlinearities (e.g., max aggregation) can change this behavior.

**Summary.** Across expressivity, connectivity, memorization, superposition, interpretability, memory equivalence, and graph structure, prior work identifies the ingredients that make attention powerful and the constraints that limit it. We contribute a capacity-centric bridge: a concrete relational task (RGR) in which the *total key dimension $D_K$* is the critical budget, with lower and upper bounds tight up to logarithmic factors, a principled multi-head advantage, and empirical thresholds that align with constructive algorithms.

