# OpenReview forum: "A Capacity-Based Rationale for Multi-Head Attention"
_ICML.cc/2026/Conference — ICML 2026 regular_

### Official Review · Reviewer_gpPQ · 2026-03-11

**Soundness:** 3
**Presentation:** 2
**Significance:** 3
**Originality:** 3
**Overall Recommendation:** 3
**Confidence:** 1

**Summary:**

This paper provide a rigorous theoretical analysis of the capacity of the self-attention key-query channel by introducing Relational Graph Recognition, showing  recovering a graph with $m’$ relations in $d_\text{model}$-dimensional embeddings requires $D_K$ to grow as $\frac{m’}{d_\text{model}}$ up to logarithmic factors. Furthermore, based on this analysis, the author propose a novel capacity-based rationale for multi-head explanation: even in permutation graphs, where all queries attend to a single target, splitting a fixed $D_K$ budget into multiple heads increases capacity by reducing interference from embedding superposition. The theoretical predictions are verified by controlled experiments.

**Compliance With Llm Reviewing Policy:**

Affirmed.

**Final Justification:**

The rebuttal clarified part of the technical argument, but it did not resolve my concerns about the paper’s presentation and overall significance.  In my view, addressing these issues would require more substantial revision to the paper’s organization, rather than minor polishing. For this reason, I maintain my final recommendation of Weak Reject.

**Key Questions For Authors:**

1. The authors obtain matching information-theoretic bounds by explicit constructions and the interpretation of "multi-head advantage" is mainly based on the "max-over-heads" variant. For the standard softmax attention with a total bond dimension of $\frac{m}{d_{model}}(\log(m))^2$, does the "noise suppression" explanation still hold?

2. The mathematical symbols appearing in the abstract are not defined, and there are also undefined mathematical symbols in the main text.

**Limitations:**

Yes.

**Strengths And Weaknesses:**

Strengths

1. Propose the RGR framework, formalize the self-attention mechanism, and prove matching information-theoretic lower bounds and upper bounds via explicit constructions.

2. Provide a capacity-based explanation for multi-head attention, where multi-heads manage additive noise by partitioning the feature space.

3. Experiments confirm the theoretical predictions: a sharp F1 transition in a narrow $D_K$ window; splitting a fixed $D_K$ across more heads reduces interference from superposition; the optimal head count increases with compression $\frac{m}{d_model}$, while per-head width at the threshold is modest.

Weaknesses

1. The theoretical analysis is dependent on the construction of a random signature and a specific solution embedding step.

2. The structure of the paper is not very clear. For example, Problem Formulation, Task and Capacity question introduced in the introduction section are suitable to be put into Preliminary section.

---

> ### Author Rebuttal · Authors · 2026-03-28
>
> **On dependence on random signatures and the specific construction.**
>
> The random-signature construction is a proof technique — it provides the simplest way to achieve near-orthogonal separation across m relationships in low dimension. However, our **lower bounds** (Appendix D, Theorem D.2) are entirely construction-independent: they hold for *any* parameterization achieving constant-margin recovery, via a metric-entropy/covering-number argument. The fact that our constructive upper bounds match these lower bounds (up to constants for permutation graphs under max-over-heads; up to a log factor for softmax) means the scaling laws are properties of the *problem*, not of the particular solution strategy. Any alternative construction must use at least as much D_K.
>
> It is also worth noting that the general structure of our constructions — projecting into subspaces where specific token relationships produce high similarity while suppressing interference — mirrors what mechanistic interpretability studies find in trained attention heads (Kamath et al., 2025). This suggests the constructions, while idealized, capture the right qualitative mechanisms.
>
> **On paper structure.**
>
> We appreciate this feedback. In the camera-ready revision, we will move the Problem Formulation and Capacity Question into a dedicated Preliminaries section, and reorganize the introduction to focus on motivation and high-level results. We will also ensure all mathematical symbols in the abstract are either defined inline or accompanied by a forward reference.
>
> **On whether "noise suppression" holds for standard softmax attention.**
>
> Yes — and the evidence spans theory, controlled experiments, and connections to practice. We walk through the full chain:
>
> 1. **Analytical proof for softmax.** In Appendix C (Theorem C.1), we formally prove that our multi-head construction works under standard scaled-softmax with sum-over-heads aggregation. Proposition C.4 shows that a single head requires asymptotically more total key dimension than multiple heads by a factor of Ω(m / (d_model · log m)) in the compressive regime. The mechanism is the same: more heads → smaller blocks → less de-embedding interference per head.
>
> 2. **Softmax experiments.** Appendix A.1 (Figure 5) confirms empirically that the multi-head advantage persists under softmax, with the capacity threshold shifting right consistent with the extra log factor in our softmax construction.
>
> 3. **Value channel experiments.** Appendix A.2 adds an OV pathway and trains on message *retrieval* (not just edge classification). The multi-head advantage remains — showing it is not an artifact of our thresholded classification objective.
>
> 4. **Full Transformer block.** Appendix A.3 tests a complete pre-LN Transformer with frozen GPT-2 embeddings on an induction-style task. Single-head models exhibit bimodal failure and saturate far below multi-head performance (Figure 7). This is our most realistic setting.
>
> 5. **Permutation graphs throughout.** All of these experiments study scenarios where each query has exactly one correct target. This means the multi-head advantage *cannot* be explained by "attending to multiple things at once," which is the conventional justification for MHA. The noise-suppression explanation is the only one consistent with all the evidence.
>
> 6. **Independent empirical support.** Zhong et al. (2025) observe in GPT-2 training that collapsing to fewer heads improves linear attention but degrades softmax attention — exactly matching our prediction that competitive nonlinearity enables multi-head noise suppression.
>
> We will add forward pointers from the main text to make the softmax results and the full evidence chain more visible in the camera-ready revision.
>
> **On undefined mathematical symbols.**
>
> We will add definitions for all symbols at first use, both in the abstract and the main text, and include a notation summary table.

---

> > ### Author Rebuttal · Reviewer_gpPQ · 2026-04-04
> >
> > Thank you for the clarification. This largely addresses my concern. I still have one small question: for standard softmax attention under a fixed total dimension budget, could the authors state more explicitly what exactly remains true compared with the main result? In particular, is the same multi-head advantage preserved up to an extra log factor, or does the threshold/scaling change more substantially? It would also help to clarify what “noise suppression” corresponds to mathematically in the softmax setting, and where this appears in the proof.

---

> > > ### Author Response · Authors · 2026-04-04
> > >
> > > Thank you for the thoughtful follow-up. We are happy to state this precisely.
> > >
> > > **What exactly holds under softmax.** For permutation graphs with standard scaled-softmax and sum-over-heads aggregation, the capacity law becomes DK = Θ(m (log m)² / d_model), compared with Θ(m log m / d_model) for max-over-heads. The difference is exactly one extra log m factor — the scaling does not change more substantially than that. The multi-head advantage is preserved: Proposition C.4 shows that a single head requires total key dimension larger than the multi-head construction by a factor of Ω(m / (d_model · log m)). This ratio remains asymptotically large throughout the compressive regime m ≫ d_model.
> > >
> > > **What "noise suppression" means mathematically in the softmax setting.** The mechanism is identical to the max-over-heads case. The score for source i and target j in head k decomposes as S^(k)_{ij} = Signal + N1 + N2 + N3, where N1, N2, N3 arise from de-embedding leakage. The dominant noise term N3 scales with the block size B (number of targets assigned to a head) as N3 ≍ (B/d_model)√(d_k log m) — see Eq. (1). More heads reduce B, which reduces N3, which is the noise suppression.
> > >
> > > In the softmax setting, this raw-score separation is then converted into probability mass via softmax calibration (Lemma C.2, Step 2 of the proof of Theorem C.1). The extra log factor arises because softmax concentration against ℓ distractors requires the logit gap to be Ω(√d_k), which forces d_k = Ω((log m)²) — a per-head floor that is absent under max-over-heads. Crucially, this softmax-specific floor and the de-embedding noise constraint (Eq. 2) simply stack (see Eq. 22 in Proposition C.4): d_k ≥ max{C₁(log ℓ)², C₂ B²/(d²_model) · log m}. The first term is the softmax floor; the second is the noise-suppression constraint that drives the multi-head advantage. Multiple heads reduce B and thus reduce the second term, exactly as in the max-over-heads case.
> > >
> > > **Where this appears.** The full softmax construction and proof are in Appendix C: Theorem C.1 (Steps 1–6) proves correctness, and Proposition C.4 gives the explicit single-head vs. multi-head comparison under the combined constraint (Eq. 22).

---

### Official Review · Reviewer_WtHa · 2026-03-13

**Soundness:** 3
**Presentation:** 2
**Significance:** 3
**Originality:** 3
**Overall Recommendation:** 4
**Confidence:** 3

**Summary:**

This paper explore lower bounds on the attention QK embedding dimension needed for the attention matrices to capture permutation graphs. The main focus is on explaining the effectiveness of multi-head attention. It is shown that in capturing permutation graphs (in their assumed embedding setups), multi-head attention requires lower embedding dimension (D_k = h x d_k), then in the case of single-head attention.

**Compliance With Llm Reviewing Policy:**

Affirmed.

**Final Justification:**

The authors have addressed my concerns. I always liked their problem setup and their results, and my main complaint was regarding presentation.

**Key Questions For Authors:**

1. While I understand the motivation behind the algorithms chosen in Section 4; can the authors reason more precisely that these mechanisms would actually be used by real attention networks?

**Limitations:**

yes

**Strengths And Weaknesses:**

#### Strengths

1. By choosing to work on permutation graphs, the authors start with a well motivated setup to attack this problem. Permutation graphs *should be* the worst situation to find it useful to do multi-head attention in. And so, the case for multi-head attention is quite convincing if they are found helpful for performing pure permutations, “where parallel attention to multiple heads is not helpful.”


#### Weaknesses

1. Although the permutation graph setup is well motivated, I still find it necessary to make a connection to dense graphs, which this paper does not seem to do quite well, atleast in the main text. I see quite interesting results presented in the Appendix sections, and I would suggest some of it be moved to the main text (ICML guidelines suggest that the main text should stand on its own, and reviewers are not required to view the supplementary appendices).


2. The plots do not look professional, and should be improved. For example, label and title font sizes on the plots are inconsistent throughout the paper.

3. The full-transformer example (Appendix A.3) does not seem easily understandable. For example, the term $l$ and $M$ were introduced, without any introduction as to what they are.

---

> ### Author Rebuttal · Authors · 2026-03-28
>
> **On the connection to dense graphs.**
>
> We agree with the reviewer that permutation graphs are well-motivated for isolating the multi-head advantage in the absence of parallel-attention benefits — and that the connection to denser graphs deserves more prominence.
>
> Appendix B.3 and E.4 contain substantial dense-graph results that we will move into the camera-ready main text. Specifically:
>
> - **Capacity scales with edges, not vertices.** For r-regular graphs with r ∈ {1, 2, 4, 8, 16} (at m = 128, d_model = 32), the empirically minimal D*_K collapses to a single linear trend when plotted against the edge-normalized quantity x = (rm / d_model) log₂ m. A one-parameter fit D*_K ≈ 0.46x holds across all densities (Figure 12). This means doubling the number of edges roughly doubles the required key budget, regardless of whether the extra edges come from more vertices or higher degree — edges, not vocabulary size, are the operative constraint.
>
> - **The multi-head advantage grows with density.** The optimal head count at the capacity threshold increases with graph degree r, tracking m'/d_model up to a small constant factor (Figure 13). Denser graphs superpose more relationships per embedding coordinate, requiring finer partitioning across heads to manage interference.
>
> - **General construction with matching bounds.** Appendix E.4 provides Algorithm 5 and Theorem E.9 for arbitrary directed graphs, achieving DK = O((m'/d_model + Δ) log m), where Δ is the maximum degree. For graphs with bounded degree skew (Δ/d_avg ≤ m/d_model — which includes all regular graphs), this simplifies to DK = Θ((m' log m) / d_model) and matches our information-theoretic lower bound up to constants for m' = O(m^{2−ε}).
>
> - **Sharp phase transitions persist.** The step-like F1 transition observed for permutation graphs carries over to denser graphs without blurring — it simply shifts rightward with increasing edge count (Figure 11).
>
> We will move the key figures (11, 12, 13) and the general-graph theorem statement into the main text in the camera-ready revision so that it stands on its own per ICML guidelines.
>
> **On plot quality.**
>
> We acknowledge the inconsistent formatting and will standardize font sizes, labels, and styling across all figures for the camera-ready version.
>
> **On the full-Transformer example (Appendix A.3).**
>
> We apologize for the unclear notation. We will add a self-contained notation table and ensure all symbols are defined before use in that section. We will also move core elements of A.3 into the main text in the camera-ready revision, as it is one of our strongest robustness results — demonstrating that the multi-head advantage persists in a full pre-LN Transformer block with real (non-orthogonal) GPT-2 embeddings, on a task that never requires attending to multiple positions.
>
> **On whether real attention networks would use these mechanisms.**
>
> Our constructions are existence proofs establishing that the QK channel *can* solve RGR within the stated budget, thereby upper-bounding capacity requirements. The fact that our empirically trained models (using standard gradient descent with no knowledge of the constructions) achieve thresholds closely matching the constructive bounds (R² = 0.992 after excluding finite-size violations, Figure 3) suggests that optimization finds solutions with similar scaling properties. Furthermore, the de-embedding + signature structure of our constructions mirrors what mechanistic interpretability finds in practice: heads that project into subspaces where specific token relationships produce high dot products, while suppressing interference from other relationships (Kamath et al., 2025). We will discuss this connection more explicitly in the camera-ready revision.

---

> > ### Author Rebuttal · Reviewer_WtHa · 2026-04-06
> >
> > I thank the authors for their responses. My concerns have been resolved (or will be once the authors move their key results to the main text, improve plots, and add a notation table), and I will raise my score to a 4.

---

### Official Review · Reviewer_5kM5 · 2026-03-13

**Soundness:** 3
**Presentation:** 2
**Significance:** 2
**Originality:** 3
**Overall Recommendation:** 4
**Confidence:** 4

**Summary:**

This paper studies the capacity of the key–query channel in self-attention, asking how many token–token relations a single attention layer can reliably represent under a fixed attention budget.

It introduces Relational Graph Recognition, which frames attention as recovering edges of a relational graph among tokens in a context.

It proves information-theoretic lower bounds on the required key dimension and provide constructive upper bounds, showing near-matching scaling with the number of relations and embedding dimension.

**Compliance With Llm Reviewing Policy:**

Affirmed.

**Final Justification:**

see the rebuttal acknowledgement

**Key Questions For Authors:**

Q1. the terminology of relation graph confuses me. do you mean the subject-predicate-object style relational graph, which is a standard terminology in knowledge graph literature, or do you mean the G itself defines relationship among nodes?

Q2. can we verify the conclusion with the MAH in llama models? or do we need to simplify the model a bit?

Q3. what do you mean by embedding superposition?

**Limitations:**

yes

**Strengths And Weaknesses:**

Strengths

- the paper has clear theoretical framing of attention capacity via the RGR abstraction.

- it estabilishes strong theoretical results with both lower bounds and constructive upper bounds.

- it provides a compelling capacity-based explanation for multi-head attention as a mechanism to reduce interference.


Weaknesses

- analysis relies on a simplified attention model; applicability to full transformers is indirect.

- experiments are mostly controlled or synthetic rather than task-level evaluations or more practical models.

---

> ### Author Rebuttal · Authors · 2026-03-28
>
> We thank the reviewer for recognizing the theoretical contributions. We address the noted weaknesses first, then the three questions.
>
> **On the simplified attention model and indirect applicability to full Transformers.**
>
> We acknowledge that our core analytical results use a simplified model — this is deliberate and, we believe, necessary to obtain tight bounds. However, the applicability to full Transformers is more direct than it may appear, for three reasons:
>
> - Our **lower bounds** (Appendix D, Theorem D.2) apply to *any* attention mechanism with constant-margin separation — including standard softmax with sum aggregation (see the "Softmax attention" paragraph following Theorem D.2). These are not artifacts of our simplified model.
>
> - We provide a **formal proof for standard scaled-softmax attention** (Appendix C, Theorem C.1), with sum-over-heads aggregation. The multi-head advantage is formally established in this setting as well (Proposition C.4).
>
> Third, we systematically validate that the phenomena predicted by the simplified analysis survive in progressively more realistic architectures: softmax normalization (Appendix A.1), an added OV/value pathway (Appendix A.2), and a **full pre-LN Transformer block with frozen GPT-2 embeddings** (Appendix A.3). In this last setting — which includes residual connections, layer normalization, an MLP, and real (non-orthogonal) embeddings — the multi-head advantage remains strong (Figure 7). The simplified model is the starting point for analysis, but the conclusions hold well beyond it.
>
> **On synthetic/controlled experiments vs. task-level evaluations.**
>
> The controlled setting is essential for testing *scaling laws*: we need to independently vary graph structure, embedding geometry, and budget to verify predicted functional relationships — something that is not possible with end-to-end LLM benchmarks where these quantities are confounded.
>
> That said, in addition to the full-Transformer experiment described above, our results connect to practice in several ways:
>
> - Section 2.1 shows that our theory provides a **unified explanation for multiple phenomena observed in trained LLMs** that previously lacked a formal account: why LLMs converge on many small heads (GPT-3: 96 heads), why MQA/GQA trades and restores quality, why softmax outperforms linear attention, and why many heads can be pruned while a small subset is critical.
>
> - Zhong et al. (2025) provide **independent empirical confirmation** in GPT-2 training: collapsing to fewer heads improves linear-kernel attention but degrades standard softmax attention — precisely as our noise-suppression theory predicts.
>
> We will expand these connections in the revised main text.
>
> **Q1: Terminology — "relational graph."**
>
> Our use of "relational graph" does not refer to subject-predicate-object triples or knowledge graphs. Rather, G = (V, E) encodes the *complete set of pairwise relationships that an attention layer must be able to recover*. The vertices V represent the full vocabulary of possible items (all m tokens/features that could appear in any context), not just the tokens present in a particular context. The edges E encode which items should attend to which: (u, v) ∈ E means "when u and v co-occur in a context, u should attend to v." Given a specific context C ⊆ V, the attention layer must recover exactly those edges of G whose endpoints both appear in C.
>
> This formulation is one of the paper's central contributions beyond the MHA rationale. It provides a clean abstraction for reasoning about attention capacity and we believe has broad utility for analyzing attention beyond the questions studied here.  We will sharpen the terminology in the revision.
>
> **Q2: Verifying the multi-head advantage in LLaMA models.**
>
> Directly measuring capacity in a pre-trained LLM is challenging for the reasons we discuss in the Introduction (superposition, polyfunctional heads, context-dependence). However, several indirect lines of evidence already exist and are consistent with our predictions: LLaMA 3 uses GQA (grouped query attention), which our theory frames as a capacity knob (Section 2.1); Zhong et al. (2025) observe the head-count tradeoff empirically in GPT-2 training. A targeted experiment would involve training models at LLaMA scale with varying (h, d_k) at fixed D_K on controlled tasks. We view this as valuable future work and will discuss its feasibility in the revision.
>
> **Q3: Embedding superposition.**
>
> When d_model < m, the m item embeddings cannot be orthogonal in R^{d_model} — they must share subspace directions. This is what we mean by "superposition": the embedding of item i carries spurious correlations with other items' embeddings. In our formal analysis, this manifests as interference noise in the attention scores. The multi-head advantage arises precisely because partitioning items across heads reduces the noise each head must contend with. We will add a clearer definition of this term early in the paper.

---

> > ### Author Rebuttal · Reviewer_5kM5 · 2026-04-04
> >
> > Thank you for the clarifications. They helped improve my understanding of the paper, particularly regarding the applicability to softmax attention and the definition of the relational graph.
> >
> > I still do not feel I fully grasp all technical details, but I have higher confidence that the core results are sound and relevant. Overall, I am comfortable maintaining my score, with slightly increased confidence.

---

### Official Review · Reviewer_EFCZ · 2026-03-14

**Soundness:** 2
**Presentation:** 2
**Significance:** 2
**Originality:** 3
**Overall Recommendation:** 4
**Confidence:** 3

**Summary:**

This paper studies the representational capacity of the self attention key-query channel through a framework named Relational Graph Recognition where a single attention layer must recover edges of a directed graph across arbitrary contexts. The complexity measure is the total key dimension D_K = h.dk. The paper includes matching information-theoretic lower bounds and constructive upper bounds showing DK must grow as m'/ d_model (up to logarithmic factors), where m' is the number of edges and d_model the embedding dimension. A key contribution is a capacity based rationale for multi head attention: even for permutation graphs where each query has a single target, splitting a fixed DK budget across multiple heads reduces interference from embedding superposition and increases capacity. Experiments validate the predicted scaling laws and phase transitions, and the multi-head advantage persists under softmax, value routing, and a full Transformer block with frozen GPT-2 embeddings.

**Compliance With Llm Reviewing Policy:**

Affirmed.

**Key Questions For Authors:**

na

**Limitations:**

included above

**Strengths And Weaknesses:**

Strengths:
- The paper's key contribution is clear and well executed. The matching lower and upper bounds for permutation graphs under the max over heads model are tight up to constant factors and the information-theoretic lower bound via metric entropy applies to both model variants, giving the results a satisfying completeness.
- The multi-head advantage result is the paper's most practically valuable finding, though parts of the understandinf why multi-head is preferred is known.
- The connection to observed LLM behaviors in Section 2.1: many small heads in GPT-3, MQA/GQA as a capacity knob, pruning/redundancy patterns, softmax vs. linear attention, is a good piece of contextual grounding that makes the theoretical results feel actionable rather than purely abstract.

Weaknesses and concerns:
- The max-over-heads aggregation is a significant departure from standard attention. While the paper justifies it as a tractable nonlinear surrogate for softmax's competitive routing, max aggregation is not used in any practical Transformer architecture and may have fundamentally different behavior: it suppresses all but the highest-scoring head for a given pair, whereas softmax sum across heads allows collaborative signal accumulation.
- The framework isolates the QK channel entirely, omitting the OV pathway by design. While it is argued the QK channel performs the routing decision and OV cannot fix systematic mis-routing this separation is not always clean in practice. Recent mechanistic interpretability work (Kamath et al. 2025 in the paper) suggests that QK and OV are coupled and participate jointly in trained circuits, so the clean separation in the RGR abstraction is only approximate.
- The evaluations are almost entirely synthetic without real data implications discussed.

---

> ### Author Rebuttal · Authors · 2026-03-28
>
> **On max-over-heads as a departure from standard attention.**
>
> We appreciate this concern. Max-over-heads is not proposed as a model of practice — it is a *tractable analytical surrogate* that preserves the key nonlinearity (suppression of cross-head interference) that makes multi-head attention (MHA) distinct from a single wide head. As we show in Section 3, purely linear aggregation collapses multi-head to single-head (Cordonnier et al., 2020a), so *some* nonlinearity is essential to study the multi-head advantage. Max is the minimal such choice.
>
> Importantly, our theoretical results are not limited to this surrogate. For **standard scaled-softmax** attention with sum-over-heads aggregation, we prove (Appendix C, Theorem C.1) that DK = Θ(m(log m)² / d_model) suffices, and we formally establish (Proposition C.4) that the multi-head advantage persists: in the compressive regime, the single-head requirement exceeds the multi-head requirement by a factor of Ω(m / (d_model · log m)). The noise-suppression mechanism is the same in both variants — partitioning items across heads reduces de-embedding interference per head — and the softmax construction achieves near-matching bounds with only an extra log factor.  The MHA advantage arises even in permutation graphs, where attending to multiple targets in parallel cannot be helpful.
>
> Empirically, we confirm this progression across four model variants: (1) max-over-heads (Section 6), (2) scaled-softmax with sum aggregation (Appendix A.1, Figure 5), (3) softmax with a value channel trained on message retrieval (Appendix A.2), and (4) a full Transformer block with frozen GPT-2 embeddings (Appendix A.3). In every case, the multi-head advantage appears at fixed total key dimension DK — and in every case, this advantage arises in scenarios where each query has exactly one correct target (i.e., permutation graphs), so the benefit of MHA cannot be attributed to attending to multiple targets in parallel.
>
> The reviewer notes that softmax "allows collaborative signal accumulation across heads." Our softmax analysis (Appendix C) accounts for exactly this: non-owner heads contribute a near-uniform background mass that sums across heads, and we show this background is controlled and does not disrupt the threshold.We will make this progression more prominent in the camera-ready main text.
>
> **On QK/OV separation.**
>
> We agree this separation is approximate in trained models. Our claim is narrower: for the *addressing* question ("where to attend"), the QK channel is the gating bottleneck — OV can amplify signal but cannot fix systematic mis-routing. The Kamath et al. (2025) work the reviewer cites actually supports this decomposition at the mechanistic level. Moreover, our Appendix A.2 and A.3 experiments add back the OV pathway and train end-to-end, and **the multi-head advantage persists in both cases** — providing direct empirical evidence that the QK-centric analysis captures the operative constraint even when OV is present and jointly trained.
>
> **On synthetic evaluations.**
>
> The synthetic setting is by design: it lets us control graph structure, embedding geometry, and budget independently — which is necessary to test scaling *laws* rather than aggregate performance. That said, we provide several connections to realistic settings:
>
> - Appendix A.3 uses **frozen GPT-2 embeddings** on an induction-style retrieval task in a full Transformer block — arguably the most realistic single-layer capacity evaluation possible — and the multi-head advantage remains strong (Figure 7).
>
> - Section 2.1 connects our predictions to widely observed LLM behaviors. Our theory offers a unified explanation for multiple phenomena that have been observed in practice but lacked a formal account: (a) why LLMs use many small heads rather than fewer large ones (GPT-3: 96 heads); (b) why MQA/GQA trades quality for speed and restoring KV groups recovers quality; (c) why softmax attention outperforms linear attention variants; and (d) why many heads can be pruned while a small specialized subset is critical.
>
> - We also describe in Section 2.1 how Zhong et al. (2025) provide independent empirical support: in GPT-2 training, they observe that collapsing to fewer heads *improves* linear-kernel attention but *degrades* standard softmax attention — exactly as our theory predicts. They also observe that multi-head outperforms single-head and hypothesize (without formalizing) that interference is the reason. Our work provides the formal framework and quantitative scaling laws that explain these observations.
>
> We will expand the discussion of these connections in the camera-ready revision.

---

> > ### Author Rebuttal · Reviewer_EFCZ · 2026-04-06
> >
> > Thank you for the response which addresses the key considerations and particularly the softmax extension in Appendix C and the GPT2 embedding experiments. Given the theoretical contributions across model variants, I maintain my current score.

---

### Decision · Program_Chairs · 2026-04-30

**Decision:**

Accept (regular)

**Comment:**

This paper studies the representational capacity of the attention layer, through a proposed "graph recognition" problem. In this problem, they derive upper and lower bounds on the representational capacity. An insight that falls out of this is that having multiple heads is useful as it prevents superposition of signals. The paper also presents experiments in support of this.

This is an interesting and strong paper. All reviewers appreciate the theory and specifically, its clarity (`EFCZ`, `5kM5`) and execution (`EFCZ`); the lower and upper bound analysis is strong (`5kM5`) and provides a satisfying understanding ( `EFCZ`).  The advantage of the multi-head attention in particular is a nice insight that all reviewers appreciated as valuable (`EFCZ`), as a compelling explanation  (`5kM5`). `WtHa` specifically notes how the  proposed permutation graph setting (where each source node is mapped to only one target node) is the most insightful setting in showcasing benefits of multi-head attention, as it's where one would least expect its benefits --- from my read of the setting, we agree with this. The setting is indeed elegant enough to gain the high-level insight with little effort.


There were a few concerns that were resolved in the rebuttal, but the ones that remain largely are the fact that evaluations are synthetic/controlled, and there are some simplifications in the theoretical analysis such as the choice of the max-aggregation over the attention heads; such simplifications or abstractions in my opinion are inevitable to make analyses possible; they also carefully and honestly describe this abstraction which is a good way to present the finding since it enables building on top of this result/debugging one's understanding etc., Finally, the experiments and the connections to known behaviors grounds the results (as `EFCZ` notes); the authors seem to have experiments in A.1 to address their max-aggregation assumption;

We would also like to alert the authors of a non-existent reference in the paper: Reference: Vardi, G., Yehudai, G., and Shamir, O. Memorization thresholds in deep neural networks. arXiv preprint arXiv:2002.10211, 2020. **Kindly fix this.**
Otherwise, we hope the authors will incorporate the feedback from the reviewers which seem helpful.

Overall, we strongly recommend publishing this at the conference and we thank the authors for the well-done submission.